# Improved Algorithms for Convex-Concave Minimax Optimization

**Yuanhao Wang**
Computer Science Department
Princeton University
yuanhao@princeton.edu

**Jian Li**
Institute for Interdisciplinary Information Sciences
Tsinghua University
lijian83@mail.tsinghua.edu.cn

## Abstract

This paper studies minimax optimization problems $\min_{\mathbf{x}} \max_{\mathbf{y}} f(\mathbf{x}, \mathbf{y})$, where $f(\mathbf{x}, \mathbf{y})$ is $m_{\mathbf{x}}$-strongly convex with respect to $\mathbf{x}$, $m_{\mathbf{y}}$-strongly concave with respect to $\mathbf{y}$ and $(L_{\mathbf{x}}, L_{\mathbf{xy}}, L_{\mathbf{y}})$-smooth. Zhang et al. [42] provided the following lower bound of the gradient complexity for any first-order method: $\Omega\left(\sqrt{\frac{L_{\mathbf{x}}}{m_{\mathbf{x}}} + \frac{L_{\mathbf{xy}}^2}{m_{\mathbf{x}} m_{\mathbf{y}}} + \frac{L_{\mathbf{y}}}{m_{\mathbf{y}}}} \ln(1/\epsilon)\right)$. This paper proposes a new algorithm with gradient complexity upper bound $\tilde{O}\left(\sqrt{\frac{L_{\mathbf{x}}}{m_{\mathbf{x}}} + \frac{L \cdot L_{\mathbf{xy}}}{m_{\mathbf{x}} m_{\mathbf{y}}} + \frac{L_{\mathbf{y}}}{m_{\mathbf{y}}}} \ln(1/\epsilon)\right)$, where $L = \max\{L_{\mathbf{x}}, L_{\mathbf{xy}}, L_{\mathbf{y}}\}$. This improves over the best known upper bound $\tilde{O}\left(\sqrt{L^2/m_{\mathbf{x}} m_{\mathbf{y}}} \ln^3(1/\epsilon)\right)$ by Lin et al. [24]. Our bound achieves linear convergence rate and tighter dependency on condition numbers, especially when $L_{\mathbf{xy}} \ll L$ (i.e., when the interaction between $\mathbf{x}$ and $\mathbf{y}$ is weak). Via reduction, our new bound also implies improved bounds for strongly convex-concave and convex-concave minimax optimization problems. When $f$ is quadratic, we can further improve the upper bound, which matches the lower bound up to a small sub-polynomial factor.

## 1 Introduction

In this paper, we study the following minimax optimization problem

$$\min_{\mathbf{x} \in \mathbb{R}^n} \max_{\mathbf{y} \in \mathbb{R}^m} f(\mathbf{x}, \mathbf{y}). \tag{1}$$

This problem can be thought as finding the equilibrium in a zero-sum two-player game, and has been studied extensively in game theory, economics and computer science. This formulation also arises in many machine learning applications, including adversarial training [26, 37], prediction and regression problems [41, 38], reinforcement learning [12, 10, 29] and generative adversarial networks [15, 2].

We study the fundamental setting where $f$ is smooth, strongly convex w.r.t. $\mathbf{x}$ and strongly concave w.r.t. $\mathbf{y}$. In particular, we consider the function class $\mathcal{F}(m_{\mathbf{x}}, m_{\mathbf{y}}, L_{\mathbf{x}}, L_{\mathbf{xy}}, L_{\mathbf{y}})$, where $m_{\mathbf{x}}$ is the strong convexity modulus, $m_{\mathbf{y}}$ is the strong concavity modulus, $L_{\mathbf{x}}$ and $L_{\mathbf{y}}$ characterize the smoothness w.r.t. $\mathbf{x}$ and $\mathbf{y}$ respectively, and $L_{\mathbf{xy}}$ characterizes the interaction between $\mathbf{x}$ and $\mathbf{y}$ (see Definition 2). The reason to consider such a function class is twofold. First, the strongly convex-strongly concave setting is fundamental. Via reduction [24], an efficient algorithm for this setting implies efficient algorithms for other settings, including strongly convex-concave, convex-concave, and non-convex-concave settings. Second, Zhang et al. [42] recently proved a gradient complexity lower bound $\Omega\left(\sqrt{\frac{L_{\mathbf{x}}}{m_{\mathbf{x}}} + \frac{L_{\mathbf{xy}}^2}{m_{\mathbf{x}} m_{\mathbf{y}}} + \frac{L_{\mathbf{y}}}{m_{\mathbf{y}}}} \cdot \ln\left(\frac{1}{\epsilon}\right)\right)$, which naturally depends on the above parameters. [1]

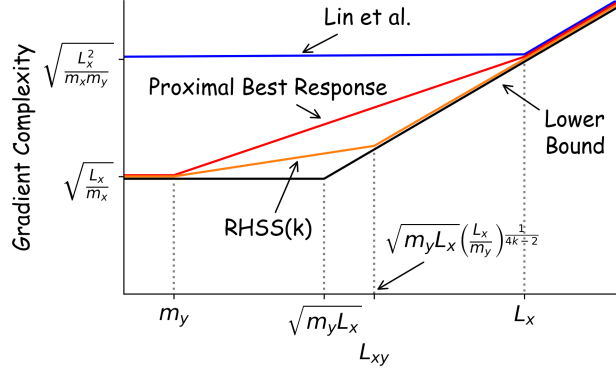

Figure 1: Comparison of previous upper bound [24], lower bound [42] and the results in this paper when $L_{\mathbf{x}} = L_{\mathbf{y}}$, $m_{\mathbf{x}} < m_{\mathbf{y}}$, ignoring logarithmic factors. The upper bounds and lower bounds are shown as a function of $L_{\mathbf{xy}}$ while other parameters are fixed.

In this setting, classic algorithms such as Gradient Descent-Ascent and ExtraGradient [22] can achieve linear convergence [40, 42]; however, their dependence on the condition number is far from optimal. Recently, Lin et al. [24] showed an upper bound of $\tilde{O}\left(\sqrt{L^2/m_{\mathbf{x}}m_{\mathbf{y}}}\ln^3(1/\epsilon)\right)$, which has a much tighter dependence on the condition number. In particular, when $L_{\mathbf{xy}} > \max\{L_{\mathbf{x}}, L_{\mathbf{y}}\}$, the dependence on the condition number matches the lower bound. However, when $L_{\mathbf{xy}} \ll \max\{L_{\mathbf{x}}, L_{\mathbf{y}}\}$, this dependence would no longer be tight (see Fig. 1 for illustration). In particular, we note that, when $x$ and $y$ are completely decoupled (i.e., $L_{\mathbf{xy}} = 0$), the optimal gradient complexity bound is $\Theta\left(\sqrt{L_{\mathbf{x}}/m_{\mathbf{x}} + L_{\mathbf{y}}/m_{\mathbf{y}}} \cdot \ln\left(1/\epsilon\right)\right)$ (the upper bound can be obtained by simply optimizing $\mathbf{x}$ and $\mathbf{y}$ separately). Moreover, Lin et al.'s result does not enjoy a linear rate, which may be undesirable if a high precision solution is needed.

In this work, we propose new algorithms in order to address these two issues. Our contribution can be summarized as follows.

1. For general functions in $\mathcal{F}(m_{\mathbf{x}}, m_{\mathbf{y}}, L_{\mathbf{x}}, L_{\mathbf{xy}}, L_{\mathbf{y}})$, we design an algorithm called Proximal Best Response (Algorithm 4), and prove a convergence rate of

$$\tilde{O}\left(\sqrt{\frac{L_{\mathbf{x}}}{m_{\mathbf{x}}} + \frac{L_{\mathbf{xy}} \cdot L}{m_{\mathbf{x}}m_{\mathbf{y}}} + \frac{L_{\mathbf{y}}}{m_{\mathbf{y}}}}\ln(1/\epsilon)\right).$$

It achieves linear convergence, and has a better dependence on condition numbers when $L_{\mathbf{xy}}$ is small (see Theorem 3 and the red line in Fig. 1).

2. We obtain tighter upper bounds for the strongly-convex concave problem and the general convex-concave problem, by reducing them to the strongly convex-strongly concave problem (See Corollary 1 and 2).

3. We also study the special case where $f$ is a quadratic function. We propose an algorithm called Recursive Hermitian-Skew-Hermitian Split (RHSS($k$)), and show that it achieves an upper bound of

$$O\left(\sqrt{\frac{L_{\mathbf{x}}}{m_{\mathbf{x}}} + \frac{L_{\mathbf{xy}}^2}{m_{\mathbf{x}}m_{\mathbf{y}}} + \frac{L_{\mathbf{y}}}{m_{\mathbf{y}}}}\left(\frac{L^2}{m_{\mathbf{x}}m_{\mathbf{y}}}\right)^{o(1)}\ln(1/\epsilon)\right).$$

Details can be found in Theorem 4 and Corollary 3. We note that the lower bound by Zhang et al. [42] holds for quadratic functions as well. Hence, our upper bound matches the gradient complexity lower bound up to a sub-polynomial factor.

## 2 Preliminaries

In this work we are interested in strongly-convex strongly-concave smooth problems. We first review some standard definitions of strong convexity and smoothness. A function $f : \mathbb{R}^n \to \mathbb{R}^m$ is

$L$-Lipschitz if $\forall \mathbf{x}, \mathbf{x}' \in \mathbb{R}^n \, \|f(\mathbf{x}) - f(\mathbf{x}')\| \leq L\|\mathbf{x} - \mathbf{x}'\|$. A function $f : \mathbb{R}^n \to \mathbb{R}^m$ is $L$-smooth if $\nabla f$ is $L$-Lipschitz. A differentiable function $\phi : \mathbb{R}^n \to \mathbb{R}$ is said to be $m$-strongly convex if for any $\mathbf{x}, \mathbf{x}' \in \mathbb{R}^n$, $\phi(\mathbf{x}') \geq \phi(\mathbf{x}) + (\mathbf{x}' - \mathbf{x})^T \nabla \phi(\mathbf{x}) + \frac{m}{2}\|\mathbf{x}' - \mathbf{x}\|^2$. If $m = 0$, we recover the definition of convexity. If $-\phi$ is $m$-strongly convex, $\phi$ is said to be $m$-strongly concave. For a function $f(\mathbf{x}, \mathbf{y})$, if $\forall \mathbf{y}$, $f(\cdot, \mathbf{y})$ is strongly convex, and $\forall \mathbf{x}$, $f(\mathbf{x}, \cdot)$ is strongly concave, then $f$ is said to be strongly convex-strongly concave.

**Definition 1.** A differentiable function $f : \mathbb{R}^n \times \mathbb{R}^m \to \mathbb{R}$ is said to be $(L_\mathbf{x}, L_\mathbf{xy}, L_\mathbf{y})$-smooth if

1. For any $\mathbf{y}$, $\nabla_\mathbf{x} f(\cdot, \mathbf{y})$ is $L_\mathbf{x}$-Lipschitz;   2. For any $\mathbf{x}$, $\nabla_\mathbf{y} f(\mathbf{x}, \cdot)$ is $L_\mathbf{y}$-Lipschitz;

3. For any $\mathbf{x}$, $\nabla_\mathbf{x} f(\mathbf{x}, \cdot)$ is $L_\mathbf{xy}$-Lipschitz;   4. For any $\mathbf{y}$, $\nabla_\mathbf{y} f(\cdot, \mathbf{y})$ is $L_\mathbf{xy}$-Lipschitz.

In this work, we are interested in functions that are strongly convex-strongly concave and smooth. Specifically, we study the following function class.

**Definition 2.** The function class $\mathcal{F}(m_\mathbf{x}, m_\mathbf{y}, L_\mathbf{x}, L_\mathbf{xy}, L_\mathbf{y})$ contains differentiable functions from $\mathbb{R}^n \times \mathbb{R}^m$ to $\mathbb{R}$ such that: 1. $\forall \mathbf{y}$, $f(\cdot, \mathbf{y})$ is $m_\mathbf{x}$-strongly convex; 2. $\forall \mathbf{x}$, $f(\mathbf{x}, \cdot)$ is $m_\mathbf{y}$-strongly concave; 3. $f$ is $(L_\mathbf{x}, L_\mathbf{xy}, L_\mathbf{y})$-smooth.

In the case where $f(\mathbf{x}, \mathbf{y})$ is twice continuously differentiable, denote the Hessian of $f$ at $(\mathbf{x}, \mathbf{y})$ by $\mathbf{H} := \begin{bmatrix} \mathbf{H}_\mathbf{xx} & \mathbf{H}_\mathbf{xy} \\ \mathbf{H}_\mathbf{yx} & \mathbf{H}_\mathbf{yy} \end{bmatrix}$. Then $\mathcal{F}(m_\mathbf{x}, m_\mathbf{y}, L_\mathbf{x}, L_\mathbf{xy}, L_\mathbf{y})$ can be characterized with the Hessian; in particular we require $m_\mathbf{x}\mathbf{I} \preccurlyeq \mathbf{H}_\mathbf{xx} \preccurlyeq L_\mathbf{x}\mathbf{I}$, $m_\mathbf{y}\mathbf{I} \preccurlyeq -\mathbf{H}_\mathbf{yy} \preccurlyeq L_\mathbf{y}\mathbf{I}$ and $\|\mathbf{H}_\mathbf{xy}\|_2 \leq L_\mathbf{xy}$.

For notational simplicity, we assume that $L_\mathbf{x} = L_\mathbf{y}$ when considering algorithms and upper bounds. This is without loss of generality, since one can define $g(\mathbf{x}, \mathbf{y}) := f((L_\mathbf{y}/L_\mathbf{x})^{1/4}\mathbf{x}, (L_\mathbf{x}/L_\mathbf{y})^{1/4}\mathbf{y})$ in order to make the two smoothness constants equal. It is not hard to show that this rescaling will not change $L_\mathbf{x}/m_\mathbf{x}$, $L_\mathbf{y}/m_\mathbf{y}$, $L_\mathbf{xy}$ and $m_\mathbf{x}m_\mathbf{y}$, and $L = \max\{L_\mathbf{x}, L_\mathbf{xy}, L_\mathbf{y}\}$ will not increase. Hence, we can make the following assumption without loss of generality. [2]

**Assumption 1.** $f \in \mathcal{F}(m_\mathbf{x}, m_\mathbf{y}, L_\mathbf{x}, L_\mathbf{xy}, L_\mathbf{y})$, and $L_\mathbf{x} = L_\mathbf{y}$.

The optimal solution of the convex-concave minimax optimization problem $\min_\mathbf{x} \max_\mathbf{y} f(\mathbf{x}, \mathbf{y})$ is the saddle point $(\mathbf{x}^*, \mathbf{y}^*)$ defined as follows.

**Definition 3.** $(\mathbf{x}^*, \mathbf{y}^*)$ is a saddle point of $f : \mathbb{R}^n \times \mathbb{R}^m \to \mathbb{R}$ if $\forall \mathbf{x} \in \mathbb{R}^n, \mathbf{y} \in \mathbb{R}^m$

$$f(\mathbf{x}, \mathbf{y}^*) \geq f(\mathbf{x}^*, \mathbf{y}^*) \geq f(\mathbf{x}^*, \mathbf{y}).$$

For strongly convex-strongly concave functions, it is well known that such a saddle point exists and is unique. Meanwhile, the saddle point is a stationary point, i.e. $\nabla f(\mathbf{x}^*, \mathbf{y}^*) = 0$, and is the minimizer of $\phi(\mathbf{x}) := \max_\mathbf{y} f(\mathbf{x}, \mathbf{y})$. For the design of numerical algorithms, we are satisfied with a close enough approximate of the saddle point, called $\epsilon$-saddle points.

**Definition 4.** $(\hat{\mathbf{x}}, \hat{\mathbf{y}})$ is an $\epsilon$-saddle point of $f$ if $\max_\mathbf{y} f(\hat{\mathbf{x}}, \mathbf{y}) - \min_\mathbf{x} f(\mathbf{x}, \hat{\mathbf{y}}) \leq \epsilon$.

Alternatively, we can also characterize optimality with the distance to the saddle point. In particular, let $\mathbf{z}^* := [\mathbf{x}^*; \mathbf{y}^*]$, $\hat{\mathbf{z}} := [\hat{\mathbf{x}}; \hat{\mathbf{y}}]$, then one may require $\|\hat{\mathbf{z}} - \mathbf{z}^*\| \leq \epsilon$. This implies that[3]

$$\max_\mathbf{y} f(\hat{\mathbf{x}}, \mathbf{y}) - \min_\mathbf{x} f(\mathbf{x}, \hat{\mathbf{y}}) \leq \frac{L^2}{\min\{m_\mathbf{x}, m_\mathbf{y}\}}\epsilon^2.$$

## 3   Related Work

There is a long line of work on the convex-concave saddle point problem. Apart from GDA and ExtraGradient [22, 40, 30, 14], other algorithms with theoretical guarantees include OGDA [36, 11, 28, 3], Hamiltonian Gradient Descent [1] and Consensus Optimization [27, 1, 3]. For the convex-concave case and strongly-convex-concave case, lower bounds have been proven by [33]. For the strongly-convex-strongly-concave case, the lower bound has been proven by [20] and [42]. Some authors have studied the special case where the interaction between $\mathbf{x}$ and $\mathbf{y}$ is bilinear [8, 9, 13] and variance reduction algorithms for finite sum objectives [7, 34]. The special case where $f$ is quadratic has also been studied extensively in the numerical analysis community [5, 6, 4].

**Algorithm 1** Alternating Best Response (ABR)

---

**Require:** $g(\cdot,\cdot)$, Initial point $\mathbf{z}_0 = [\mathbf{x}_0; \mathbf{y}_0]$, precision $\epsilon$, parameters $m_{\mathbf{x}}, m_{\mathbf{y}}, L_{\mathbf{x}}, L_{\mathbf{y}}$

$\quad \kappa_{\mathbf{x}} := L_{\mathbf{x}}/m_{\mathbf{x}}, \kappa_{\mathbf{y}} := L_{\mathbf{y}}/m_{\mathbf{y}}, T \leftarrow \left\lceil \log_2\left( \frac{4\sqrt{\kappa_{\mathbf{x}}+\kappa_{\mathbf{y}}}}{\epsilon} \right) \right\rceil$

$\quad$ **for** $t = 0, \cdots, T$ **do**

$\qquad$ Run AGD on $g(\cdot, \mathbf{y}_t)$ from $\mathbf{x}_t$ for $\Theta(\sqrt{\kappa_{\mathbf{x}}}\ln(\kappa_{\mathbf{x}}))$ steps to get $\mathbf{x}_{t+1}$

$\qquad$ Run AGD on $-g(\mathbf{x}_{t+1}, \cdot)$ from $\mathbf{y}_t$ for $\Theta(\sqrt{\kappa_{\mathbf{y}}}\ln(\kappa_{\mathbf{y}}))$ steps to get $\mathbf{y}_{t+1}$

$\quad$ **end for**

---

The convex-concave saddle point problem can also be seen as a special case of variational inequalities with Lipschitz monotone operators [30, 21, 14, 19, 40]. Some existing algorithms for the saddle point problem, such as ExtraGradient, achieve the optimal rate in this more general setting as well [30, 40].

Going beyond the convex-concave setting, some researchers have also studied the nonconvex-concave case recently [23, 39, 35, 24, 25, 31, 32], with the goal being finding a stationary point of the nonconvex function $\phi(\mathbf{x}) := \max_{\mathbf{y}} f(\mathbf{x}, \mathbf{y})$. By reducing to the strongly convex-strongly concave setting, [24] has achieved state-of-the-art results for nonconvex-concave problems.

## 4  Linear Convergence and Refined Dependence on $L_{\mathbf{xy}}$ in General Cases

### 4.1  Alternating Best Response

Let us first consider the extreme case where $L_{\mathbf{xy}} = 0$. In this case, there is no interaction between $\mathbf{x}$ and $\mathbf{y}$, and $f(\mathbf{x}, \mathbf{y})$ can be simply written as $h_1(\mathbf{x}) - h_2(\mathbf{y})$, where $h_1$ and $h_2$ are strongly convex functions. Thus, in this case, the following trivial algorithm solves the problem

$$\mathbf{x}^* \leftarrow \arg\min_{\mathbf{x}} f(\mathbf{x}, \mathbf{y}_0), \quad \mathbf{y}^* \leftarrow \arg\max_{\mathbf{y}} f(\mathbf{x}^*, \mathbf{y}).$$

In other words, the equilibrium can be found by directly playing the best response to each other once.

Now, let us consider the case where $L_{\mathbf{xy}}$ is nonzero but small. In this case, would the best response dynamics converge to the saddle point? Specifically, consider the following procedure:

$$\begin{cases} \mathbf{x}_{t+1} & \leftarrow \arg\min_{\mathbf{x}}\{f(\mathbf{x}, \mathbf{y}_t)\} \\ \mathbf{y}_{t+1} & \leftarrow \arg\max_{\mathbf{y}}\{f(\mathbf{x}_{t+1}, \mathbf{y})\} \end{cases}. \tag{2}$$

Let us define $\mathbf{y}^*(\mathbf{x}) := \arg\max_{\mathbf{y}} f(\mathbf{x}, \mathbf{y})$ and $\mathbf{x}^*(\mathbf{y}) := \arg\min_{\mathbf{x}} f(\mathbf{x}, \mathbf{y})$. Because $\mathbf{y}^*(\mathbf{x})$ is $L_{\mathbf{xy}}/m_{\mathbf{y}}$-Lipschitz and $\mathbf{x}^*(\mathbf{y})$ is $L_{\mathbf{xy}}/m_{\mathbf{x}}$-Lipschitz [4],

$$\|\mathbf{x}_{t+1} - \mathbf{x}^*\| = \|\mathbf{x}^*(\mathbf{y}_t) - \mathbf{x}^*(\mathbf{y}^*)\| \leq \frac{L_{\mathbf{xy}}}{m_{\mathbf{x}}}\|\mathbf{y}_t - \mathbf{y}^*\|$$

$$= \frac{L_{\mathbf{xy}}}{m_{\mathbf{x}}}\|\mathbf{y}^*(\mathbf{x}_t) - \mathbf{y}^*(\mathbf{x}^*)\| \leq \frac{L_{\mathbf{xy}}^2}{m_{\mathbf{x}}m_{\mathbf{y}}}\|\mathbf{x}_t - \mathbf{x}^*\|.$$

Thus, when $L_{\mathbf{xy}}^2 < m_{\mathbf{x}}m_{\mathbf{y}}$, (2) is indeed a contraction. In fact, we can further replace the exact solution of the inner optimization problems with Nesterov's Accelerated Gradient Descent (AGD) for constant number of steps, as described in Algorithm 1.

The following theorem holds for the Alternating Best Response algorithm. The proof of the theorem, as well as a detailed version of Algorithm 1 can be found in the Supplementary Material.

**Theorem 1.** *If $g \in \mathcal{F}(m_{\mathbf{x}}, m_{\mathbf{y}}, L_{\mathbf{x}}, L_{\mathbf{xy}}, L_{\mathbf{y}})$ and $L_{\mathbf{xy}} \leq \frac{1}{2}\sqrt{m_{\mathbf{x}}m_{\mathbf{y}}}$, Alternating Best Response returns $(\mathbf{x}_T, \mathbf{y}_T)$ such that*

$$\|\mathbf{x}_T - \mathbf{x}^*\| + \|\mathbf{y}_T - \mathbf{y}^*\| \leq \epsilon\left(\|\mathbf{x}_0 - \mathbf{x}^*\| + \|\mathbf{y}_0 - \mathbf{y}^*\|\right),$$

*and the number of gradient evaluations is bounded by (with $\kappa_{\mathbf{x}} = L_{\mathbf{x}}/m_{\mathbf{x}}$, $\kappa_{\mathbf{y}} = L_{\mathbf{y}}/m_{\mathbf{y}}$)*

$$O\left(\left(\sqrt{\kappa_{\mathbf{x}} + \kappa_{\mathbf{y}}}\right) \cdot \ln\left(\kappa_{\mathbf{x}}\kappa_{\mathbf{y}}\right) \ln\left(\kappa_{\mathbf{x}}\kappa_{\mathbf{y}}/\epsilon\right)\right).$$

Note that when $L_{\mathbf{xy}}$ is small, Zhang et al's lower bound [42] can be written as $\Omega\left(\sqrt{\kappa_{\mathbf{x}} + \kappa_{\mathbf{y}}}\ln(1/\epsilon)\right)$. Thus Alternating Best Response matches this lower bound up to logarithmic factors.

## 4.2 Accelerated Proximal Point for Minimax Optimization

In the previous subsection, we showed that Alternating Best Response matches the lower bound when the interaction term $L_{\mathbf{xy}}$ is sufficiently small. However, in order to apply the algorithm to functions with $L_{\mathbf{xy}} > \frac{1}{2}\sqrt{m_{\mathbf{x}}m_{\mathbf{y}}}$, we need another algorithmic component, namely the accelerated proximal point algorithm [16, 24].

For a minimax optimization problem $\min_{\mathbf{x}} \max_{\mathbf{y}} f(\mathbf{x}, \mathbf{y})$, define $\phi(\mathbf{x}) := \max_{\mathbf{y}} f(\mathbf{x}, \mathbf{y})$. Suppose that we run the accelerated proximal point algorithm on $\phi(\mathbf{x})$ with proximal parameter $\beta$: then the number of iterations can be easily bounded, while in each iteration one needs to solve a proximal problem $\min_{\mathbf{x}} \left\{ \phi(\mathbf{x}) + \beta \|\mathbf{x} - \hat{\mathbf{x}}_t\|^2 \right\}$. The key observation is that, this is equivalent to solving a minimax optimization problem $\min_{\mathbf{x}} \max_{\mathbf{y}} \left\{ f(\mathbf{x}, \mathbf{y}) + \beta \|\mathbf{x} - \hat{\mathbf{x}}_t\|^2 \right\}$. Thus, via accelerated proximal point, we are able to reduce solving $\min_{\mathbf{x}} \max_{\mathbf{y}} f(\mathbf{x}, \mathbf{y})$ to solving $\min_{\mathbf{x}} \max_{\mathbf{y}} \left\{ f(\mathbf{x}, \mathbf{y}) + \beta \|\mathbf{x} - \hat{\mathbf{x}}_t\|^2 \right\}$.

This is exactly the idea behind Algorithm 2 (the idea was also used in [24]). In the algorithm, $M$ is a positive constant characterizing the precision of solving the subproblem, where we require $M \geq \mathrm{poly}(\frac{L}{m_{\mathbf{x}}}, \frac{L}{m_{\mathbf{y}}}, \frac{\beta}{m_{\mathbf{x}}})$. If $M \to \infty$, the algorithm exactly becomes an instance of accelerated proximal point on $\phi(\mathbf{x}) = \max_{\mathbf{y}} f(\mathbf{x}, \mathbf{y})$.

---

**Algorithm 2** Accelerated Proximal Point Algorithm for Minimax Optimization

---

**Require:** Initial point $\mathbf{z}_0 = [\mathbf{x}_0; \mathbf{y}_0]$, proximal parameter $\beta$, strongly-convex modulus $m_{\mathbf{x}}$

$\quad \hat{\mathbf{x}}_0 \leftarrow \mathbf{x}_0, \kappa \leftarrow \beta/m_{\mathbf{x}}, \theta \leftarrow \frac{2\sqrt{\kappa}-1}{2\sqrt{\kappa}+1}, \tau \leftarrow \frac{1}{2\sqrt{\kappa}+4\kappa}$

$\quad$ **for** $t = 1, \cdots, T$ **do**

$\qquad$ Suppose $(\mathbf{x}_t^*, \mathbf{y}_t^*) = \min_{\mathbf{x}} \max_{\mathbf{y}} f(\mathbf{x}, \mathbf{y}) + \beta\|\mathbf{x} - \hat{\mathbf{x}}_{t-1}\|^2$. Find $(\mathbf{x}_t, \mathbf{y}_t)$ such that

$$\|\mathbf{x}_t - \mathbf{x}_t^*\| + \|\mathbf{y}_t - \mathbf{y}_t^*\| \leq \frac{1}{M} \left( \|\mathbf{x}_{t-1} - \mathbf{x}_t^*\| + \|\mathbf{y}_{t-1} - \mathbf{y}_t^*\| \right)$$

$\qquad \hat{\mathbf{x}}_t \leftarrow \mathbf{x}_t + \theta(\mathbf{x}_t - \mathbf{x}_{t-1}) + \tau(\mathbf{x}_t - \hat{\mathbf{x}}_{t-1})$

$\quad$ **end for**

---

The following theorem can be shown for Algorithm 2. The proof can be found in the Supplementary Material, and is based on the proof of Theorem 4.1 in [24].

**Theorem 2.** *The number of iterations needed by Algorithm 2 to produce* $(\mathbf{x}_T, \mathbf{y}_T)$ *such that*

$$\|\mathbf{x}_T - \mathbf{x}^*\| + \|\mathbf{y}_T - \mathbf{y}^*\| \leq \epsilon \left( \|\mathbf{x}_0 - \mathbf{x}^*\| + \|\mathbf{y}_0 - \mathbf{y}^*\| \right)$$

*is at most* ($\kappa = \beta/m_{\mathbf{x}}$)

$$\hat{T} = 8\sqrt{\kappa} \cdot \ln \left( \frac{28\kappa^2 L}{m_{\mathbf{y}}} \sqrt{\frac{L^2}{m_{\mathbf{x}}m_{\mathbf{y}}}} \cdot \frac{1}{\epsilon} \right). \tag{3}$$

## 4.3 Proximal Alternating Best Response

With the two algorithmic components, namely Alternating Best Response and Accelerated Proximal Point in place, we can now combine them and design an efficient algorithm for general strongly convex-strongly concave functions. The high-level idea is to exploit the accelerated proximal point algorithm twice to reduce a general problem into one solvable by Alternating Best Response.

To start with, let us consider a strongly-convex-strongly-concave function $f(\mathbf{x}, \mathbf{y})$, and apply Algorithm 2 for $f$ with proximal parameter $\beta = L_{\mathbf{xy}}$. By Theorem 2, the algorithm can converge in $\tilde{O}\left( \sqrt{\frac{L_{\mathbf{xy}}}{m_{\mathbf{x}}}} \right)$ iterations, while in each iteration we need to solve a regularized minimax problem

$$\min_{\mathbf{x}} \max_{\mathbf{y}} \left\{ f(\mathbf{x}, \mathbf{y}) + \beta \|\mathbf{x} - \hat{\mathbf{x}}_{t-1}\|^2 \right\}.$$

This is equivalent to $\min_{\mathbf{y}} \max_{\mathbf{x}} \left\{ -f(\mathbf{x}, \mathbf{y}) - \beta \|\mathbf{x} - \hat{\mathbf{x}}_{t-1}\|^2 \right\}$ [5], so we can apply Algorithm 2 once more to this problem with parameter $\beta = L_{\mathbf{xy}}$. This procedure would require $\tilde{O}\left( \sqrt{\frac{L_{\mathbf{xy}}}{m_{\mathbf{y}}}} \right)$ iterations,

and in each iteration, one needs to solve a minimax problem of the form

$$\min_{\mathbf{y}} \max_{\mathbf{x}} \left\{ -f(\mathbf{x}, \mathbf{y}) - \beta \|\mathbf{x} - \hat{\mathbf{x}}_{t-1}\|^2 + \beta \|\mathbf{y} - \hat{\mathbf{y}}_{t'-1}\|^2 \right\}$$
$$= - \min_{\mathbf{x}} \max_{\mathbf{y}} \left\{ f(\mathbf{x}, \mathbf{y}) + \beta \|\mathbf{x} - \hat{\mathbf{x}}_{t-1}\|^2 - \beta \|\mathbf{y} - \hat{\mathbf{y}}_{t'-1}\|^2 \right\}.$$

Hence, we reduced the original problem to a problem that is $2\beta$-strongly convex with respect to $\mathbf{x}$ and $2\beta$-strongly concave with respect to $\mathbf{y}$. Now the interaction between $\mathbf{x}$ and $\mathbf{y}$ is (relatively) much weaker and one can easily see that $L_{\mathbf{xy}} \leq \frac{1}{2}\sqrt{2\beta \cdot 2\beta}$. Consequently the final problem can be solved in $\tilde{O}\left(\frac{L_{\mathbf{x}}}{L_{\mathbf{xy}}}\right)$ gradient evaluations using the Alternating Best Response algorithm. We first consider the case where $L_{\mathbf{xy}} > \max\{m_{\mathbf{x}}, m_{\mathbf{y}}\}$. The total gradient complexity would thus be

$$\tilde{O}\left(\sqrt{\frac{L_{\mathbf{xy}}}{m_{\mathbf{x}}}}\right) \cdot \tilde{O}\left(\sqrt{\frac{L_{\mathbf{xy}}}{m_{\mathbf{y}}}}\right) \cdot \tilde{O}\left(\sqrt{\frac{L}{L_{\mathbf{xy}}}}\right) = \tilde{O}\left(\sqrt{\frac{L \cdot L_{\mathbf{xy}}}{m_{\mathbf{x}} m_{\mathbf{y}}}}\right).$$

In order to deal with the case where $L_{\mathbf{xy}} < \max\{m_{\mathbf{x}}, m_{\mathbf{y}}\}$, we shall choose $\beta_1 = \max\{L_{\mathbf{xy}}, m_{\mathbf{x}}\}$ for the first level of proximal point, and $\beta_2 = \max\{L_{\mathbf{xy}}, m_{\mathbf{y}}\}$ for the second level of proximal point. In this case, the total gradient complexity bound can be shown to be

$$\tilde{O}\left(\sqrt{\frac{\beta_1}{m_{\mathbf{x}}}}\right) \cdot \tilde{O}\left(\sqrt{\frac{\beta_2}{m_{\mathbf{y}}}}\right) \cdot \tilde{O}\left(\sqrt{\frac{L}{\beta_1} + \frac{L}{\beta_2}}\right) = \tilde{O}\left(\sqrt{\frac{L_{\mathbf{x}}}{m_{\mathbf{x}}} + \frac{L \cdot L_{\mathbf{xy}}}{m_{\mathbf{x}} m_{\mathbf{y}}} + \frac{L_{\mathbf{y}}}{m_{\mathbf{y}}}}\right).$$

A formal description of the algorithm is provided in Algorithm 4, and a formal statement of the complexity upper bound is provided in Theorem 3. The proof can be found in the Supplementary Material.

**Theorem 3.** *Assume that $f \in \mathcal{F}(m_{\mathbf{x}}, m_{\mathbf{y}}, L_{\mathbf{x}}, L_{\mathbf{xy}}, L_{\mathbf{y}})$. In Algorithm 4, the gradient complexity to produce $(\mathbf{x}_T, \mathbf{y}_T)$ such that $\|\mathbf{z}_T - \mathbf{z}^*\| \leq \epsilon$ is*

$$O\left(\sqrt{\frac{L_{\mathbf{x}}}{m_{\mathbf{x}}} + \frac{L \cdot L_{\mathbf{xy}}}{m_{\mathbf{x}} m_{\mathbf{y}}} + \frac{L_{\mathbf{y}}}{m_{\mathbf{y}}}} \cdot \ln^3\left(\frac{L^2}{m_{\mathbf{x}} m_{\mathbf{y}}}\right) \ln\left(\frac{L^2}{m_{\mathbf{x}} m_{\mathbf{y}}} \cdot \frac{\|\mathbf{z}_0 - \mathbf{z}^*\|}{\epsilon}\right)\right).$$

---

**Algorithm 3** APPA-ABR

---

**Require:** $g(\cdot, \cdot)$, Initial point $\mathbf{z}_0 = [\mathbf{x}_0; \mathbf{y}_0]$, precision parameter $M_1$
1: $\beta_2 \leftarrow \max\{m_{\mathbf{y}}, L_{\mathbf{xy}}\}, M_2 \leftarrow \frac{96 L^{2.5}}{m_{\mathbf{x}} m_{\mathbf{y}}^{1.5}}$
2: $\hat{\mathbf{y}}_0 \leftarrow \mathbf{y}_0, \kappa \leftarrow \beta_2/m_{\mathbf{y}}, \theta \leftarrow \frac{2\sqrt{\kappa}-1}{2\sqrt{\kappa}+1}, \tau \leftarrow \frac{1}{2\sqrt{\kappa}+4\kappa}, t \leftarrow 0$
3: **repeat**
4: $\quad t \leftarrow t + 1$
5: $\quad (\mathbf{x}_t, \mathbf{y}_t) \leftarrow \text{ABR}(g(\mathbf{x}, \mathbf{y}) - \beta_2 \|\mathbf{y} - \hat{\mathbf{y}}_{t-1}\|^2, [\mathbf{x}_{t-1}; \mathbf{y}_{t-1}], 1/M_2, 2\beta_1, 2\beta_2, 3L, 3L)$
6: $\quad \hat{\mathbf{y}}_t \leftarrow \mathbf{y}_t + \theta(\mathbf{y}_t - \mathbf{y}_{t-1}) + \tau(\mathbf{y}_t - \hat{\mathbf{y}}_{t-1})$
7: **until** $\|\nabla g(\mathbf{x}_t, \mathbf{y}_t)\| \leq \frac{\min\{m_{\mathbf{x}}, m_{\mathbf{y}}\}}{9LM_1} \|\nabla g(\mathbf{x}_0, \mathbf{y}_0)\|$

---

**Algorithm 4** Proximal Best Response

---

**Require:** Initial point $\mathbf{z}_0 = [\mathbf{x}_0; \mathbf{y}_0]$
1: $\beta_1 \leftarrow \max\{m_{\mathbf{x}}, L_{\mathbf{xy}}\}, M_1 \leftarrow \frac{80 L^3}{m_{\mathbf{x}}^{1.5} m_{\mathbf{y}}^{1.5}}$
2: $\hat{\mathbf{x}}_0 \leftarrow \mathbf{x}_0, \kappa \leftarrow \beta_1/m_{\mathbf{x}}, \theta \leftarrow \frac{2\sqrt{\kappa}-1}{2\sqrt{\kappa}+1}, \tau \leftarrow \frac{1}{2\sqrt{\kappa}+4\kappa}$
3: **for** $t = 1, \cdots, T$ **do**
4: $\quad (\mathbf{x}_t, \mathbf{y}_t) \leftarrow \text{APPA-ABR}(f(\mathbf{x}, \mathbf{y}) + \beta_1 \|\mathbf{x} - \hat{\mathbf{x}}_{t-1}\|^2, [\mathbf{x}_{t-1}, \mathbf{y}_{t-1}], M_1)$
5: $\quad \hat{\mathbf{x}}_t \leftarrow \mathbf{x}_t + \theta(\mathbf{x}_t - \mathbf{x}_{t-1}) + \tau(\mathbf{x}_t - \hat{\mathbf{x}}_{t-1})$
6: **end for**

---

## 4.4 Implications of Theorem 3

Theorem 3 improves over the results of Lin et al. in two ways. First, Lin et al.'s upper bound has a $\ln^3(1/\epsilon)$ factor, while our algorithm enjoys linear convergence. Second, our result has a better dependence on $L_{\mathbf{xy}}$. To see this, note that when $L_{\mathbf{xy}} \ll L$, $\frac{L_{\mathbf{x}}}{m_{\mathbf{x}}} + \frac{L \cdot L_{\mathbf{xy}}}{m_{\mathbf{x}} m_{\mathbf{y}}} + \frac{L_{\mathbf{y}}}{m_{\mathbf{y}}} \ll \frac{L_{\mathbf{x}}}{m_{\mathbf{x}}} + \frac{L^2}{m_{\mathbf{x}} m_{\mathbf{y}}} + \frac{L_{\mathbf{y}}}{m_{\mathbf{y}}} \le \frac{3L^2}{m_{\mathbf{x}} m_{\mathbf{y}}}$. This is also illustrated by Fig. 1, where Proximal Best Response (the red line) significantly outperforms Lin et al.'s result (the blue line) when $L_{\mathbf{xy}} \ll L$. In particular, Proximal Best Response matches the lower bound when $L_{\mathbf{xy}} > L_{\mathbf{x}}$ or when $L_{\mathbf{xy}} < \max\{m_{\mathbf{x}}, m_{\mathbf{y}}\}$; in between, it is able to gracefully interpolate the two cases.

As shown by Lin et al. [24], convex-concave problems and strongly convex-concave problems can be reduced to strongly convex-strongly concave problems. Hence, Theorem 3 naturally implies improved algorithms for convex-concave and strongly convex-concave problems.

**Corollary 1.** *If $f(\mathbf{x}, \mathbf{y})$ is $(L_{\mathbf{x}}, L_{\mathbf{xy}}, L_{\mathbf{y}})$-smooth and $m_{\mathbf{x}}$-strongly convex w.r.t. $\mathbf{x}$, via reduction to Theorem 3, the gradient complexity of finding an $\epsilon$-saddle point is $\tilde{O}\left(\sqrt{\frac{m_{\mathbf{x}} \cdot L_{\mathbf{y}} + L \cdot L_{\mathbf{xy}}}{m_{\mathbf{x}} \epsilon}}\right)$.*

**Corollary 2.** *If $f(\mathbf{x}, \mathbf{y})$ is $(L_{\mathbf{x}}, L_{\mathbf{xy}}, L_{\mathbf{y}})$-smooth and convex-concave, via reduction to Theorem 3, the gradient complexity to produce an $\epsilon$-saddle point is $\tilde{O}\left(\sqrt{\frac{L_{\mathbf{x}} + L_{\mathbf{y}}}{\epsilon}} + \frac{\sqrt{L \cdot L_{\mathbf{xy}}}}{\epsilon}\right)$.*

The precise statement as well as the proofs can be found in Section 6 of Supplementary Material. We remark that the reduction is for constrained minimax optimization, and Theorem 3 holds for constrained problems after simple modifications to the algorithm.

## 5 Near Optimal Dependence on $L_{\mathbf{xy}}$ in Quadratic Cases

We can see that proximal best response has near optimal dependence on condition numbers when $L_{\mathbf{xy}} > L_{\mathbf{x}}$ or when $L_{\mathbf{xy}} < \max\{m_{\mathbf{x}}, m_{\mathbf{y}}\}$. However, when $L_{\mathbf{xy}}$ falls in between, there is still a significant gap between the upper bound and the lower bound. In this section, we try to close this gap for quadratic functions; i.e. we assume that

$$f(\mathbf{x}, \mathbf{y}) = \frac{1}{2}\mathbf{x}^T \mathbf{A} \mathbf{x} + \mathbf{x}^T \mathbf{B} \mathbf{y} - \frac{1}{2}\mathbf{y}^T \mathbf{C} \mathbf{y} + \mathbf{u}^T \mathbf{x} + \mathbf{v}^T \mathbf{y}. \tag{4}$$

The reason to consider quadratic functions is threefold. First, the lower bound instance by [42] is a quadratic function; thus, this lower bound applies to quadratic functions as well, so it would be interesting to match the lower bound for quadratic functions first. Second, quadratic functions are considerably easier to analyze. Third, finding the saddle point of quadratic functions is an important problem on its own, and has many applications (see [6] and references therein).

Our assumption that $f \in \mathcal{F}(m_{\mathbf{x}}, m_{\mathbf{y}}, L_{\mathbf{x}}, L_{\mathbf{xy}}, L_{\mathbf{y}})$ now becomes assumptions on the singular values of matrices: $m_{\mathbf{x}}\mathbf{I} \preccurlyeq \mathbf{A} \preccurlyeq L_{\mathbf{x}}\mathbf{I}$, $m_{\mathbf{y}}\mathbf{I} \preccurlyeq \mathbf{C} \preccurlyeq L_{\mathbf{y}}\mathbf{I}$, $\|\mathbf{B}\|_2 \le L_{\mathbf{xy}}$. In this case, the unique saddle point is given by the solution to a linear system

$$\begin{bmatrix} \mathbf{x}^* \\ \mathbf{y}^* \end{bmatrix} = \mathbf{J}^{-1}\mathbf{b} = \begin{bmatrix} \mathbf{A} & \mathbf{B} \\ -\mathbf{B}^T & \mathbf{C} \end{bmatrix}^{-1} \begin{bmatrix} -\mathbf{u} \\ \mathbf{v} \end{bmatrix}.$$

Throughout this section we assume that $L_{\mathbf{x}} = L_{\mathbf{y}}$ and $m_{\mathbf{x}} < m_{\mathbf{y}}$, which are without loss of generality, and that $m_{\mathbf{y}} < L_{\mathbf{xy}}$, as otherwise proximal best response is already near-optimal.

### 5.1 Hermitian-Skew-Hermitian-Split

We now focus on how to solve the linear system $\mathbf{J}\mathbf{z} = \mathbf{b}$, where $\mathbf{J} := \begin{bmatrix} \mathbf{A} & \mathbf{B} \\ -\mathbf{B}^T & \mathbf{C} \end{bmatrix}$ is positive definite but not symmetric. A straightforward way to solve this asymmetric linear system is apply conjugate gradient to solve the normal equation $\mathbf{J}^T\mathbf{J}\mathbf{z} = \mathbf{J}^T\mathbf{b}$. However the complexity of this approach is $O\left(\frac{L}{\min\{m_{\mathbf{x}}, m_{\mathbf{y}}\}}\right)$, which is much worse than the lower bound. Instead, we utilize the Hermitian-Skew-Hermitian Split (HSS) algorithm [5], which is designed to solve positive definite

asymmetric systems. Define

$$\mathbf{G} := \begin{bmatrix} \mathbf{A} & 0 \\ 0 & \mathbf{C} \end{bmatrix}, \ \mathbf{S} := \begin{bmatrix} 0 & \mathbf{B} \\ -\mathbf{B}^T & 0 \end{bmatrix}, \ \mathbf{P} := \begin{bmatrix} \alpha \mathbf{I} + \beta \mathbf{A} & \\ & \mathbf{I} + \beta \mathbf{C} \end{bmatrix},$$

where $\alpha$ and $\beta$ are constants to be determined. Let $\mathbf{z}_t := [\mathbf{x}_t; \mathbf{y}_t]$. Then HSS runs as

$$\begin{cases} (\eta \mathbf{P} + \mathbf{G}) \mathbf{z}_{t+1/2} &= (\eta \mathbf{P} - \mathbf{S}) \mathbf{z}_t + \mathbf{b}, \\ (\eta \mathbf{P} + \mathbf{S}) \mathbf{z}_{t+1} &= (\eta \mathbf{P} - \mathbf{G}) \mathbf{z}_{t+1/2} + \mathbf{b}. \end{cases} \tag{5}$$

Here $\eta > 0$ is another constant. In this procedure, it can be shown that

$$\mathbf{z}_{t+1} - \mathbf{z}^* = (\eta \mathbf{P} + \mathbf{S})^{-1} (\eta \mathbf{P} - \mathbf{G}) (\eta \mathbf{P} + \mathbf{G})^{-1} (\eta \mathbf{P} - \mathbf{S}) (\mathbf{z}_t - \mathbf{z}^*).$$

The key observation of HSS is that the equation above is a contraction.

**Lemma 1** ([5]). *Define* $M(\eta) := (\eta \mathbf{P} + \mathbf{S})^{-1} (\eta \mathbf{P} - \mathbf{G}) (\eta \mathbf{P} + \mathbf{G})^{-1} (\eta \mathbf{P} - \mathbf{S})$. *Then*[6]

$$\rho(\mathbf{M}(\eta)) \leq \|\mathbf{M}(\eta)\|_2 \leq \max_{\lambda_i \in sp(\mathbf{P}^{-1}\mathbf{G})} \left| \frac{\lambda_i - \eta}{\lambda_i + \eta} \right| < 1.$$

Lemma 1 provides an upper bound on the iteration complexity of HSS, as in the original analysis of HSS [5]. However, it does not consider the computational cost per iteration. In particular, the matrix $\eta \mathbf{P} + \mathbf{S}$ is also asymmetric, and in fact corresponds to another quadratic minimax optimization problem. The original HSS paper did not consider how to solve this subproblem for general $\mathbf{P}$. Our idea is to solve the subproblem recursively, as explained in the next subsection.

## 5.2 Recursive HSS

---
**Algorithm 5** RHSS($k$) (Recursive Hermitian-skew-Hermitian Split)

---
**Require:** Initial point $[\mathbf{x}_0; \mathbf{y}_0]$, precision $\epsilon$, parameters $m_\mathbf{x}, m_\mathbf{y}, L_\mathbf{xy}$

$t \leftarrow 0, M_1 \leftarrow \frac{192L^5}{m_\mathbf{x}^2 m_\mathbf{y}^3}, M_2 \leftarrow \frac{16L_\mathbf{xy}}{m_\mathbf{y}}, \alpha \leftarrow \frac{m_\mathbf{x}}{m_\mathbf{y}}, \beta \leftarrow L_\mathbf{xy}^{-\frac{2}{k}} m_\mathbf{y}^{-\frac{k-2}{k}}, \eta \leftarrow L_\mathbf{xy}^{\frac{1}{k}} m_\mathbf{y}^{1-\frac{1}{k}}, \tilde{\epsilon} \leftarrow \frac{m_\mathbf{x}\epsilon}{L_\mathbf{xy}+L_\mathbf{x}}$

**repeat**

$$\begin{bmatrix} \mathbf{r}_1 \\ \mathbf{r}_2 \end{bmatrix} \leftarrow \begin{bmatrix} \eta(\alpha\mathbf{I} + \beta\mathbf{A}) & -\mathbf{B} \\ \mathbf{B}^T & \eta(\mathbf{I} + \beta\mathbf{C}) \end{bmatrix} \begin{bmatrix} \mathbf{x}_t \\ \mathbf{y}_t \end{bmatrix} + \begin{bmatrix} -\mathbf{u} \\ \mathbf{v} \end{bmatrix}$$

Use conjugate gradient with initial point $[\mathbf{x}_t; \mathbf{y}_t]$ and precision $1/M_1$ to solve

$$\begin{bmatrix} \mathbf{x}_{t+1/2} \\ \mathbf{y}_{t+1/2} \end{bmatrix} \leftarrow \begin{bmatrix} \eta(\alpha\mathbf{I} + \beta\mathbf{A}) + \mathbf{A} & \\ & \eta(\mathbf{I} + \beta\mathbf{C}) + \mathbf{C} \end{bmatrix}^{-1} \begin{bmatrix} \mathbf{r}_1 \\ \mathbf{r}_2 \end{bmatrix}$$

$$\begin{bmatrix} \mathbf{w}_1 \\ \mathbf{w}_2 \end{bmatrix} \leftarrow \begin{bmatrix} \eta\alpha\mathbf{I} + \eta\beta\mathbf{A} - \mathbf{A} & 0 \\ 0 & \eta(\mathbf{I} + \beta\mathbf{C}) - \mathbf{C} \end{bmatrix} \begin{bmatrix} \mathbf{x}_{t+1/2} \\ \mathbf{y}_{t+1/2} \end{bmatrix} + \begin{bmatrix} -\mathbf{u} \\ \mathbf{v} \end{bmatrix}$$

Call RHSS($k - 1$) with initial point $[\mathbf{x}_t; \mathbf{y}_t]$ and precision $1/M_2$ to solve

$$\begin{bmatrix} \mathbf{x}_{t+1} \\ \mathbf{y}_{t+1} \end{bmatrix} \leftarrow \begin{bmatrix} \eta(\alpha\mathbf{I} + \beta\mathbf{A}) & \mathbf{B} \\ -\mathbf{B}^T & \eta(\mathbf{I} + \beta\mathbf{C}) \end{bmatrix}^{-1} \begin{bmatrix} \mathbf{w}_1 \\ \mathbf{w}_2 \end{bmatrix}$$

$t \leftarrow t + 1$
**until** $\|\mathbf{J}\mathbf{z}_t - \mathbf{b}\| \leq \tilde{\epsilon} \|\mathbf{J}\mathbf{z}_0 - \mathbf{b}\|$

---

In this subsection, we describe our algorithm Recursive Hermitian-skew-Hermitian Split, or RHSS($k$), which uses HSS in $k - 1$ levels of recursion. Specifically, RHSS($k$) calls HSS with parameters $\alpha = m_\mathbf{x}/m_\mathbf{y}, \beta = L_\mathbf{xy}^{-\frac{2}{k}} m_\mathbf{y}^{-\frac{k-2}{k}}, \eta = L_\mathbf{xy}^{\frac{1}{k}} m_\mathbf{y}^{\frac{k-1}{k}}$. In each iteration, it solves two linear systems. The first one, which is associated with $\eta \mathbf{P} + \mathbf{G}$, can be solved with Conjugate Gradient [18] as $\eta \mathbf{P} + \mathbf{G}$ is symmetric positive definite. The second one is associated with

$$\eta\mathbf{P} + \mathbf{S} = \begin{bmatrix} \eta(\alpha\mathbf{I} + \beta\mathbf{A}) & \mathbf{B} \\ -\mathbf{B}^T & \eta(\mathbf{I} + \beta\mathbf{C}) \end{bmatrix},$$

which is equivalent to a quadratic minimax optimization problem. RHSS($k$) then makes a recursive call RHSS($k-1$) to solve this subproblem. When $k = 1$, we simply run the Proximal Best Response algorithm (Algorithm 4). A detailed description of RHSS($k$) for $k \geq 2$ is given in Algorithm 5.

Our main result for RHSS($k$) is the following theorem. Note that for an algorithm on quadratic functions, the number of matrix-vector products is the same as the gradient complexity.

**Theorem 4.** *There exists constants $C_1$, $C_2$, such that the number of matrix-vector products needed to find $(\mathbf{x}_T, \mathbf{y}_T)$ such that $\|\mathbf{z}_T - \mathbf{z}^*\| \leq \epsilon$ is at most*

$$\sqrt{\frac{L_{\mathbf{xy}}^2}{m_{\mathbf{x}} m_{\mathbf{y}}} + \left(\frac{L_{\mathbf{x}}}{m_{\mathbf{x}}} + \frac{L_{\mathbf{y}}}{m_{\mathbf{y}}}\right)\left(1 + \left(\frac{L_{\mathbf{xy}}}{\max\{m_{\mathbf{x}}, m_{\mathbf{y}}\}}\right)^{\frac{1}{k}}\right)} \cdot \left(C_1 \ln\left(\frac{C_2 L^2}{m_{\mathbf{x}} m_{\mathbf{y}}}\right)\right)^{k+3} \ln\left(\frac{\|\mathbf{z}_0 - \mathbf{z}^*\|}{\epsilon}\right).$$

$$(6)$$

If $k$ is chosen as a fixed constant, the comparison of (6) and the lower bound [42] is illustrated in Fig. 1. One can see that as $k$ increases, the upper bound of RHSS($k$) gradually fits the lower bound (as long as $k$ is a constant). By optimizing $k$, we can also show the following corollary.

**Corollary 3.** *When $k = \Theta\left(\sqrt{\ln\left(\frac{L^2}{m_{\mathbf{x}} m_{\mathbf{y}}}\right) / \ln\ln\left(\frac{L^2}{m_{\mathbf{x}} m_{\mathbf{y}}}\right)}\right)$, the number of matrix vector products that RHSS($k$) needs to find $\mathbf{z}_T$ such that $\|\mathbf{z}_T - \mathbf{z}^*\| \leq \epsilon$ is*

$$\sqrt{\frac{L_{\mathbf{xy}}^2}{m_{\mathbf{x}} m_{\mathbf{y}}} + \frac{L_{\mathbf{x}}}{m_{\mathbf{x}}} + \frac{L_{\mathbf{y}}}{m_{\mathbf{y}}}} \cdot \ln\left(\frac{\|\mathbf{z}_0 - \mathbf{z}^*\|}{\epsilon}\right) \cdot \left(\frac{L^2}{m_{\mathbf{x}} m_{\mathbf{y}}}\right)^{o(1)}.$$

In other words, for the quadratic saddle point problem, RHSS($k$) with the optimal choice of $k$ matches the lower bound up to a sub-polynomial factor.

## Broader Impact

This work is purely theoretical and does not present foreseeable societal consequences.

## Acknowledgments and Disclosure of Funding

The research is supported in part by the National Natural Science Foundation of China Grant 61822203, 61772297, 61632016, 61761146003, and the Zhongguancun Haihua Institute for Frontier Information Technology, Turing AI Institute of Nanjing and Xi'an Institute for Interdisciplinary Information Core Technology. The authors thank Kefan Dong, Guodong Zhang and Chi Jin for helpful discussions.

## Footnotes

[1] This lower bound is also proved by Ibrahim et al. [20]. Although their result is stated for a narrower class of algorithms, their proof actually works for the broader class of algorithms considered in [42].

[2]Note that this rescaling also does not change the lower bound.

[3]See Fact 4 in Supplementary Material for proof.

[4] See Fact 1 in Supplementary Material for proof.

[5]Although Sion's Theorem does not apply here as we considered unconstrained problem, we can still exchange the order since the function is strongly-convex-strongly-concave [17].

[6]Here $\rho(\cdot)$ stands for the spectral radius of a matrix, and $sp(\cdot)$ stands for its spectrum.

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
