[Supplementary Material]

# Supplementary Material for: Improved Algorithms for Convex-Concave Minimax Optimization

## 1 Some Useful Properties

In this section, we review some useful properties of functions in $\mathcal{F}(m_{\mathbf{x}}, m_{\mathbf{y}}, L_{\mathbf{x}}, L_{\mathbf{xy}}, L_{\mathbf{y}})$. Some of the facts are known (see e.g., [3], [8]) and we provide the proofs for completeness.

**Fact 1.** Suppose $f \in \mathcal{F}(m_{\mathbf{x}}, m_{\mathbf{y}}, L_{\mathbf{x}}, L_{\mathbf{xy}}, L_{\mathbf{y}})$. Let us define $\mathbf{y}^*(\mathbf{x}) := \arg\max_{\mathbf{y}} f(\mathbf{x}, \mathbf{y})$, $\mathbf{x}^*(\mathbf{y}) := \arg\min_{\mathbf{x}} f(\mathbf{x}, \mathbf{y})$, $\phi(\mathbf{x}) := \max_{\mathbf{y}} f(\mathbf{x}, \mathbf{y})$ and $\psi(\mathbf{y}) := \min_{\mathbf{x}} f(\mathbf{x}, \mathbf{y})$. Then, we have that

1. $\mathbf{y}^*$ is $L_{\mathbf{xy}}/m_{\mathbf{y}}$-Lipschitz, $\mathbf{x}^*$ is $L_{\mathbf{xy}}/m_{\mathbf{x}}$-Lipschitz;

2. $\phi(\mathbf{x})$ is $m_{\mathbf{x}}$-strongly convex and $L_{\mathbf{x}} + L_{\mathbf{xy}}^2/m_{\mathbf{y}}$-smooth; $\psi(\mathbf{y})$ is $m_{\mathbf{y}}$-strongly concave and $L_{\mathbf{y}} + L_{\mathbf{xy}}^2/m_{\mathbf{x}}$-smooth.

*Proof.* 1. Consider arbitrary $\mathbf{x}$ and $\mathbf{x}'$. By definition, $\nabla_{\mathbf{y}} f(\mathbf{x}, \mathbf{y}^*(\mathbf{x})) = \nabla_{\mathbf{y}} f(\mathbf{x}', \mathbf{y}^*(\mathbf{x}')) = \mathbf{0}$. By the definition of $(L_{\mathbf{x}}, L_{\mathbf{xy}}, L_{\mathbf{y}})$-smoothness, $\|\nabla_{\mathbf{y}} f(\mathbf{x}', \mathbf{y}^*(\mathbf{x}))\| \le L_{\mathbf{xy}} \|\mathbf{x} - \mathbf{x}'\|$. Thus

$$m_{\mathbf{y}} \|\mathbf{y}^*(\mathbf{x}) - \mathbf{y}^*(\mathbf{x}')\| \le \|\nabla_{\mathbf{y}} f(\mathbf{x}', \mathbf{y}^*(\mathbf{x}))\| \le L_{\mathbf{xy}} \|\mathbf{x} - \mathbf{x}'\|.$$

This proves that $y^*(\cdot)$ is $L_{\mathbf{xy}}/m_{\mathbf{y}}$-Lipschitz. Similarly $\mathbf{x}^*(\cdot)$ is $L_{\mathbf{xy}}/m_{\mathbf{x}}$-Lipschitz.

2. By Danskin's Theorem, $\nabla \phi(\mathbf{x}) = \nabla_{\mathbf{x}} f(\mathbf{x}, \mathbf{y}^*(\mathbf{x}))$. Thus, $\forall \mathbf{x}, \mathbf{x}'$

$$\begin{aligned}
\|\nabla \phi(\mathbf{x}) - \nabla \phi(\mathbf{x}')\| &= \|\nabla_{\mathbf{x}} f(\mathbf{x}, \mathbf{y}^*(\mathbf{x})) - \nabla_{\mathbf{x}} f(\mathbf{x}', \mathbf{y}^*(\mathbf{x}'))\| \\
&\le \|\nabla_{\mathbf{x}} f(\mathbf{x}, \mathbf{y}^*(\mathbf{x})) - \nabla_{\mathbf{x}} f(\mathbf{x}, \mathbf{y}^*(\mathbf{x}'))\| + \|\nabla_{\mathbf{x}} f(\mathbf{x}, \mathbf{y}^*(\mathbf{x}')) - \nabla_{\mathbf{x}} f(\mathbf{x}', \mathbf{y}^*(\mathbf{x}'))\| \\
&\le L_{\mathbf{xy}} \cdot \|\mathbf{y}^*(\mathbf{x}) - \mathbf{y}^*(\mathbf{x}')\| + L_{\mathbf{x}} \|\mathbf{x} - \mathbf{x}'\| \\
&\le \left( L_{\mathbf{x}} + \frac{L_{\mathbf{xy}}^2}{m_{\mathbf{y}}} \right) \|\mathbf{x} - \mathbf{x}'\|.
\end{aligned}$$

On the other hand, $\forall \mathbf{x}, \mathbf{x}'$,

$$\begin{aligned}
\phi(\mathbf{x}') - \phi(\mathbf{x}) - (\mathbf{x}' - \mathbf{x})^T \nabla \phi(\mathbf{x}) &= f(\mathbf{x}', \mathbf{y}^*(\mathbf{x}')) - f(\mathbf{x}, \mathbf{y}^*(\mathbf{x})) - (\mathbf{x}' - \mathbf{x})^T \nabla_{\mathbf{x}} f(\mathbf{x}, \mathbf{y}^*(\mathbf{x})) \\
&\ge f(\mathbf{x}', \mathbf{y}^*(\mathbf{x})) - f(\mathbf{x}, \mathbf{y}^*(\mathbf{x})) - (\mathbf{x}' - \mathbf{x})^T \nabla_{\mathbf{x}} f(\mathbf{x}, \mathbf{y}^*(\mathbf{x})) \\
&\ge \frac{m_{\mathbf{x}}}{2} \|\mathbf{x}' - \mathbf{x}\|^2.
\end{aligned}$$

Thus $\phi(\mathbf{x})$ is $m_{\mathbf{x}}$-strongly convex and $\left( L_{\mathbf{x}} + \frac{L_{\mathbf{xy}}^2}{m_{\mathbf{y}}} \right)$-smooth. By symmetric arguments, one can show that $\psi(\mathbf{y})$ is $m_{\mathbf{y}}$-strongly concave and $\left( L_{\mathbf{y}} + \frac{L_{\mathbf{xy}}^2}{m_{\mathbf{x}}} \right)$-smooth. $\square$

**Fact 2.** Let $\mathbf{z} := [\mathbf{x}; \mathbf{y}]$ and $\mathbf{z}^* := [\mathbf{x}^*; \mathbf{y}^*]$. Then

$$\frac{1}{\sqrt{2}} \left( \|\mathbf{x} - \mathbf{x}^*\| + \|\mathbf{y} - \mathbf{y}^*\| \right) \le \|\mathbf{z} - \mathbf{z}^*\| \le \|\mathbf{x} - \mathbf{x}^*\| + \|\mathbf{y} - \mathbf{y}^*\|.$$

*Proof.* This can be easily proven using the AM-GM inequality. $\square$

**Fact 3.** Let $\mathbf{z} := [\mathbf{x}; \mathbf{y}] \in \mathbb{R}^{m+n}$, $\mathbf{z}^* := [\mathbf{x}^*; \mathbf{y}^*]$. Then
$$\min\{m_{\mathbf{x}}, m_{\mathbf{y}}\}\|\mathbf{z} - \mathbf{z}^*\| \leq \|\nabla f(\mathbf{x}, \mathbf{y})\| \leq 2L\|\mathbf{z} - \mathbf{z}^*\|.$$

*Proof.* By properties of strong convexity [6], $\forall \mathbf{x}, \mathbf{y}$
$$f(\mathbf{x}, \mathbf{y}^*(\mathbf{x})) - f(\mathbf{x}, \mathbf{y}) \leq \frac{1}{2m_{\mathbf{y}}}\|\nabla_{\mathbf{y}} f(\mathbf{x}, \mathbf{y})\|^2.$$

Similarly,
$$f(\mathbf{x}, \mathbf{y}) - f(\mathbf{x}^*(\mathbf{y}), \mathbf{y}) \leq \frac{1}{2m_{\mathbf{x}}}\|\nabla_{\mathbf{x}} f(\mathbf{x}, \mathbf{y})\|^2.$$

Thus,
$$\|\nabla f(\mathbf{x}, \mathbf{y})\|^2 = \|\nabla_{\mathbf{x}} f(\mathbf{x}, \mathbf{y})\|^2 + \|\nabla_{\mathbf{y}} f(\mathbf{x}, \mathbf{y})\|^2$$
$$\geq 2\min\{m_{\mathbf{x}}, m_{\mathbf{y}}\}\left(\phi(\mathbf{x}) - \psi(\mathbf{y})\right).$$

Here $\phi(\cdot) = \max_{\mathbf{y}} f(\cdot, \mathbf{y})$, $\psi(\cdot) = \min_{\mathbf{x}} f(\mathbf{x}, \cdot)$. By Proposition 1, $\phi$ is $m_{\mathbf{x}}$-strongly convex while $\psi$ is $m_{\mathbf{y}}$-strongly concave. Hence
$$\phi(\mathbf{x}) - \psi(\mathbf{y}) \geq \frac{\min\{m_{\mathbf{x}}, m_{\mathbf{y}}\}}{2}\left(\|\mathbf{x} - \mathbf{x}^*\|^2 + \|\mathbf{y} - \mathbf{y}^*\|^2\right) = \frac{\min\{m_{\mathbf{x}}, m_{\mathbf{y}}\}}{2}\|\mathbf{z} - \mathbf{z}^*\|^2.$$

It follows that $\|\nabla f(\mathbf{x}, \mathbf{y})\| \geq \min\{m_{\mathbf{x}}, m_{\mathbf{y}}\}\|\mathbf{z} - \mathbf{z}^*\|$. On the other hand,
$$\|\nabla_{\mathbf{x}} f(\mathbf{x}, \mathbf{y})\| \leq L_{\mathbf{xy}}\|\mathbf{y} - \mathbf{y}^*\| + L_{\mathbf{x}}\|\mathbf{x} - \mathbf{x}^*\|,$$
$$\|\nabla_{\mathbf{y}} f(\mathbf{x}, \mathbf{y})\| \leq L_{\mathbf{xy}}\|\mathbf{x} - \mathbf{x}^*\| + L_{\mathbf{y}}\|\mathbf{y} - \mathbf{y}^*\|.$$

As a result $\|\nabla f(\mathbf{x}, \mathbf{y})\|^2 \leq L\left(\|\mathbf{x} - \mathbf{x}^*\| + \|\mathbf{y} - \mathbf{y}^*\|\right)^2 \leq 4L^2\|\mathbf{z} - \mathbf{z}^*\|^2.$

$\square$

**Fact 4.** Let $\hat{\mathbf{z}} = [\hat{\mathbf{x}}; \hat{\mathbf{y}}]$. Then $\|\hat{\mathbf{z}} - \mathbf{z}^*\| \leq \epsilon$ implies
$$\max_{\mathbf{y}} f(\hat{\mathbf{x}}, \mathbf{y}) - \min_{\mathbf{x}} f(\mathbf{x}, \hat{\mathbf{y}}) \leq \frac{L^2}{\min\{m_{\mathbf{x}}, m_{\mathbf{y}}\}}\epsilon^2.$$

*Proof.* Define $\phi(\mathbf{x}) = \max_{\mathbf{y}} f(\mathbf{x}, \mathbf{y})$ and $\psi(\mathbf{y}) = \min_{\mathbf{x}} f(\mathbf{x}, \mathbf{y})$. Then
$$\max_{\mathbf{y}} f(\hat{\mathbf{x}}, \mathbf{y}) - \min_{\mathbf{x}} f(\mathbf{x}, \hat{\mathbf{y}}) = \phi(\hat{\mathbf{x}}) - \psi(\hat{\mathbf{y}}).$$

By Fact 1, $\phi$ is $(L_{\mathbf{x}} + L_{\mathbf{xy}}^2/m_{\mathbf{x}})$-smooth while $\psi$ is $(L_{\mathbf{y}} + L_{\mathbf{xy}}^2/m_{\mathbf{x}})$-smooth. Since $\phi(\mathbf{x}^*) = \psi(\mathbf{y}^*)$, $\nabla\phi(\mathbf{x}^*) = \mathbf{0}$, $\nabla\psi(\mathbf{y}^*) = \mathbf{0}$,
$$\phi(\hat{\mathbf{x}}) - \psi(\hat{\mathbf{y}}) \leq \frac{1}{2}\left(L_{\mathbf{x}} + \frac{L_{\mathbf{xy}^2}}{m_{\mathbf{x}}}\right)\|\hat{\mathbf{x}} - \mathbf{x}^*\|^2 + \frac{1}{2}\left(L_{\mathbf{y}} + \frac{L_{\mathbf{xy}^2}}{m_{\mathbf{y}}}\right)\|\hat{\mathbf{y}} - \mathbf{y}^*\|^2$$
$$\leq \frac{1}{2}\left(L + \frac{L_{\mathbf{xy}^2}}{\min\{m_{\mathbf{x}}, m_{\mathbf{y}}\}}\right)\left(\|\hat{\mathbf{x}} - \mathbf{x}^*\|^2 + \|\hat{\mathbf{y}} - \mathbf{y}^*\|^2\right)$$
$$\leq \frac{L^2}{\min\{m_{\mathbf{x}}, m_{\mathbf{y}}\}}\epsilon^2.$$

$\square$

## 1.1 Accelerated Gradient Descent

Nesterov's Accelerated Gradient Descent [5] is an optimal first-order algorithm for smooth and convex functions. Here we present a version of AGD for minimizing an $l$-smooth and $m$-strongly convex functions $g(\cdot)$. It is a crucial building block for the algorithms in this work.

The following classical theorem holds for AGD. It implies that the complexity is $O\left(\sqrt{\kappa}\ln\left(\frac{1}{\epsilon}\right)\right)$, which greatly improves over the $O\left(\kappa\ln\left(\frac{1}{\epsilon}\right)\right)$ bound for gradient descent.

**Lemma 2.** *([6, Theorem 2.2.3]) In the AGD algorithm,*
$$\|\mathbf{x}_T - \mathbf{x}^*\|^2 \leq (\kappa + 1)\|\mathbf{x}_0 - \mathbf{x}^*\|^2 \cdot \left(1 - \frac{1}{\sqrt{\kappa}}\right)^T.$$

**Algorithm I** AGD($g$, $\mathbf{x}_0$, $T$) [6]

---

**Require:** Initial point $\mathbf{x}_0$, smoothness constant $l$, strongly-convex modulus $m$, number of iterations $T$
    $\tilde{\mathbf{x}}_0 \leftarrow \mathbf{x}_0, \eta \leftarrow 1/l, \kappa \leftarrow l/m, \theta \leftarrow (\sqrt{\kappa} - 1)/(\sqrt{\kappa} + 1)$
    **for** $t = 1, \cdots, T$ **do**
        $\mathbf{x}_t \leftarrow \tilde{\mathbf{x}}_{t-1} - \eta \nabla g(\tilde{\mathbf{x}}_{t-1})$
        $\tilde{\mathbf{x}}_t \leftarrow \mathbf{x}_t + \theta(\mathbf{x}_t - \mathbf{x}_{t-1})$
    **end for**

---

## 2 Proof of Theorem 1

We will start by giving a precise statement of Algorithm 1.

---

**Algorithm 1** Alternating Best Response (ABR)

---

**Require:** $g(\cdot, \cdot)$, Initial point $\mathbf{z}_0 = [\mathbf{x}_0; \mathbf{y}_0]$, precision $\epsilon$, parameters $m_\mathbf{x}, m_\mathbf{y}, L_\mathbf{x}, L_\mathbf{y}$
    $\kappa_\mathbf{x} := L_\mathbf{x}/m_\mathbf{x}, \kappa_\mathbf{y} := L_\mathbf{y}/m_\mathbf{y}, T \leftarrow \left\lceil \log_2 \left( \frac{4\sqrt{\kappa_\mathbf{x} + \kappa_\mathbf{y}}}{\epsilon} \right) \right\rceil$
    **for** $t = 0, \cdots, T$ **do**
        $\mathbf{x}_{t+1} \leftarrow \text{AGD}(g(\cdot, \mathbf{y}_t), \mathbf{x}_t, 2\sqrt{\kappa_\mathbf{x}} \ln(24\kappa_\mathbf{x}))$
        $\mathbf{y}_{t+1} \leftarrow \text{AGD}(-g(\mathbf{x}_{t+1}, \cdot), \mathbf{y}_t, 2\sqrt{\kappa_\mathbf{y}} \ln(24\kappa_\mathbf{y}))$
    **end for**

---

We proceed to prove Theorem 1.

**Theorem 1.** *If $g \in \mathcal{F}(m_\mathbf{x}, m_\mathbf{y}, L_\mathbf{x}, L_{\mathbf{xy}}, L_\mathbf{y})$ and $L_{\mathbf{xy}} < \frac{1}{2}\sqrt{m_\mathbf{x} m_\mathbf{y}}$, Alternating Best Response returns* $(\mathbf{x}_T, \mathbf{y}_T)$ *such that*

$$\|\mathbf{x}_T - \mathbf{x}^*\| + \|\mathbf{y}_T - \mathbf{y}^*\| \leq \epsilon \left( \|\mathbf{x}_0 - \mathbf{x}^*\| + \|\mathbf{y}_0 - \mathbf{y}^*\| \right),$$

*using ($\kappa_\mathbf{x} = L_\mathbf{x}/m_\mathbf{x}$, $\kappa_\mathbf{y} = L_\mathbf{y}/m_\mathbf{y}$)*

$$O\left( \left( \sqrt{\kappa_\mathbf{x} + \kappa_\mathbf{y}} \right) \cdot \ln\left( \kappa_\mathbf{x} \kappa_\mathbf{y} \right) \ln\left( \frac{\kappa_\mathbf{x} \kappa_\mathbf{y}}{\epsilon} \right) \right).$$

*gradient evaluations.*

*Proof.* Define $\tilde{\mathbf{x}}_{t+1} := \arg\min_\mathbf{x} f(\mathbf{x}, \mathbf{y}_t)$. Let us define $\mathbf{y}^*(\mathbf{x}) := \arg\max_\mathbf{y} f(\mathbf{x}, \mathbf{y})$, $\mathbf{x}^*(\mathbf{y}) := \arg\min_\mathbf{x} f(\mathbf{x}, \mathbf{y})$ and $\phi(\mathbf{x}) := \max_\mathbf{y} f(\mathbf{x}, \mathbf{y})$. Also define $\tilde{\mathbf{x}}_{t+1} := \arg\min_\mathbf{x} f(\mathbf{x}, \mathbf{y}^*(\mathbf{x}_t))$ and $\hat{\mathbf{x}}_{t+1} := \arg\min_\mathbf{x} f(\mathbf{x}, \mathbf{y}_t)$.

The basic idea is the following. Because $\mathbf{y}^*(\cdot)$ is $L_{\mathbf{xy}}/m_\mathbf{y}$-Lipschitz and $\mathbf{x}^*(\cdot)$ is $L_{\mathbf{xy}}/m_\mathbf{x}$-Lipschitz (Fact 1),

$$\|\mathbf{x}^*(\mathbf{y}_t) - \mathbf{x}^*\| = \|\mathbf{x}^*(\mathbf{y}_t) - \mathbf{x}^*(\mathbf{y}^*)\| \leq \frac{L_{\mathbf{xy}}}{m_\mathbf{x}} \|\mathbf{y}_t - \mathbf{y}^*\|,$$

$$\|\mathbf{y}^*(\mathbf{x}_{t+1}) - \mathbf{y}^*\| = \|\mathbf{y}^*(\mathbf{x}_{t+1}) - \mathbf{y}^*(\mathbf{x}^*)\| \leq \frac{L_{\mathbf{xy}}}{m_\mathbf{y}} \|\mathbf{x}_{t+1} - \mathbf{x}^*\|.$$

By a standard analysis of accelerated gradient descent (Lemma 2), since $\hat{\mathbf{x}}_{t+1} = \mathbf{x}^*(\mathbf{y}_t)$ is the minimum of $f(\cdot, \mathbf{y}_t)$ and $\mathbf{x}_t$ is the initial point,

$$\|\mathbf{x}_{t+1} - \hat{\mathbf{x}}_{t+1}\|^2 \leq (\kappa_\mathbf{x} + 1)\|\mathbf{x}_t - \hat{\mathbf{x}}_{t+1}\|^2 \cdot \left( 1 - \frac{1}{\sqrt{\kappa_\mathbf{x}}} \right)^{2\sqrt{\kappa_\mathbf{x}} \ln(24\kappa_\mathbf{x})}$$

$$\leq \|\mathbf{x}_t - \hat{\mathbf{x}}_{t+1}\|^2 \cdot (\kappa_\mathbf{x} + 1) \cdot \exp\left\{ -2\ln(24\kappa_\mathbf{x}) \right\}$$

$$\leq \frac{1}{256} \|\mathbf{x}_t - \hat{\mathbf{x}}_{t+1}\|^2.$$

That is,

$$\|\mathbf{x}_{t+1} - \mathbf{x}^*(\mathbf{y}_t)\| \leq \frac{1}{16} \|\mathbf{x}_t - \mathbf{x}^*(\mathbf{y}_t)\| \leq \frac{1}{16} \left( \|\mathbf{x}_t - \mathbf{x}^*\| + \|\mathbf{x}^*(\mathbf{y}_t) - \mathbf{x}^*\| \right).$$

Thus

$$\|\mathbf{x}_{t+1} - \mathbf{x}^*\| \le \|\mathbf{x}_{t+1} - \mathbf{x}^*(\mathbf{y}_t)\| + \|\mathbf{x}^*(\mathbf{y}_t) - \mathbf{x}^*\| \le \frac{17}{16} \cdot \frac{L_{\mathbf{xy}}}{m_{\mathbf{x}}} \|\mathbf{y}_t - \mathbf{y}^*\| + \frac{1}{16} \|\mathbf{x}_t - \mathbf{x}^*\|. \qquad (1)$$

Similarly,

$$\|\mathbf{y}_{t+1} - \mathbf{y}^*(\mathbf{x}_{t+1})\| \le \frac{1}{16} \|\mathbf{y}_t - \mathbf{y}^*(\mathbf{x}_{t+1})\| \le \frac{1}{16} \left( \|\mathbf{y}_t - \mathbf{y}^*\| + \|\mathbf{y}^*(\mathbf{x}_{t+1}) - \mathbf{y}^*\| \right).$$

Thus

$$\begin{aligned}
\|\mathbf{y}_{t+1} - \mathbf{y}^*\| &\le \|\mathbf{y}_{t+1} - \mathbf{y}^*(\mathbf{x}_{t+1})\| + \|\mathbf{y}^*(\mathbf{x}_{t+1}) - \mathbf{y}^*\| \\
&\le \frac{17}{16} \cdot \frac{L_{\mathbf{xy}}}{m_{\mathbf{y}}} \|\mathbf{x}_{t+1} - \mathbf{x}^*\| + \frac{1}{16} \|\mathbf{y}_t - \mathbf{y}^*\| \\
&\le \left( \frac{17^2}{16^2} \cdot \frac{L_{\mathbf{xy}}^2}{m_{\mathbf{x}} m_{\mathbf{y}}} + \frac{1}{16} \right) \|\mathbf{y}_t - \mathbf{y}^*\| + \frac{17 L_{\mathbf{xy}}}{256 m_{\mathbf{y}}} \|\mathbf{x}_t - \mathbf{x}^*\| \\
&\le 0.35 \|\mathbf{y}_t - \mathbf{y}^*\| + \frac{17 L_{\mathbf{xy}}}{256 m_{\mathbf{y}}} \|\mathbf{x}_t - \mathbf{x}^*\|. \qquad (2)
\end{aligned}$$

Define $C := 4\sqrt{m_{\mathbf{y}}/m_{\mathbf{x}}}$. By adding (1) and $C$ times (2), one gets

$$\begin{aligned}
\|\mathbf{x}_{t+1} - \mathbf{x}^*\| + C\|\mathbf{y}_{t+1} - \mathbf{y}^*\| &\le \left( \frac{1}{16} + \frac{17 L_{\mathbf{xy}}}{64\sqrt{m_{\mathbf{x}} m_{\mathbf{y}}}} \right) \|\mathbf{x}_t - \mathbf{x}^*\| + \left( 0.35 C + \frac{17}{16} \cdot \frac{L_{\mathbf{xy}}}{m_{\mathbf{x}}} \right) \|\mathbf{y}_t - \mathbf{y}^*\| \\
&\le \frac{1}{2} \|\mathbf{x}_t - \mathbf{x}^*\| + \left( 0.35 + \frac{17 L_{\mathbf{xy}}}{64\sqrt{m_{\mathbf{x}} m_{\mathbf{y}}}} \right) C \|\mathbf{y}_t - \mathbf{y}^*\| \\
&\le \frac{1}{2} \left( \|\mathbf{x}_t - \mathbf{x}^*\| + C\|\mathbf{y}_t - \mathbf{y}^*\| \right).
\end{aligned}$$

It follows that

$$\|\mathbf{x}_T - \mathbf{x}^*\| + C\|\mathbf{y}_T - \mathbf{y}^*\| \le 2^{-T} \left( \|\mathbf{x}_0 - \mathbf{x}^*\| + C\|\mathbf{y}_0 - \mathbf{y}^*\| \right).$$

If $C \ge 1$, then

$$\|\mathbf{x}_T - \mathbf{x}^*\| + \|\mathbf{y}_T - \mathbf{y}^*\| \le 4\sqrt{\frac{m_{\mathbf{y}}}{m_{\mathbf{x}}}} \cdot 2^{-T} \cdot \left( \|\mathbf{x}_0 - \mathbf{x}^*\| + \|\mathbf{y}_0 - \mathbf{y}^*\| \right).$$

On the other hand, if $C < 1$, then

$$\|\mathbf{x}_T - \mathbf{x}^*\| + \|\mathbf{y}_T - \mathbf{y}^*\| \le \frac{2^{-T}}{C} \left( \|\mathbf{x}_0 - \mathbf{x}^*\| + \|\mathbf{y}_0 - \mathbf{y}^*\| \right) = \sqrt{\frac{m_{\mathbf{x}}}{m_{\mathbf{y}}}} \cdot 2^{-T-1} \left( \|\mathbf{x}_0 - \mathbf{x}^*\| + \|\mathbf{y}_0 - \mathbf{y}^*\| \right).$$

Since $\max\{m_{\mathbf{x}}/m_{\mathbf{y}}, m_{\mathbf{y}}/m_{\mathbf{x}}\} \le L_{\mathbf{x}}/\min\{m_{\mathbf{x}}, m_{\mathbf{y}}\}$,

$$\|\mathbf{x}_T - \mathbf{x}^*\| + \|\mathbf{y}_T - \mathbf{y}^*\| \le 4\sqrt{\frac{L_{\mathbf{x}}}{\min\{m_{\mathbf{x}}, m_{\mathbf{y}}\}}} \cdot 2^{-T} \left( \|\mathbf{x}_0 - \mathbf{x}^*\| + \|\mathbf{y}_0 - \mathbf{y}^*\| \right). \qquad (3)$$

The theorem follows from this inequality. $\qquad\square$

## 3  Proof of Theorem 2

**Theorem 2.** *Assume that* $M \ge 20\kappa\sqrt{2\kappa + \frac{L}{m_{\mathbf{x}}} + \frac{L_{\mathbf{xy}}^2}{m_{\mathbf{x}} m_{\mathbf{y}}}} \left( 1 + \frac{L}{m_{\mathbf{y}}} \right)$. *The number of iterations needed by Algorithm 2 to produce* $(\mathbf{x}_T, \mathbf{y}_T)$ *such that*

$$\|\mathbf{x}_T - \mathbf{x}^*\| + \|\mathbf{y}_T - \mathbf{y}^*\| \le \epsilon \left( \|\mathbf{x}_0 - \mathbf{x}^*\| + \|\mathbf{y}_0 - \mathbf{y}^*\| \right)$$

*is at most* $(\kappa = \beta/m_{\mathbf{x}})$

$$\hat{T} = 8\sqrt{\kappa} \cdot \ln\left( \frac{28\kappa^2 L}{m_{\mathbf{y}}} \sqrt{\frac{L^2}{m_{\mathbf{x}} m_{\mathbf{y}}}} \cdot \frac{1}{\epsilon} \right). \qquad (4)$$

---

**Algorithm II** Inexact Accelerated Proximal Point Algorithm (Inexact APPA)

---

**Require:** Initial point $\mathbf{x}_0$, proximal parameter $\beta$, strongly convex module $m$

    $\hat{\mathbf{x}}_0 \leftarrow \mathbf{x}_0, \kappa \leftarrow \beta/m, \theta \leftarrow \frac{2\sqrt{\kappa}-1}{2\sqrt{\kappa}+1}, \tau \leftarrow \frac{1}{2\sqrt{\kappa}+4\kappa}$

    **for** $t = 1, \cdots, T$ **do**

        Find $\mathbf{x}_t$ such that $g(\mathbf{x}_t) + \beta\|\mathbf{x}_t - \hat{\mathbf{x}}_{t-1}\|^2 \leq \min_{\mathbf{x}}\{g(\mathbf{x}) + \beta\|\mathbf{x} - \hat{\mathbf{x}}_{t-1}\|^2\} + \delta_t$

        $\hat{\mathbf{x}}_t \leftarrow \mathbf{x}_t + \theta(\mathbf{x}_t - \mathbf{x}_{t-1}) + \tau(\mathbf{x}_t - \hat{\mathbf{x}}_{t-1})$

    **end for**

---

Before proving the theorem, we would first state the inexact accelerated proximal point algorithm [3], which is the basis of Algorithm 2.

The following two lemmas about the inexact APPA algorithm follow from the proof of Theorem 4.1 [3] in an earlier version of the paper. Here we provide their proofs for completeness.

**Lemma 3.** *Suppose that $\{(\mathbf{x}_t, \hat{\mathbf{x}}_t)\}_{t \geq 0}$ are generated by running the inexact APPA algorithm on $g(\cdot)$. Then $\forall t \geq 1, \forall \mathbf{x}$,*

$$g(\mathbf{x}) \geq g(\mathbf{x}_t) - 2\beta(\mathbf{x} - \mathbf{x}_t)^T(\mathbf{x}_t - \hat{\mathbf{x}}_{t-1}) + \frac{m}{4}\|\mathbf{x} - \mathbf{x}_t\|^2 - 7\kappa\delta_t.$$

*Proof of Lemma 3.* By definition

$$g(\mathbf{x}_t) + \beta\|\mathbf{x}_t - \hat{\mathbf{x}}_{t-1}\| \leq \min_{\mathbf{x}}\{g(\mathbf{x}) + \beta\|\mathbf{x} - \hat{\mathbf{x}}_{t-1}\|^2\} + \delta_t.$$

Define $\mathbf{x}_t^* := \arg\min_{\mathbf{x}}\{g(\mathbf{x}) + \beta\|\mathbf{x} - \hat{\mathbf{x}}_{t-1}\|^2\}$. By the $m$-strong convexity of $g(\cdot)$, we have $\forall \mathbf{x}$,

$$g(\mathbf{x}) + \beta\|\mathbf{x} - \hat{\mathbf{x}}_{t-1}\|^2 \geq g(\mathbf{x}_t^*) + \beta\|\mathbf{x}_t^* - \hat{\mathbf{x}}_{t-1}\|^2 + \left(\frac{m}{2} + \beta\right)\|\mathbf{x} - \mathbf{x}_t^*\|^2.$$

Equivalently,

$$g(\mathbf{x}) \geq g(\mathbf{x}_t) + \beta\|\mathbf{x}_t - \hat{\mathbf{x}}_{t-1}\|^2 - \beta\|\mathbf{x} - \hat{\mathbf{x}}_{t-1}\|^2 + \left(\beta + \frac{m}{2}\right)\|\mathbf{x} - \mathbf{x}_t^*\|^2 - \delta_t$$

$$= g(\mathbf{x}_t) - 2\beta(\mathbf{x} - \mathbf{x}_t)^T(\mathbf{x}_t - \hat{\mathbf{x}}_{t-1}) - \beta\|\mathbf{x} - \mathbf{x}_t\|^2 + \left(\beta + \frac{m}{2}\right)\|\mathbf{x} - \mathbf{x}_t^*\|^2 - \delta_t.$$

On the other hand, we have

$$\left(\beta + \frac{m}{2}\right)\|\mathbf{x} - \mathbf{x}_t^*\|^2 - \beta\|\mathbf{x} - \mathbf{x}_t\|^2 = \frac{m\|\mathbf{x} - \mathbf{x}_t\|^2}{2} + (2\beta + m)(\mathbf{x} - \mathbf{x}_t)^T(\mathbf{x}_t - \mathbf{x}_t^*) + (\beta + \frac{m}{2})\|\mathbf{x}_t - \mathbf{x}_t^*\|^2.$$

By Cauchy-Schwarz Inequality,

$$(\mathbf{x} - \mathbf{x}_t)^T(\mathbf{x}_t - \mathbf{x}_t^*) \geq -\frac{m\|\mathbf{x} - \mathbf{x}_t\|^2}{4(2\beta + m)} - (1 + 2\kappa)\|\mathbf{x}_t - \mathbf{x}_t^*\|^2.$$

Putting the pieces together yields

$$g(\mathbf{x}) \geq g(\mathbf{x}_t) - 2\beta(\mathbf{x} - \mathbf{x}_t)^T(\mathbf{x}_t - \hat{\mathbf{x}}_{t-1}) + \frac{m\|\mathbf{x} - \mathbf{x}_t\|^2}{2} + (\beta + \frac{m}{2})\|\mathbf{x}_t - \mathbf{x}_t^*\|^2$$

$$- \frac{m\|\mathbf{x} - \mathbf{x}_t\|^2}{4} - (1 + 2\kappa)(2\beta + m)\|\mathbf{x}_t - \mathbf{x}_t^*\|^2 - \delta_t$$

$$= g(\mathbf{x}_t) - 2\beta(\mathbf{x} - \mathbf{x}_t)^T(\mathbf{x}_t - \hat{\mathbf{x}}_{t-1}) + \frac{m\|\mathbf{x} - \mathbf{x}_t\|^2}{4} - (2\beta + m)(\frac{1}{2} + 2\kappa)\|\mathbf{x}_t - \mathbf{x}_t^*\|^2 - \delta_t.$$

Also, since $g(\mathbf{x}) + \beta\|\mathbf{x} - \hat{\mathbf{x}}_{t-1}\|^2$ is $(2\beta + m)$-strongly convex,

$$\|\mathbf{x}_t - \mathbf{x}_t^*\|^2 \leq \frac{2}{2\beta + m}\left(g(\mathbf{x}_t) + \beta\|\mathbf{x}_t - \hat{\mathbf{x}}_{t-1}\|^2 - \min_{\mathbf{x}}\{g(\mathbf{x}) + \beta\|\mathbf{x} - \hat{\mathbf{x}}_{t-1}\|^2\}\right) \leq \frac{2\delta_t}{2\beta + m}.$$

Thus

$$g(\mathbf{x}) \geq g(\mathbf{x}_t) - 2\beta(\mathbf{x} - \mathbf{x}_t)^T(\mathbf{x}_t - \hat{\mathbf{x}}_{t-1}) + \frac{m\|\mathbf{x} - \mathbf{x}_t\|^2}{4} - 7\kappa\delta_t.$$

$\square$

**Lemma 4.** *Suppose that $\{\mathbf{x}_t\}_{t\geq 0}$ is generated by running the inexact APPA algorithm on $g(\cdot)$. There exists a sequence $\{\Lambda_t\}_{t\geq 0}$ such that*

1. $\Lambda_t \geq g(\mathbf{x}_t)$

2. $\Lambda_0 - g(\mathbf{x}^*) \leq 2(g(\mathbf{x}_0) - g(\mathbf{x}^*))$

3. $\Lambda_{t+1} - g(\mathbf{x}^*) \leq \left(1 - \frac{1}{2\sqrt{\kappa}}\right)(\Lambda_t - g(\mathbf{x}^*)) + 11\kappa\delta_{t+1}$

*Proof of Lemma 4.* Let us slightly abuse notation, and define a sequence of functions $\{\Lambda(\mathbf{x})\}_{t\geq 0}$ first:

$$\Lambda_0(\mathbf{x}) := g(\mathbf{x}_0) + \frac{m\|\mathbf{x} - \mathbf{x}_0\|^2}{4},$$

$$\Lambda_{t+1}(\mathbf{x}) := \frac{1}{2\sqrt{\kappa}}\left(g(\mathbf{x}_{t+1}) + 2\beta(\hat{\mathbf{x}}_t - \mathbf{x}_{t+1})^T(\mathbf{x} - \mathbf{x}_{t+1}) + \frac{m\|\mathbf{x} - \mathbf{x}_{t+1}\|^2}{4} + 14\kappa^{3/2}\delta_{t+1}\right) + \left(1 - \frac{1}{2\sqrt{\kappa}}\right)\Lambda_t(\mathbf{x}).$$

The sequence $\{\Lambda_t\}_{t\geq 0}$ in the lemma is then defined as $\Lambda_t := \Lambda_t(\mathbf{x}^*)$. Note that later we do not need to make use of the explicit definition of $\Lambda_t$.

From the definition, Property 2 is straightforward, as

$$\Lambda_0 - g(\mathbf{x}^*) = \frac{m\|\mathbf{x}^* - \mathbf{x}_0\|^2}{4} + g(\mathbf{x}_0) - g(\mathbf{x}^*) \leq \frac{1}{2}(g(\mathbf{x}_0) - g(\mathbf{x}^*)) + g(\mathbf{x}_0) - g(\mathbf{x}^*).$$

Now, let us show $\Lambda_t \geq \min_{\mathbf{x}} \Lambda_t(\mathbf{x}) \geq g(\mathbf{x}_t)$ using induction. Let $\mathbf{w}_t := \arg\min_{\mathbf{x}} \Lambda_t(\mathbf{x})$ and $\Lambda_t^* := \min_{\mathbf{x}} \Lambda_t(\mathbf{x})$. Observe that $\Lambda_t(\mathbf{x})$ is always a quadratic function of the form $\Lambda_t(\mathbf{x}) = \Lambda_t^* + \frac{m}{4}\|\mathbf{x} - \mathbf{w}_t\|^2$. Then the following recursions hold for $\mathbf{w}_t$ and $\Lambda_t^*$:

$$\mathbf{w}_{t+1} = (1 - \frac{1}{2\sqrt{\kappa}})\mathbf{w}_t + 2\sqrt{\kappa}(\mathbf{x}_{t+1} - \hat{\mathbf{x}}_t) + \frac{\mathbf{x}_{t+1}}{2\sqrt{\kappa}},$$

$$\Lambda_{t+1}^* = \left(1 - \frac{1}{2\sqrt{\kappa}}\right)\Lambda_t^* + \frac{1}{2\sqrt{\kappa}}\left(g(\mathbf{x}_{t+1}) + 14\kappa^{3/2}\delta_{t+1}\right)$$

$$+ \frac{1}{2\sqrt{\kappa}}\left(1 - \frac{1}{2\sqrt{\kappa}}\right)\left(\frac{m\|\mathbf{x}_{t+1} - \mathbf{w}_t\|^2}{4} + 2\beta(\hat{\mathbf{x}}_t - \mathbf{x}_{t+1})^T(\mathbf{w}_t - \mathbf{x}_{t+1})\right).$$

The recursion for $\mathbf{w}_{t+1}$ can be derived by differentiating both sides in the recusion of $\Lambda_t(\mathbf{x})$, while the recursion for $\Lambda_{t+1}^*$ can be derived by plugging the recursion for $\mathbf{w}_{t+1}$ into $\Lambda_{t+1}^* = \Lambda_{t+1}(\mathbf{w}_{t+1})$.

Now, assume that $\Lambda_t^* \geq g(\mathbf{x}_t)$ for $t \leq T - 1$. Then

$$\Lambda_T^* \geq \left(1 - \frac{1}{2\sqrt{\kappa}}\right)g(\mathbf{x}_{T-1}) + \frac{1}{2\sqrt{\kappa}}\left(g(\mathbf{x}_T) + 14\kappa^{3/2}\delta_T\right)$$

$$+ \frac{1}{2\sqrt{\kappa}}\left(1 - \frac{1}{2\sqrt{\kappa}}\right)\left(\frac{m\|\mathbf{x}_T - \mathbf{w}_{T-1}\|^2}{4} + 2\beta(\hat{\mathbf{x}}_{T-1} - \mathbf{x}_T)^T(\mathbf{w}_{T-1} - \mathbf{x}_T)\right). \quad (5)$$

Applying Lemma 3 with $\mathbf{x} = \mathbf{x}_{T-1}$ yields

$$g(\mathbf{x}_{T-1}) \geq g(\mathbf{x}_T) + 2\beta(\mathbf{x}_{T-1} - \mathbf{x}_T)^T(\hat{\mathbf{x}}_{T-1} - \mathbf{x}_T) + \frac{m\|\mathbf{x}_{T-1} - \mathbf{x}_T\|^2}{4} - 7\kappa\delta_T. \quad (6)$$

Summing (5) and (6) gives

$$\Lambda_T^* \geq g(\mathbf{x}_T) + 2\beta(1 - \frac{1}{2\sqrt{\kappa}})(\hat{\mathbf{x}}_{T-1} - \mathbf{x}_T)^T\left[(\mathbf{x}_{T-1} - \mathbf{x}_T) + \frac{\mathbf{w}_{T-1} - \mathbf{x}_T}{2\sqrt{\kappa}}\right]$$

$$\geq g(\mathbf{x}_T) + 2\beta(1 - \frac{1}{2\sqrt{\kappa}})(\hat{\mathbf{x}}_{T-1} - \mathbf{x}_T)^T\left[(\mathbf{x}_{T-1} - \hat{\mathbf{x}}_{T-1}) + \frac{\mathbf{w}_{T-1} - \hat{\mathbf{x}}_{T-1}}{2\sqrt{\kappa}}\right].$$

The second inequality follows from

$$(\hat{\mathbf{x}}_{T-1} - \mathbf{x}_T)^T\left(\mathbf{x}_T - \hat{\mathbf{x}}_{T-1} + \frac{\mathbf{x}_T - \hat{\mathbf{x}}_{T-1}}{2\sqrt{\kappa}}\right) \leq 0.$$

By the update formula

$$\hat{\mathbf{x}}_{t+1} = \mathbf{x}_{t+1} + \frac{2\sqrt{\kappa}-1}{2\sqrt{\kappa}+1}(\mathbf{x}_{t+1}-\mathbf{x}_t) + \frac{1}{2\sqrt{\kappa}+4\kappa}(\mathbf{x}_{t+1}-\hat{\mathbf{x}}_t)$$

and the recursive rule for $\mathbf{w}_t$, we get

$$
\begin{aligned}
&(\mathbf{x}_{t+1}-\hat{\mathbf{x}}_{t+1}) + \frac{1}{2\sqrt{\kappa}}(\mathbf{w}_{t+1}-\hat{\mathbf{x}}_{t+1}) \\
=& \mathbf{x}_{t+1} + \frac{1}{2\sqrt{\kappa}}\left[\left(1-\frac{1}{2\sqrt{\kappa}}\mathbf{w}_t + 2\sqrt{\kappa}(\mathbf{x}_{t+1}-\hat{\mathbf{x}}_t) + \frac{\mathbf{x}_{t+1}}{2\sqrt{\kappa}}\right)\right] \\
&- \left(1+\frac{2}{\sqrt{\kappa}}\right)\left(\mathbf{x}_{t+1}+\frac{2\sqrt{\kappa}-1}{2\sqrt{\kappa}+1}(\mathbf{x}_{t+1}-\mathbf{x}_t)+\frac{1}{2\sqrt{\kappa}+4\kappa}(\mathbf{x}_{t+1}-\hat{\mathbf{x}}_t)\right) \\
=& \left(1-\frac{2}{\sqrt{\kappa}}\right)\left[\mathbf{x}_t + \frac{1}{2\sqrt{\kappa}}\mathbf{w}_t - \left(1+\frac{1}{2\sqrt{\kappa}}\hat{\mathbf{x}}_t\right)\right].
\end{aligned}
$$

Meanwhile, when $t=0$, $\mathbf{x}_t = \hat{\mathbf{x}}_t = \mathbf{w}_t = \mathbf{x}_0$. Thus, by induction, we have for any $t$, $(\mathbf{x}_t - \hat{\mathbf{x}}_t) + \frac{1}{2\sqrt{\kappa}}(\mathbf{w}_t - \hat{\mathbf{x}}_t) = 0$. As a result $\Lambda_T^* \geq g(\mathbf{x}_T)$. Again, by induction, this holds for all $T$. This proves Property 1 in the lemma.

Let us now focus on the final property. Combining Lemma 3 and the recursion for $\Lambda_t(\mathbf{x})$,

$$\Lambda_{t+1} = \Lambda_{t+1}(\mathbf{x}^*) \leq \left(1-\frac{1}{2\sqrt{\kappa}}\right)\Lambda_t + \frac{1}{2\sqrt{\kappa}}\left(g(\mathbf{x}^*) + 14\kappa^{3/2}\delta_{t+1} + 7\kappa\delta_{t+1}\right).$$

It follows that

$$\Lambda_{t+1} - g(\mathbf{x}^*) \leq \left(1-\frac{1}{2\sqrt{\kappa}}\right)(\Lambda_t - g(\mathbf{x}^*)) + 11\kappa\delta_{t+1}.$$

This is exactly Property 3. $\qquad\square$

Now we are ready to prove Theorem 2.

*Proof.* Define $\phi(\mathbf{x}) := \max_{\mathbf{y}} f(\mathbf{x},\mathbf{y})$ and $\hat{L} := L + L_{\mathbf{xy}}^2/m_{\mathbf{y}}$. Then $\phi(\mathbf{x})$ is $m_{\mathbf{x}}$-strongly convex and $\hat{L}$-smooth. Observe that

$$
\begin{aligned}
\mathbf{x}_t^* &= \arg\min_{\mathbf{x}}\left[\phi(\mathbf{x}) + \beta\|\mathbf{x}-\hat{\mathbf{x}}_{t-1}\|^2\right], \\
\mathbf{y}_t^* &= \arg\max_{\mathbf{y}}\left[f(\mathbf{x}_t^*,\mathbf{y})\right].
\end{aligned}
$$

Thus Algorithm 2 is an instance of the inexact APPA algorithm on $\phi(\mathbf{x})$ with proximal parameter $\beta$ and strongly convex module $m_{\mathbf{x}}$, and with

$$
\begin{aligned}
\delta_t &= \phi(\mathbf{x}_t) + \beta\|\mathbf{x}_t-\hat{\mathbf{x}}_{t-1}\|^2 - \min_{\mathbf{x}}\left\{\phi(\mathbf{x}) + \beta\|\mathbf{x}-\hat{\mathbf{x}}_{t-1}\|^2\right\} \\
&\leq \frac{\hat{L}+2\beta}{2}\|\mathbf{x}_t-\mathbf{x}_t^*\|^2.
\end{aligned}
\tag{7}
$$

Here we used the fact that, for a $L$-smooth function $g(\cdot)$ whose minimum is $\mathbf{x}^*$, $g(\mathbf{x}) - g(\mathbf{x}^*) \leq \frac{L}{2}\|\mathbf{x}-\mathbf{x}^*\|^2$. Define $C_1 := \|\mathbf{x}_0 - \mathbf{x}^*\| + \|\mathbf{y}_0 - \mathbf{y}^*\|$ and $C_0 := 44\kappa\sqrt{\kappa}\frac{\hat{L}+2\beta}{2}C_1^2$. Let us state the following induction hypothesis

$$\Delta_t := \Lambda_t - \phi(\mathbf{x}^*) \leq C_0\left(1-\frac{1}{4\sqrt{\kappa}}\right)^t,\tag{8}$$

$$\epsilon_t := \|\mathbf{x}_t - \mathbf{x}_t^*\| + \|\mathbf{y}_t - \mathbf{y}_t^*\| \leq C_1\left(1-\frac{1}{4\sqrt{\kappa}}\right)^{\frac{t}{2}}.\tag{9}$$

It is easy to verify that with our choice of $C_0$ and $C_1$, both (8) and (9) hold for $t = 0$.

Now, assume that (8) and (9) hold for $\tau = 1, 2, \cdots, t$. Define $\mathbf{y}^*(\cdot) := \arg\max_{\mathbf{y}} f(\cdot, \mathbf{y})$. By Fact 1, $\mathbf{y}^*(\cdot)$ is $(L/m_{\mathbf{y}})$-Lipschitz. Thus

$$
\begin{aligned}
\|\mathbf{y}_t - \mathbf{y}_{t+1}^*\| &\leq \|\mathbf{y}_t^* - \mathbf{y}_{t+1}^*\| + \|\mathbf{y}_t - \mathbf{y}_t^*\| \\
&\leq \|\mathbf{y}^*(\mathbf{x}_t^*) - \mathbf{y}^*(\mathbf{x}_{t+1}^*)\| + \epsilon_t \\
&\leq \frac{L}{m_{\mathbf{y}}} \cdot \left( \|\mathbf{x}_t^* - \mathbf{x}_t\| + \|\mathbf{x}_t - \mathbf{x}_{t+1}^*\| \right) + \epsilon_t \\
&\leq \left( \frac{L}{m_{\mathbf{y}}} + 1 \right) \epsilon_t + \frac{L}{m_{\mathbf{y}}} \|\mathbf{x}_t - \mathbf{x}_{t+1}^*\|.
\end{aligned}
$$

It follows that

$$
\epsilon_{t+1} \leq \frac{1}{M} \left[ \|\mathbf{x}_t - \mathbf{x}_{t+1}^*\| + \|\mathbf{y}_t - \mathbf{y}_{t+1}^*\| \right] \leq \frac{1 + \frac{L}{m_{\mathbf{y}}}}{M} \cdot \left( \|\mathbf{x}_t - \mathbf{x}_{t+1}^*\| + \epsilon_t \right). \tag{10}
$$

Note that by Lemma 4 and the induction hypothesis (8)

$$
\phi(\mathbf{x}_{t+1}^*) - \phi(\mathbf{x}^*) \leq \left( 1 - \frac{1}{2\sqrt{\kappa}} \right) \Delta_t \leq C_0 \left( 1 - \frac{1}{4\sqrt{\kappa}} \right)^t.
$$

By the $m_{\mathbf{x}}$-strong convexity of $\phi(\cdot)$ (Fact 1),

$$
\|\mathbf{x}_{t+1}^* - \mathbf{x}^*\| \leq \sqrt{\frac{2}{m_{\mathbf{x}}} \left( \phi(\mathbf{x}_{t+1}^*) - \phi(\mathbf{x}^*) \right)} \leq \sqrt{\frac{2C_0}{m_{\mathbf{x}}}} \left( 1 - \frac{1}{4\sqrt{\kappa}} \right)^{\frac{t}{2}}.
$$

Meanwhile

$$
\|\mathbf{x}_t - \mathbf{x}^*\| \leq \sqrt{\frac{2}{m_{\mathbf{x}}} \left( \phi(\mathbf{x}_t) - \phi(\mathbf{x}^*) \right)} \leq \sqrt{\frac{2C_0}{m_{\mathbf{x}}}} \left( 1 - \frac{1}{4\sqrt{\kappa}} \right)^{\frac{t}{2}}.
$$

Therefore

$$
\|\mathbf{x}_t - \mathbf{x}_{t+1}^*\| \leq \|\mathbf{x}_t - \mathbf{x}^*\| + \|\mathbf{x}_{t+1}^* - \mathbf{x}^*\| \leq 2\sqrt{\frac{2C_0}{m_{\mathbf{x}}}} \left( 1 - \frac{1}{4\sqrt{\kappa}} \right)^{\frac{t}{2}}. \tag{11}
$$

By (10), (9) and the fact that $M \geq 20\kappa \sqrt{2\kappa + \frac{\hat{L}}{m_{\mathbf{x}}}} (1 + L/m_{\mathbf{y}})$

$$
\begin{aligned}
\epsilon_{t+1} &\leq \frac{1 + \frac{L}{m_{\mathbf{y}}}}{M} \left( 2\sqrt{\frac{2C_0}{m_{\mathbf{x}}}} + C_1 \right) \left( 1 - \frac{1}{4\sqrt{\kappa}} \right)^{\frac{t}{2}} \\
&\leq \frac{1 + \frac{L}{m_{\mathbf{y}}}}{M} \cdot \left( 1 + 2\sqrt{\frac{44\kappa^{1.5}(\hat{L} + 2\beta)}{m_{\mathbf{x}}}} \right) C_1 \left( 1 - \frac{1}{4\sqrt{\kappa}} \right)^{\frac{t}{2}} \qquad (C_0 = 44\kappa^{1.5} \frac{\hat{L}+2\beta}{2} C_1^2) \\
&\leq \frac{1 + 2\sqrt{44}\kappa\sqrt{\frac{\hat{L}+2\beta}{m_{\mathbf{x}}}}}{20\kappa\sqrt{2\kappa + \frac{\hat{L}}{m_{\mathbf{x}}}}} \cdot C_1 \left( 1 - \frac{1}{4\sqrt{\kappa}} \right)^{\frac{t}{2}} \qquad (2\sqrt{44} + 1 < 15) \\
&\leq \frac{3}{4} C_1 \left( 1 - \frac{1}{4\sqrt{\kappa}} \right)^{\frac{t}{2}} \leq C_1 \left( 1 - \frac{1}{4\sqrt{\kappa}} \right)^{\frac{t+1}{2}}.
\end{aligned}
$$

Therefore (9) holds for $t + 1$. Meanwhile, by (7) and Lemma 4,

$$
\begin{aligned}
\Delta_{t+1} &\leq \left( 1 - \frac{1}{2\sqrt{\kappa}} \right) \Delta_t + 11\kappa \cdot \frac{\hat{L} + 2\beta}{2} \epsilon_{t+1}^2 \\
&\leq \left( 1 - \frac{1}{2\sqrt{\kappa}} \right) C_0 \left( 1 - \frac{1}{4\sqrt{\kappa}} \right)^t + 11\kappa \cdot \frac{\hat{L} + 2\beta}{2} \cdot C_1^2 \left( 1 - \frac{1}{4\sqrt{\kappa}} \right)^t \\
&= C_0 \left( 1 - \frac{1}{4\sqrt{\kappa}} \right)^{t+1},
\end{aligned}
$$

where we used the fact that

$$11\kappa \cdot \frac{\hat{L} + 2\beta}{2} \cdot C_1^2 = \frac{1}{4\sqrt{\kappa}} \cdot 44\kappa^{1.5} \frac{\hat{L} + 2\beta}{2} C_1^2 = \frac{C_0}{4\sqrt{\kappa}}.$$

Thus (8) also holds for $t + 1$. By induction on $t$, we can see that (8) and (9) both hold for all $t \geq 0$.

As a result,

$$\|\mathbf{x}_T - \mathbf{x}^*\| \leq \sqrt{\frac{2}{m_\mathbf{x}} [\phi(\mathbf{x}_T) - \phi(\mathbf{x}^*)]} \leq \sqrt{\frac{2}{m_\mathbf{x}} \cdot 44\kappa\sqrt{\kappa} \frac{\hat{L} + 2\beta}{2} C_1^2 \left(1 - \frac{1}{4\sqrt{\kappa}}\right)^{\frac{T}{2}}}$$

$$\leq C_1 \left(1 - \frac{1}{4\sqrt{\kappa}}\right)^{\frac{T}{2}} \sqrt{88\kappa\sqrt{\kappa} \cdot \left(\frac{L^2}{m_\mathbf{x} m_\mathbf{y}} + \kappa\right)}.$$

Meanwhile,

$$\|\mathbf{y}_T - \mathbf{y}^*\| \leq \|\mathbf{y}_T - \mathbf{y}^*(\mathbf{x}_T)\| + \|\mathbf{y}^* - \mathbf{y}^*(\mathbf{x}_T)\| \leq \epsilon_T + \frac{L_\mathbf{xy}}{m_\mathbf{y}} \|\mathbf{x}_T - \mathbf{x}^*\|.$$

Therefore

$$\|\mathbf{x}_T - \mathbf{x}^*\| + \|\mathbf{y}_T - \mathbf{y}^*\| \leq \epsilon_T + \left(\frac{L_\mathbf{xy}}{m_\mathbf{y}} + 1\right) \|\mathbf{x}_T - \mathbf{x}^*\|$$

$$\leq C_1 \left(1 - \frac{1}{4\sqrt{\kappa}}\right)^{\frac{T}{2}} + \frac{2L}{m_\mathbf{y}} \cdot C_1 \left(1 - \frac{1}{4\sqrt{\kappa}}\right)^{\frac{T}{2}} \cdot \sqrt{88\kappa\sqrt{\kappa} \cdot \left(\frac{L^2}{m_\mathbf{x} m_\mathbf{y}} + \kappa\right)}$$

$$\leq C_1 \left(1 - \frac{1}{4\sqrt{\kappa}}\right)^{\frac{T}{2}} \cdot \left[1 + \frac{27\kappa^2 L}{m_\mathbf{y}} \sqrt{\frac{L^2}{m_\mathbf{x} m_\mathbf{y}}}\right]$$

$$\leq \frac{28\kappa^2 L}{m_\mathbf{y}} \sqrt{\frac{L^2}{m_\mathbf{x} m_\mathbf{y}}} \cdot \left(1 - \frac{1}{4\sqrt{\kappa}}\right)^{\frac{T}{2}} \cdot (\|\mathbf{x}_0 - \mathbf{x}^*\| + \|\mathbf{y}_0 - \mathbf{y}^*\|),$$

which proves the theorem.

$\square$

## 4  Proof of Theorem 3

**Theorem 3.** *Assume that $f \in \mathcal{F}(m_\mathbf{x}, m_\mathbf{y}, L_\mathbf{x}, L_\mathbf{xy}, L_\mathbf{y})$. In Algorithm 4, the gradient complexity to produce $(\mathbf{x}_T, \mathbf{y}_T)$ such that $\|\mathbf{z}_T - \mathbf{z}^*\| \leq \epsilon$ is*

$$O\left(\sqrt{\frac{L_\mathbf{x}}{m_\mathbf{x}} + \frac{L \cdot L_\mathbf{xy}}{m_\mathbf{x} m_\mathbf{y}} + \frac{L_\mathbf{y}}{m_\mathbf{y}}} \cdot \ln^3\left(\frac{L^2}{m_\mathbf{x} m_\mathbf{y}}\right) \ln\left(\frac{L^2}{m_\mathbf{x} m_\mathbf{y}} \cdot \frac{\|\mathbf{z}_0 - \mathbf{z}^*\|}{\epsilon}\right)\right).$$

*Proof.* We start the proof by verifying $f(\mathbf{x}, \mathbf{y}) + \beta_1 \|\mathbf{x} - \hat{\mathbf{x}}\|^2 - \beta_2 \|\mathbf{y} - \hat{\mathbf{y}}\|^2$ can indeed be solved by calling ABR$(\cdot, [\mathbf{x}_0; \mathbf{y}_0], 1/M_2, 2\beta_1, 2\beta_2, 3L, 3L)$. Observe that $L_\mathbf{xy} \leq \beta_1, \beta_2 \leq L$. Since $f(\mathbf{x}, \mathbf{y}) + \beta_1 \|\mathbf{x} - \hat{\mathbf{x}}\|^2 - \beta_2 \|\mathbf{y} - \hat{\mathbf{y}}\|^2$ is $2\beta_1$-strongly convex w.r.t. $\mathbf{x}$ and $2\beta_2$-strongly concave w.r.t. $\mathbf{y}$, we can see that $\frac{1}{2}\sqrt{2\beta_1 \cdot 2\beta_2} \geq L_\mathbf{xy}$. We can also verify that $f(\mathbf{x}, \mathbf{y}) + \beta_1 \|\mathbf{x} - \hat{\mathbf{x}}\|^2 - \beta_2 \|\mathbf{y} - \hat{\mathbf{y}}\|^2$ is $3L$-smooth, which follows from the fact that $L + \max\{2\beta_1, 2\beta_2\} \leq 3L$.

Therefore, we can apply Theorem 1 and conclude that at line 5 of Algorithm 3

$$\|\mathbf{x}_t - \mathbf{x}_t^*\| + \|\mathbf{y}_t - \mathbf{y}_t^*\| \leq \frac{1}{M_2} (\|\mathbf{x}_{t-1} - \mathbf{x}_t^*\| + \|\mathbf{y}_{t-1} - \mathbf{y}_t^*\|),$$

where $(\mathbf{x}_t^*, \mathbf{y}_t^*) := \min_\mathbf{x} \max_\mathbf{y} \{g(\mathbf{x}, \mathbf{y}) - \beta_2 \|\mathbf{y} - \mathbf{y}_{t-1}\|^2\}$, [1] and such $(\mathbf{x}_t, \mathbf{y}_t)$ is found in a gradient complexity of

$$O\left(\sqrt{\frac{L}{\beta_1} + \frac{L}{\beta_2}} \cdot \ln\left(\frac{L^2}{\beta_1 \beta_2}\right) \ln\left(\frac{L^2}{\beta_1 \beta_2} \cdot M_2\right)\right) = O\left(\sqrt{\frac{L}{\beta_1} + \frac{L}{\beta_2}} \cdot \ln^2\left(\frac{L^2}{m_\mathbf{x} m_\mathbf{y}}\right)\right).$$

Next, we verify that Algorithm 3 is an instance of Algorithm 2 on the function $\hat{g}(\mathbf{x}, \mathbf{y}) := -g(\mathbf{y}, \mathbf{x})$. Notice that

$$\min_{\mathbf{y}} \max_{\mathbf{x}} \left\{ -g(\mathbf{x}, \mathbf{y}) + \beta \|\mathbf{y} - \hat{\mathbf{y}}\|^2 \right\} = -\min_{\mathbf{x}} \max_{\mathbf{y}} \left\{ g(\mathbf{x}, \mathbf{y}) - \beta \|\mathbf{y} - \hat{\mathbf{y}}\|^2 \right\}.$$

That is, $\min_{\mathbf{x}} \max_{\mathbf{y}} \left\{ g(\mathbf{x}, \mathbf{y}) - \|\mathbf{y} - \hat{\mathbf{y}}\|^2 \right\}$ has the same saddle point as $-g(\mathbf{x}, \mathbf{y}) + \beta \|\mathbf{y} - \hat{\mathbf{y}}\|^2$. Thus, we only need to verify that

$$M_2 \geq 20 \cdot \frac{\beta_2}{m_{\mathbf{y}}'} \left(1 + \frac{L'}{m_{\mathbf{x}}'}\right) \sqrt{\frac{2\beta_2}{m_{\mathbf{y}}'} + \frac{L'}{m_{\mathbf{y}}'} + \frac{L_{\mathbf{xy}}^2}{m_{\mathbf{x}} m_{\mathbf{y}}}}, \tag{12}$$

where $(m_{\mathbf{x}}', m_{\mathbf{y}}', L', L_{\mathbf{xy}}, L_{\mathbf{y}}')$ are parameters for $f(\mathbf{x}, \mathbf{y}) + \beta_1 \|\mathbf{x}\|^2$, and $L' = \max\{L_{\mathbf{xy}}, L_{\mathbf{x}}', L_{\mathbf{y}}'\}$. Note that $m_{\mathbf{x}}' \geq m_{\mathbf{x}} + 2\beta_1$, $m_{\mathbf{y}}' = m_{\mathbf{y}}$, $L_{\mathbf{x}}' = L_{\mathbf{y}}' \leq L + 2\beta_1$, $L_{\mathbf{xy}} \leq \beta_1, \beta_2 \leq L$. Thus

$$\text{RHS of (12)} \leq 20 \cdot \frac{\beta_2}{m_{\mathbf{y}}} \sqrt{\frac{2\beta_2}{m_{\mathbf{y}}'} + \frac{L + 2\beta_1}{m_{\mathbf{y}}} + \frac{L_{\mathbf{xy}}^2}{m_{\mathbf{y}}(m_{\mathbf{x}} + 2\beta_1)}} \cdot \left(1 + \frac{L + 2\beta_1}{m_{\mathbf{x}} + 2\beta_1}\right)$$

$$\leq 20 \cdot \frac{L}{m_{\mathbf{y}}} \sqrt{\frac{2L}{m_{\mathbf{y}}} + \frac{3L}{m_{\mathbf{y}}} + \frac{L_{\mathbf{xy}}}{2m_{\mathbf{y}}}} \left(1 + \frac{L}{m_{\mathbf{x}}}\right)$$

$$\leq \frac{96 L^{2.5}}{m_{\mathbf{x}} m_{\mathbf{y}}^{1.5}} = M_2.$$

Therefore, Algorithm 3 is indeed an instance of Inexact APPA (Algorithm II). Notice that by the stopping condition of Algorithm 3,

$$(\|\mathbf{x}_t - \mathbf{x}^*\| + \|\mathbf{y}_t - \mathbf{y}^*\|) \leq \frac{\sqrt{2}}{\min\{m_{\mathbf{x}}, m_{\mathbf{y}}\}} \|\nabla g(\mathbf{x}_t, \mathbf{y}_t)\| \qquad \text{(Fact 3 and 2)}$$

$$\leq \frac{\sqrt{2}}{\min\{m_{\mathbf{x}}, m_{\mathbf{y}}\}} \cdot \frac{\min\{m_{\mathbf{x}}, m_{\mathbf{y}}\}}{9 L M_1} \|\nabla g(\mathbf{x}_0, \mathbf{y}_0)\|$$

$$\leq \frac{\sqrt{2}}{\min\{m_{\mathbf{x}}, m_{\mathbf{y}}\}} \cdot \frac{\min\{m_{\mathbf{x}}, m_{\mathbf{y}}\}}{9 L M_1} \cdot 6L \left(\|\mathbf{x}_0 - \mathbf{x}^*\| + \|\mathbf{y}_0 - \mathbf{y}^*\|\right)$$

$$\leq \frac{1}{M_1} \left(\|\mathbf{x}_0 - \mathbf{x}^*\| + \|\mathbf{y}_0 - \mathbf{y}^*\|\right).$$

Thus when Algorithm 3 returns,

$$\|\mathbf{x}_t - \mathbf{x}^*\| + \|\mathbf{y}_t - \mathbf{y}^*\| \leq \frac{1}{M_1} \left(\|\mathbf{x}_0 - \mathbf{x}^*\| + \|\mathbf{y}_0 - \mathbf{y}^*\|\right) \tag{13}$$

On the other hand, suppose that

$$\|\mathbf{x}_t - \mathbf{x}^*\| + \|\mathbf{y}_t - \mathbf{y}^*\| \leq \frac{1}{M_1} \frac{\min\{m_{\mathbf{x}}, m_{\mathbf{y}}\}}{12L} \cdot \left(\|\mathbf{x}_0 - \mathbf{x}^*\| + \|\mathbf{y}_0 - \mathbf{y}^*\|\right),$$

we can show that

$$\|\nabla g(\mathbf{x}_t, \mathbf{y}_t)\| \leq 6L \left(\|\mathbf{x}_t - \mathbf{x}^*\| + \|\mathbf{y}_t - \mathbf{y}^*\|\right)$$

$$\leq \frac{\min\{m_{\mathbf{x}}, m_{\mathbf{y}}\}}{2 M_1} \left(\|\mathbf{x}_0 - \mathbf{x}^*\| + \|\mathbf{y}_0 - \mathbf{y}^*\|\right)$$

$$\leq \frac{1}{M_1} \|\nabla g(\mathbf{x}_0, \mathbf{y}_0)\|.$$

Thus in this case Algorithm 3 must return. By Theorem 2, we can see that Algorithm 3 always returns in at most

$$O\left(\sqrt{\frac{\beta_2}{m_{\mathbf{y}}}} \cdot \ln\left(\frac{L^2}{m_{\mathbf{x}} m_{\mathbf{y}}} \cdot \frac{12L}{\min\{m_{\mathbf{x}}, m_{\mathbf{y}}\}} M_1\right)\right) = O\left(\sqrt{\frac{\beta_2}{m_{\mathbf{y}}}} \cdot \ln\left(\frac{L^2}{m_{\mathbf{x}} m_{\mathbf{y}}}\right)\right) \tag{14}$$

iterations.

Finally, we verify that Algorithm 4 is an instance of Algorithm 2 on $f(\mathbf{x}, \mathbf{y})$ with parameter $\beta_1$. Note that by (13), we only need to verify that

$$M_1 = \frac{80L^3}{m_{\mathbf{x}}^{1.5} m_{\mathbf{y}}^{1.5}} \geq 20 \cdot \frac{\beta_1}{m_{\mathbf{x}}} \sqrt{\frac{2\beta_1}{m_{\mathbf{x}}} + \frac{L}{m_{\mathbf{x}}} + \frac{L_{\mathbf{xy}}^2}{m_{\mathbf{x}} m_{\mathbf{y}}}} \left(1 + \frac{L}{m_{\mathbf{y}}}\right).$$

Observe that

$$20 \cdot \frac{\beta_1}{m_{\mathbf{x}}} \sqrt{\frac{2\beta_1}{m_{\mathbf{x}}} + \frac{L}{m_{\mathbf{x}}} + \frac{L_{\mathbf{xy}}^2}{m_{\mathbf{x}} m_{\mathbf{y}}}} \left(1 + \frac{L}{m_{\mathbf{y}}}\right) \leq 20 \cdot \frac{L}{m_{\mathbf{x}}} \sqrt{\frac{2L}{m_{\mathbf{x}}} + \frac{L}{m_{\mathbf{x}}} + \frac{L^2}{m_{\mathbf{x}} m_{\mathbf{y}}}} \cdot \frac{2L}{m_{\mathbf{y}}}$$

$$\leq 20 \cdot \frac{L}{m_{\mathbf{x}}} \cdot \sqrt{\frac{4L^2}{m_{\mathbf{x}} m_{\mathbf{y}}}} \cdot \frac{2L}{m_{\mathbf{y}}} = M_1.$$

Therefore Algorithm 4 is indeed an instance of Algorithm 2 on $f(\mathbf{x}, \mathbf{y})$. As a result, by Theorem 2, the number of iterations needed such that $\|\mathbf{z}_T - \mathbf{z}^*\| \leq \epsilon$ is

$$O\left(\sqrt{\frac{\beta_1}{m_{\mathbf{x}}}} \cdot \ln\left(\frac{L^2}{m_{\mathbf{x}} m_{\mathbf{y}}} \cdot \frac{\|\mathbf{z}_0 - \mathbf{z}^*\|}{\epsilon}\right)\right). \tag{15}$$

We now compute the total gradient complexity. Recall that $\beta_1 = \max\{m_{\mathbf{x}}, L_{\mathbf{xy}}\}$, while $\beta_2 = \max\{m_{\mathbf{y}}, L_{\mathbf{xy}}\}$. By (15), (14) and (4), the total gradient complexity of Algorithm 4 to reach $\|\mathbf{z}_T - \mathbf{z}^*\| \leq \epsilon$ is

$$O\left(\sqrt{\frac{\beta_1}{m_{\mathbf{x}}}} \cdot \ln\left(\frac{L^2}{m_{\mathbf{x}} m_{\mathbf{y}}} \cdot \frac{\|\mathbf{z}_0 - \mathbf{z}^*\|}{\epsilon}\right) \cdot \sqrt{\frac{\beta_2}{m_{\mathbf{y}}}} \cdot \ln\left(\frac{L^2}{m_{\mathbf{x}} m_{\mathbf{y}}}\right) \cdot \sqrt{\frac{L}{\beta_1} + \frac{L}{\beta_2}} \cdot \ln^2\left(\frac{L^2}{m_{\mathbf{x}} m_{\mathbf{y}}}\right)\right)$$

$$= O\left(\sqrt{\frac{L(\beta_1 + \beta_2)}{m_{\mathbf{x}} m_{\mathbf{y}}}} \cdot \ln^3\left(\frac{L^2}{m_{\mathbf{x}} m_{\mathbf{y}}}\right) \ln\left(\frac{L^2}{m_{\mathbf{x}} m_{\mathbf{y}}} \cdot \frac{\|\mathbf{z}_0 - \mathbf{z}^*\|}{\epsilon}\right)\right).$$

If $L_{\mathbf{xy}} \geq \max\{m_{\mathbf{x}}, m_{\mathbf{y}}\}$, then $\beta_1 = \beta_2 = L_{\mathbf{xy}}$, so

$$\sqrt{\frac{L(\beta_1 + \beta_2)}{m_{\mathbf{x}} m_{\mathbf{y}}}} = \sqrt{\frac{2L \cdot L_{\mathbf{xy}}}{m_{\mathbf{x}} m_{\mathbf{y}}}} \leq 2\sqrt{\frac{L_{\mathbf{x}}}{m_{\mathbf{x}}} + \frac{L \cdot L_{\mathbf{xy}}}{m_{\mathbf{x}} m_{\mathbf{y}}} + \frac{L_{\mathbf{y}}}{m_{\mathbf{y}}}}.$$

Now consider the case where $L_{\mathbf{xy}} < \max\{m_{\mathbf{x}}, m_{\mathbf{y}}\}$. Without loss of generality, assume that $m_{\mathbf{x}} \leq m_{\mathbf{y}}$. Suppose that $L_{\mathbf{xy}} < m_{\mathbf{y}}$, then $L = L_{\mathbf{x}}$, $\beta_2 = m_{\mathbf{y}}$, while $\beta_1 \leq m_{\mathbf{y}}$. Hence

$$\sqrt{\frac{L(\beta_1 + \beta_2)}{m_{\mathbf{x}} m_{\mathbf{y}}}} \leq \sqrt{\frac{L_{\mathbf{x}} \cdot 2m_{\mathbf{y}}}{m_{\mathbf{x}} m_{\mathbf{y}}}} = \sqrt{\frac{2L_{\mathbf{x}}}{m_{\mathbf{x}}}} \leq 2\sqrt{\frac{L_{\mathbf{x}}}{m_{\mathbf{x}}} + \frac{L \cdot L_{\mathbf{xy}}}{m_{\mathbf{x}} m_{\mathbf{y}}} + \frac{L_{\mathbf{y}}}{m_{\mathbf{y}}}}.$$

Thus, in either case, $\sqrt{\frac{L(\beta_1 + \beta_2)}{m_{\mathbf{x}} m_{\mathbf{y}}}} = O\left(\sqrt{\frac{L_{\mathbf{x}}}{m_{\mathbf{x}}} + \frac{L \cdot L_{\mathbf{xy}}}{m_{\mathbf{x}} m_{\mathbf{y}}} + \frac{L_{\mathbf{y}}}{m_{\mathbf{y}}}}\right)$. We conclude that the total gradient complexity of Algorithm 4 to find a point $\mathbf{z}_T = [\mathbf{x}_T; \mathbf{y}_T]$ such that $\|\mathbf{z}_T - \mathbf{z}^*\| \leq \epsilon$ is

$$O\left(\sqrt{\frac{L_{\mathbf{x}}}{m_{\mathbf{x}}} + \frac{L \cdot L_{\mathbf{xy}}}{m_{\mathbf{x}} m_{\mathbf{y}}} + \frac{L_{\mathbf{y}}}{m_{\mathbf{y}}}} \cdot \ln^3\left(\frac{L^2}{m_{\mathbf{x}} m_{\mathbf{y}}}\right) \ln\left(\frac{L^2}{m_{\mathbf{x}} m_{\mathbf{y}}} \cdot \frac{\|\mathbf{z}_0 - \mathbf{z}^*\|}{\epsilon}\right)\right).$$

$\square$

## 5  Application to Constrained Problems

In the constrained minimax optimization problem, $\mathbf{x}$ is constrained to a compact convex set $\mathcal{X} \subseteq \mathbb{R}^n$ while $\mathbf{y}$ is constrained to a compact convex set $\mathcal{Y} \subseteq \mathbb{R}^m$. For constrained minimax optimization problems, saddle points are defined as follows.

**Definition 5.** $(\mathbf{x}^*, \mathbf{y}^*)$ is a saddle point of $f : \mathcal{X} \times \mathcal{Y} \to \mathbb{R}$ if $\forall \mathbf{x} \in \mathcal{X}, \mathbf{y} \in \mathcal{Y}$,

$$f(\mathbf{x}, \mathbf{y}^*) \geq f(\mathbf{x}^*, \mathbf{y}^*) \geq f(\mathbf{x}^*, \mathbf{y}).$$

**Definition 6.** $(\hat{\mathbf{x}}, \hat{\mathbf{y}})$ is an $\epsilon$-saddle point of $f : \mathcal{X} \times \mathcal{Y} \to \mathbb{R}$ if

$$\max_{\mathbf{y} \in \mathcal{Y}} f(\hat{\mathbf{x}}, \mathbf{y}) - \min_{\mathbf{x} \in \mathcal{X}} f(\mathbf{x}, \hat{\mathbf{y}}) \le \epsilon.$$

We will use $P_{\mathcal{X}}[\cdot]$ to denote the projection onto convex set $\mathcal{X}$. Assuming efficient projection oracles, our algorithms can all be easily adapted to the constrained case. In particular, for Algorithm 1, we only need to replace AGD with the constrained version; that is, set $\mathbf{x}_t \leftarrow P_{\mathcal{X}}[\tilde{\mathbf{x}}_{t-1} - \eta \nabla g(\tilde{x}_{t-1})]$.

For Algorithm 3 and 4, the modified versions are presented below. The only significant change is the addition of a projected gradient descent-ascent step in line 5-6 of Algorithm 3 and line 5-6 and 9-10 of Algorithm 4.

### 5.1 Algorithmic Modifications

---
**Algorithm III** AGD($g$, $\mathbf{x}_0$, $T$) with Projections [6, (2.2.63)]

---
**Require:** Initial point $\mathbf{x}_0$, smoothness constant $l$, strongly-convex modulus $m$, number of iterations $T$
1: $\eta \leftarrow 1/l$, $\kappa \leftarrow l/m$, $\theta \leftarrow (\sqrt{\kappa} - 1)/(\sqrt{\kappa} + 1)$
2: $\mathbf{x}_1 \leftarrow P_{\mathcal{X}}[\mathbf{x}_0 - \eta \nabla g(\mathbf{x}_0)]$, $\tilde{\mathbf{x}}_1 \leftarrow \mathbf{x}_1$
3: **for** $t = 2, \cdots, T + 1$ **do**
4: $\quad \mathbf{x}_t \leftarrow P_{\mathcal{X}}[\tilde{\mathbf{x}}_{t-1} - \eta \nabla g(\tilde{\mathbf{x}}_{t-1})]$
5: $\quad \tilde{\mathbf{x}}_t \leftarrow \mathbf{x}_t + \theta(\mathbf{x}_t - \mathbf{x}_{t-1})$
6: **end for**

---

For Algorithm 1, the only necessary modification is to add projection steps to the Accelerated Gradient Descent Procedure. The reason for the extra gradient step on line 2 is technical. From the original analysis [6, Theorem 2.2.3], it only follows that

$$\|\mathbf{x}_{T+1} - \mathbf{x}^*\|^2 \le \left[ \|\mathbf{x}_1 - \mathbf{x}^*\|^2 + \frac{2}{m}\left(f(\mathbf{x}_1) - f(\mathbf{x}^*)\right) \right] \cdot \left(1 - \frac{1}{\sqrt{\kappa}}\right)^T.$$

For constrained problems, $f(\mathbf{x}_1) - f(\mathbf{x}^*) \le \frac{L}{2}\|\mathbf{x}_1 - \mathbf{x}^*\|^2$ does not hold. However, with the initial projected gradient step, it can be shown that $\|\mathbf{x}_1 - \mathbf{x}^*\| \le \|\mathbf{x}_0 - \mathbf{x}^*\|$ and that $f(\mathbf{x}_1) - f(\mathbf{x}^*) \le \frac{L}{2}\|\mathbf{x}_0 - \mathbf{x}^*\|^2$ (see Lemma 6). Thus

$$\|\mathbf{x}_{T+1} - \mathbf{x}^*\|^2 \le (\kappa + 1)\|\mathbf{x}_0 - \mathbf{x}^*\|^2 \left(1 - \frac{1}{\sqrt{\kappa}}\right)^T.$$

For Algorithm 3 and 4, the modified versions are presented below.

---
**Algorithm 3** APPA-ABR (for Constrained Optimization)

---
**Require:** $g(\cdot, \cdot)$, Initial point $\mathbf{z}_0 = [\mathbf{x}_0; \mathbf{y}_0]$, precision parameter $M_1$
1: $\beta_2 \leftarrow \max\{m_{\mathbf{y}}, L_{\mathbf{xy}}\}$, $M_2 \leftarrow \frac{200L^3}{m_{\mathbf{x}} m_{\mathbf{y}}^2}$
2: $\hat{\mathbf{y}}_0 \leftarrow \mathbf{y}_0$ $\kappa \leftarrow \beta_2/m_{\mathbf{y}}$, $\theta \leftarrow \frac{2\sqrt{\kappa}-1}{2\sqrt{\kappa}+1}$, $\tau \leftarrow \frac{1}{2\sqrt{\kappa}+4\kappa}$, $T \leftarrow \left\lceil 8\sqrt{\kappa} \ln\left(\frac{400\kappa^2 L^2 M_1}{m_{\mathbf{x}}\sqrt{m_{\mathbf{x}} m_{\mathbf{y}}}}\right) \right\rceil$
3: **for** $t = 1, \cdots, T$ **do**
4: $\quad (\mathbf{x}_t', \mathbf{y}_t') \leftarrow \text{ABR}(g(\mathbf{x}, \mathbf{y}) - \beta_2\|\mathbf{y} - \hat{\mathbf{y}}_{t-1}\|^2, [\mathbf{x}_{t-1}; \mathbf{y}_{t-1}], 1/M_2, 2\beta_1, 2\beta_2, 3L, 3L)$
5: $\quad \mathbf{x}_t \leftarrow P_{\mathcal{X}}\left[\mathbf{x}_t' - \frac{1}{6L}\nabla_{\mathbf{x}} g(\mathbf{x}_t', \mathbf{y}_t')\right]$
6: $\quad \mathbf{y}_t \leftarrow P_{\mathcal{Y}}\left[\mathbf{y}_t' + \frac{1}{6L}\left(\nabla_{\mathbf{y}} g(\mathbf{x}_t', \mathbf{y}_t') - 2\beta_2(\mathbf{y}_t' - \hat{\mathbf{y}}_{t-1})\right)\right]$
7: $\quad \hat{\mathbf{y}}_t \leftarrow \mathbf{y}_t + \theta(\mathbf{y}_t - \mathbf{y}_{t-1}) + \tau(\mathbf{y}_t - \hat{\mathbf{y}}_{t-1})$
8: **end for**

---

The most significant change is the addition of a projected gradient descent-ascent step in line 5-6 of Algorithm 3 and line 5-6 and 9-10 of Algorithm 4. The reason for this modification is very similar to that of the initial projected gradient descent step for AGD. For unconstrained problems, a small distance to the saddle point implies a small duality gap (Fact 4); however this may not be true for constrained problems, since the saddle point may no longer be a stationary point. This is also true for minimization: if $\mathbf{x}^* = \arg\min_{\mathbf{x} \in \mathcal{X}} g(\mathbf{x})$ where $g(\mathbf{x})$ is a $L$-smooth function $g(\mathbf{x}) - g(\mathbf{x}^*) \le \frac{L}{2}\|\mathbf{x} - \mathbf{x}^*\|^2$ may not hold.

---

**Algorithm 4** Proximal Best Response (for Constrained Optimization)

---

**Require:** Initial point $\mathbf{z}_0 = [\mathbf{x}_0; \mathbf{y}_0]$

1: $\beta_1 \leftarrow \max\{m_{\mathbf{x}}, L_{\mathbf{xy}}\}, M_1 \leftarrow \frac{120L^{3.5}}{m_{\mathbf{x}}^2 m_{\mathbf{y}}^{1.5}}$

2: $\hat{\mathbf{x}}_0 \leftarrow \mathbf{x}_0, \kappa \leftarrow \beta_1/m_{\mathbf{x}}, \theta \leftarrow \frac{2\sqrt{\kappa}-1}{2\sqrt{\kappa}+1}, \tau \leftarrow \frac{1}{2\sqrt{\kappa}+4\kappa}$

3: **for** $t = 1, \cdots, T$ **do**

4: $\quad (\mathbf{x}_t', \mathbf{y}_t') \leftarrow \text{APPA-ABR}(f(\mathbf{x}, \mathbf{y}) + \beta_1\|\mathbf{x} - \hat{\mathbf{x}}_{t-1}\|^2, [\mathbf{x}_{t-1}, \mathbf{y}_{t-1}], M_1)$

5: $\quad \mathbf{x}_t \leftarrow P_{\mathcal{X}}\left[\mathbf{x}_t' - \frac{1}{6L}\left(\nabla_{\mathbf{x}} f(\mathbf{x}_t', \mathbf{y}_t') + 2\beta_1(\mathbf{x}_t' - \hat{\mathbf{x}}_{t-1})\right)\right]$

6: $\quad \mathbf{y}_t \leftarrow P_{\mathcal{Y}}\left[\mathbf{y}_t' + \frac{1}{6L}\nabla_{\mathbf{y}} f(\mathbf{x}_t', \mathbf{y}_t')\right]$

7: $\quad \hat{\mathbf{x}}_t \leftarrow \mathbf{x}_t + \theta(\mathbf{x}_t - \mathbf{x}_{t-1}) + \tau(\mathbf{x}_t - \hat{\mathbf{x}}_{t-1})$

8: **end for**

9: $\hat{\mathbf{x}} \leftarrow P_{\mathcal{X}}\left[\mathbf{x}_T - \frac{1}{2L}\nabla_{\mathbf{x}} f(\mathbf{x}_T, \mathbf{y}_T)\right]$

10: $\hat{\mathbf{y}} \leftarrow P_{\mathcal{Y}}\left[\mathbf{y}_T + \frac{1}{2L}\nabla_{\mathbf{y}} f(\mathbf{x}_T, \mathbf{y}_T)\right]$

---

Fortunately, there is a simple fix to this problem. By applying projected gradient descent-ascent once, we can assure that a small distance implies small duality gap. This is specified by the following lemma, which is the key reason why our result can be adapted to the constrained problem.

**Lemma 5.** *Suppose that* $f \in \mathcal{F}(m_{\mathbf{x}}, m_{\mathbf{y}}, L_{\mathbf{x}}, L_{\mathbf{xy}}, L_{\mathbf{y}})$, $(\mathbf{x}^*, \mathbf{y}^*)$ *is a saddle point of* $f$, $\mathbf{z}_0 = (\mathbf{x}_0, \mathbf{y}_0)$ *satisfies* $\|\mathbf{z}_0 - \mathbf{z}^*\| \leq \epsilon$. *Let* $\hat{\mathbf{z}} = (\hat{\mathbf{x}}, \hat{\mathbf{y}})$ *be the result of one projected GDA update, i.e.*

$$\hat{\mathbf{x}} \leftarrow P_{\mathcal{X}}\left[\mathbf{x}_0 - \frac{1}{2L}\nabla_{\mathbf{x}} f(\mathbf{x}_0, \mathbf{y}_0)\right],$$

$$\hat{\mathbf{y}} \leftarrow P_{\mathcal{Y}}\left[\mathbf{y}_0 + \frac{1}{2L}\nabla_{\mathbf{y}} f(\mathbf{x}_0, \mathbf{y}_0)\right].$$

*Then* $\|\hat{\mathbf{z}} - \mathbf{z}^*\| \leq \epsilon$, *and*

$$\max_{\mathbf{y} \in \mathcal{Y}} f(\hat{\mathbf{x}}, \mathbf{y}) - \min_{\mathbf{x} \in \mathcal{X}} f(\mathbf{x}, \hat{\mathbf{y}}) \leq 2\left(1 + \frac{L_{\mathbf{xy}}^2}{\min\{m_{\mathbf{x}}, m_{\mathbf{y}}\}^2}\right)L\epsilon^2.$$

The proof of Lemma 5 is deferred to Sec. 5.3.

Because we would use Lemma 5 to replace (7) in the analysis of Algorithm 3 and 4, we would need to accordingly increase $M_1$ to $\frac{120L^{3.5}}{m_{\mathbf{x}}^2 m_{\mathbf{y}}^{1.5}}$ and $M_2$ to $\frac{200L^3}{m_{\mathbf{x}} m_{\mathbf{y}}^2}$. Apart from this, another minor change in Algorithm 3 is that it would terminate after a fixed number of iterations instead of based on a termination criterion. The number of iterations is chosen such that $\|\mathbf{x}_T - \mathbf{x}^*\| + \|\mathbf{y}_T - \mathbf{y}^*\| \leq \frac{1}{M_1}[\|\mathbf{x}_0 - \mathbf{x}^*\| + \|\mathbf{y}_0 - \mathbf{y}^*\|]$ is guaranteed.

## 5.2 Modification of Analysis

We now claim that after modifications to the algorithms, Theorem 3 holds for constrained cases.

**Theorem 3.** *(Modified) Assume that* $f \in \mathcal{F}(m_{\mathbf{x}}, m_{\mathbf{y}}, L_{\mathbf{x}}, L_{\mathbf{xy}}, L_{\mathbf{y}})$. *In Algorithm 4, the gradient complexity to find an* $\epsilon$-saddle point

$$O\left(\sqrt{\frac{L_{\mathbf{x}}}{m_{\mathbf{x}}} + \frac{L \cdot L_{\mathbf{xy}}}{m_{\mathbf{x}} m_{\mathbf{y}}} + \frac{L_{\mathbf{y}}}{m_{\mathbf{y}}}} \cdot \ln^3\left(\frac{L^2}{m_{\mathbf{x}} m_{\mathbf{y}}}\right) \ln\left(\frac{L^2}{m_{\mathbf{x}} m_{\mathbf{y}}} \cdot \frac{L\|\mathbf{z}_0 - \mathbf{z}^*\|^2}{\epsilon}\right)\right).$$

The proof of this theorem is, for the most part, the same as the unconstrained version. Hence, we only need to point out parts of the original proof that need to be modified for the constrained case.

To start with, Theorem 1 holds in the constrained case. The proof of Theorem 1 only relies on the analysis of AGD and the Lipschitz properties in Fact 1, and both still hold for constrained problems. (See [3, Lemma B.2] for the proof of Fact 1 in constrained problems.)

As for Theorem 2, the key modification is about (7). As argued above, (7) uses the property $g(\mathbf{x}) - g(\mathbf{x}^*) \leq \frac{L}{2}\|\mathbf{x} - \mathbf{x}^*\|^2$, which does not hold in constrained problems, since the optimum may not be a stationary point.

Here, we would use Lemma 5 to derive a similar bound to replace (7). Note that originally (7) is only used to derive $\delta_t \leq \frac{\hat{L}+2\beta}{2}\epsilon_t^2$. Using Lemma 5, we can replace this with

$$\delta_t \leq \max_{\mathbf{y}\in\mathcal{Y}}\left\{f(\mathbf{x}_t,\mathbf{y}) + \beta\|\mathbf{x}_t - \hat{\mathbf{x}}_{t-1}\|^2\right\} - \min_{\mathbf{x}\in\mathcal{X}}\left\{f(\mathbf{x},\mathbf{y}_t) + \beta\|\mathbf{x} - \hat{\mathbf{x}}_{t-1}\|^2\right\}$$

$$\leq 2\left(1 + \frac{L_{\mathbf{xy}}^2}{m_{\mathbf{x}}m_{\mathbf{y}}}\right)L\epsilon_t^2.$$

Accordingly, we can change $C_0$ to $44\kappa\sqrt{\kappa}\cdot 2L\left(1 + \frac{L_{\mathbf{xy}}^2}{m_{\mathbf{x}}m_{\mathbf{y}}}\right)C_1^2$, and the assumption on $M$ to $M \geq 20\kappa\sqrt{\frac{4L}{m_{\mathbf{x}}}\left(1 + \frac{L_{\mathbf{xy}}^2}{m_{\mathbf{x}}m_{\mathbf{y}}}\right)}\left(1 + \frac{L}{m_{\mathbf{y}}}\right)$. Then Theorem 2 would hold for the constrained case as well.

Finally, as for Theorem 3, we need to re-verify that $M_1$ and $M_2$ satisfy the new assumptions of $M$ in order to apply Theorem 2. Observe that

$$20\cdot\frac{\beta_2}{m_{\mathbf{y}}}\cdot\sqrt{\frac{4(L+2\beta_1)}{m_{\mathbf{y}}}\cdot\left(1 + \frac{L_{\mathbf{xy}}^2}{2\beta_1\cdot m_{\mathbf{y}}}\right)}\cdot\left(1 + \frac{L}{m_{\mathbf{x}}}\right)$$

$$\leq 20\cdot\frac{L}{m_{\mathbf{y}}}\cdot\sqrt{\frac{18L^3}{m_{\mathbf{x}}m_{\mathbf{y}}^2}}\cdot\frac{2L}{m_{\mathbf{x}}} \leq \frac{200L^3}{m_{\mathbf{x}}m_{\mathbf{y}}^2} = M_2,$$

and that

$$20\cdot\frac{\beta_1}{m_{\mathbf{x}}}\cdot\sqrt{\frac{4L}{m_{\mathbf{x}}}\cdot\frac{2L_{\mathbf{xy}}^2}{m_{\mathbf{x}}m_{\mathbf{y}}}}\cdot\frac{2L}{m_{\mathbf{y}}} \leq \frac{80\sqrt{2}L^{3.5}}{m_{\mathbf{x}}^2m_{\mathbf{y}}^{1.5}} \leq M_1.$$

It follows that the number of iterations needed to find $\|\mathbf{z}_T - \mathbf{z}^*\| \leq \epsilon$ is

$$O\left(\sqrt{\frac{L_{\mathbf{x}}}{m_{\mathbf{x}}} + \frac{L\cdot L_{\mathbf{xy}}}{m_{\mathbf{x}}m_{\mathbf{y}}} + \frac{L_{\mathbf{y}}}{m_{\mathbf{y}}}}\cdot\ln^3\left(\frac{L^2}{m_{\mathbf{x}}m_{\mathbf{y}}}\right)\ln\left(\frac{L^2}{m_{\mathbf{x}}m_{\mathbf{y}}}\cdot\frac{\|\mathbf{z}_0 - \mathbf{z}^*\|}{\epsilon}\right)\right).$$

It follows from Lemma 5 that the duality gap of $(\hat{\mathbf{x}},\hat{\mathbf{y}})$ is at most

$$\max_{\mathbf{y}\in\mathcal{Y}} f(\hat{\mathbf{x}},\mathbf{y}) - \min_{\mathbf{x}\in\mathcal{X}} f(\mathbf{x},\hat{\mathbf{y}}) \leq 2\left(1 + \frac{L_{\mathbf{xy}}^2}{\min\{m_{\mathbf{x}},m_{\mathbf{y}}\}^2}\right)L\epsilon^2.$$

Resetting $\epsilon$ to $\sqrt{\frac{\epsilon\min\{m_{\mathbf{x}},m_{\mathbf{y}}\}^2}{4L^3}}$ proves the theorem.

### 5.3 Properties of Projected Gradient

**Lemma 6.** *If* $g : \mathcal{X} \to \mathbb{R}$ *is* $L$-*smooth,* $\mathbf{x}^* = \arg\min_{\mathbf{x}\in\mathcal{X}} g(\mathbf{x})$, $\hat{\mathbf{x}} = P_{\mathcal{X}}\left[\mathbf{x}_0 - \frac{1}{L}\nabla g(\mathbf{x}_0)\right]$, *then* $\|\hat{\mathbf{x}} - \mathbf{x}^*\| \leq \|\mathbf{x}_0 - \mathbf{x}^*\|$, *and* $g(\hat{\mathbf{x}}) - g(\mathbf{x}^*) \leq \frac{L}{2}\|\mathbf{x}_0 - \mathbf{x}^*\|^2$.

*Proof.* By Corollary 2.2.1 [6], $(\mathbf{x}_0 - \hat{\mathbf{x}})^T(\mathbf{x}_0 - \mathbf{x}^*) \geq \frac{1}{2}\|\hat{\mathbf{x}} - \mathbf{x}_0\|^2$. Therefore

$$\|\hat{\mathbf{x}} - \mathbf{x}^*\|^2 = \|(\mathbf{x}_0 - \mathbf{x}^*) + (\hat{\mathbf{x}} - \mathbf{x}_0)\|^2$$
$$= \|\mathbf{x}_0 - \mathbf{x}^*\|^2 + 2(\mathbf{x}_0 - \mathbf{x}^*)^T(\hat{\mathbf{x}} - \mathbf{x}_0) + \|\hat{\mathbf{x}} - \mathbf{x}_0\|^2$$
$$\leq \|\mathbf{x}_0 - \mathbf{x}^*\|^2.$$

Meanwhile, note that $\hat{\mathbf{x}} = \arg\min_{\mathbf{x}\in\mathcal{X}}\left\{\nabla g(\mathbf{x}_0)^T\mathbf{x} + \frac{L}{2}\|\mathbf{x} - \mathbf{x}_0\|^2\right\}$. By the optimality condition and the $L$-strong convexity of $\nabla g(\mathbf{x}_0)^T\mathbf{x} + \frac{L}{2}\|\mathbf{x} - \mathbf{x}_0\|^2$, we have

$$\nabla g(\mathbf{x}_0)^T\hat{\mathbf{x}} + \frac{L}{2}\|\hat{\mathbf{x}} - \mathbf{x}_0\|^2 + \frac{L}{2}\|\mathbf{x}_1 - \mathbf{x}^*\|^2 \leq \nabla g(\mathbf{x}_0)^T\mathbf{x}^* + \frac{L}{2}\|\mathbf{x}^* - \mathbf{x}_0\|^2.$$

Thus

$$\nabla g(\mathbf{x}_0)^T(\hat{\mathbf{x}} - \mathbf{x}^*) \leq \frac{L}{2}\left[\|\mathbf{x}^* - \mathbf{x}_0\|^2 - \|\hat{\mathbf{x}} - \mathbf{x}_0\|^2 - \|\hat{\mathbf{x}} - \mathbf{x}^*\|^2\right].$$

It follows that

$$g(\hat{\mathbf{x}}) - g(\mathbf{x}^*) \le \nabla g(\hat{\mathbf{x}})^T (\hat{\mathbf{x}} - \mathbf{x}^*)$$

$$= \nabla g(\mathbf{x}_0)^T (\hat{\mathbf{x}} - \mathbf{x}^*) + (\nabla g(\hat{\mathbf{x}}) - \nabla g(\mathbf{x}_0))^T (\hat{\mathbf{x}} - \mathbf{x}^*)$$

$$\le \frac{L}{2} \|\mathbf{x}^* - \mathbf{x}_0\|^2 \underbrace{- \frac{L}{2} \|\hat{\mathbf{x}} - \mathbf{x}_0\|^2 - \frac{L}{2} \|\hat{\mathbf{x}} - \mathbf{x}^*\|^2 + L \|\hat{\mathbf{x}} - \mathbf{x}_0\| \cdot \|\hat{\mathbf{x}} - \mathbf{x}^*\|}_{\le 0}$$

$$\le \frac{L}{2} \|\mathbf{x}^* - \mathbf{x}_0\|^2.$$

$\square$

We then prove Lemma 5.

*Proof of Lemma 5.* This can be seen as a special case of Proposition 2.2 [4]. Define the gradient descent-ascent field to be $F(\mathbf{z}) := \begin{bmatrix} \nabla_{\mathbf{x}} f(\mathbf{x}, \mathbf{y}) \\ -\nabla_{\mathbf{y}} f(\mathbf{x}, \mathbf{y}) \end{bmatrix}$. Note that the $\hat{\mathbf{z}}$ can also be written as

$$\hat{\mathbf{z}} = \underset{\mathbf{z} \in \mathcal{X} \times \mathcal{Y}}{\arg\min} \left\{ L \|\mathbf{z} - \mathbf{z}_0\|^2 + F(\mathbf{z}_0)^T \mathbf{z} \right\}.$$

Now, define $\mathbf{z}' = (\mathbf{x}', \mathbf{y}')$ to be

$$\mathbf{x}' \leftarrow P_{\mathcal{X}} \left[ \mathbf{x}_0 - \frac{1}{2L} \nabla_{\mathbf{x}} f(\hat{\mathbf{x}}, \hat{\mathbf{y}}) \right],$$

$$\mathbf{y}' \leftarrow P_{\mathcal{Y}} \left[ \mathbf{y}_0 + \frac{1}{2L} \nabla_{\mathbf{y}} f(\hat{\mathbf{x}}, \hat{\mathbf{y}}) \right].$$

In other words, $\mathbf{z}' = \arg\min_{\mathbf{z} \in \mathcal{X} \times \mathcal{Y}} \left\{ L \|\mathbf{z} - \mathbf{z}_0\|^2 + F(\hat{\mathbf{z}})^T \mathbf{z} \right\}$. By the optimality condition and $2L$-strong convexity of $L \|\mathbf{z} - \mathbf{z}_0\|^2 + F(\hat{\mathbf{z}})^T \mathbf{z}$, for any $\mathbf{z} \in \mathcal{X} \times \mathcal{Y}$,

$$L \|\mathbf{z}' - \mathbf{z}_0\|^2 + F(\hat{\mathbf{z}})^T \mathbf{z}' + L \|\mathbf{z}' - \mathbf{z}\|^2 \le L \|\mathbf{z} - \mathbf{z}_0\|^2 + F(\hat{\mathbf{z}})^T \mathbf{z}.$$

Similarly, by optimality of $\hat{\mathbf{z}}$,

$$L \|\hat{\mathbf{z}} - \mathbf{z}_0\|^2 + F(\mathbf{z}_0)^T \hat{\mathbf{z}} + L \|\mathbf{z}' - \hat{\mathbf{z}}\|^2 \le L \|\mathbf{z}' - \mathbf{z}_0\|^2 + F(\mathbf{z}_0)^T \mathbf{z}'.$$

Thus

$$F(\hat{\mathbf{z}})^T (\hat{\mathbf{z}} - \mathbf{z}) = F(\hat{\mathbf{z}})^T (\mathbf{z}' - \mathbf{z}) + F(\hat{\mathbf{z}})^T (\hat{\mathbf{z}} - \mathbf{z}')$$

$$= F(\hat{\mathbf{z}})^T (\mathbf{z}' - \mathbf{z}) + F(\mathbf{z}_0)^T (\hat{\mathbf{z}} - \mathbf{z}') + (F(\hat{\mathbf{z}}) - F(\mathbf{z}_0))^T (\hat{\mathbf{z}} - \mathbf{z}')$$

$$\le L \left( \|\mathbf{z} - \mathbf{z}_0\|^2 - \|\mathbf{z}' - \mathbf{z}_0\|^2 - \|\mathbf{z}' - \mathbf{z}\|^2 \right) + (F(\hat{\mathbf{z}}) - F(\mathbf{z}_0))^T (\hat{\mathbf{z}} - \mathbf{z}')$$

$$\quad + L \left( \|\mathbf{z}' - \mathbf{z}_0\|^2 - \|\hat{\mathbf{z}} - \mathbf{z}_0\|^2 - \|\mathbf{z}' - \hat{\mathbf{z}}\|^2 \right)$$

$$\le L \left( \|\mathbf{z} - \mathbf{z}_0\|^2 - \|\mathbf{z}' - \mathbf{z}\|^2 \right) + 2L \|\hat{\mathbf{z}} - \mathbf{z}_0\| \cdot \|\hat{\mathbf{z}} - \mathbf{z}'\| - L \|\hat{\mathbf{z}} - \mathbf{z}'\|^2 - L \|\hat{\mathbf{z}} - \mathbf{z}_0\|^2$$

$$\le L \left( \|\mathbf{z} - \mathbf{z}_0\|^2 - \|\mathbf{z}' - \mathbf{z}\|^2 \right).$$

Here we used the fact that for any $\mathbf{z}_1, \mathbf{z}_2$, $\|F(\mathbf{z}_1) - F(\mathbf{z}_2)\| \le 2L \|\mathbf{z}_1 - \mathbf{z}_2\|$. Note that (by convexity and concavity)

$$F(\hat{\mathbf{z}})^T (\hat{\mathbf{z}} - \mathbf{z}) = \nabla_{\mathbf{x}} f(\hat{\mathbf{x}}, \hat{\mathbf{y}})^T (\hat{\mathbf{x}} - \mathbf{x}) - \nabla_{\mathbf{y}} f(\hat{\mathbf{x}}, \hat{\mathbf{y}})^T (\hat{\mathbf{y}} - \mathbf{y})$$

$$\ge [f(\hat{\mathbf{x}}, \hat{\mathbf{y}}) - f(\mathbf{x}, \hat{\mathbf{y}})] + [f(\hat{\mathbf{x}}, \mathbf{y}) - f(\hat{\mathbf{x}}, \hat{\mathbf{y}})]$$

$$\ge f(\hat{\mathbf{x}}, \mathbf{y}) - f(\mathbf{x}, \hat{\mathbf{y}}).$$

If we choose $\mathbf{x}$ and $\mathbf{y}$ to be $\mathbf{x}^*(\hat{\mathbf{y}})$ and $\mathbf{y}^*(\hat{\mathbf{x}})$, we can see that

$$\max_{\mathbf{y} \in \mathcal{Y}} f(\hat{\mathbf{x}}, \mathbf{y}) - \min_{\mathbf{x} \in \mathcal{X}} f(\mathbf{x}, \hat{\mathbf{y}}) \le L \|\mathbf{z} - \mathbf{z}_0\|^2$$

$$\le 2L \|\mathbf{z} - \mathbf{z}^*\|^2 + 2L \|\mathbf{z}^* - \mathbf{z}_0\|^2$$

$$\le 2L \|\mathbf{x}^*(\hat{\mathbf{y}}) - \mathbf{x}^*\|^2 + 2L \|\mathbf{y}^*(\hat{\mathbf{x}}) - \mathbf{y}^*\|^2 + 2L \|\mathbf{z}^* - \mathbf{z}_0\|^2$$

$$\le \frac{2 L_{\mathbf{xy}}^2}{\min\{m_{\mathbf{x}}, m_{\mathbf{y}}\}^2} \cdot L \|\hat{\mathbf{z}} - \mathbf{z}^*\|^2 + 2L \|\mathbf{z}^* - \mathbf{z}_0\|^2.$$

By Corollary 2.2.1 [6], $(\mathbf{x}_0 - \hat{\mathbf{x}})^T (\mathbf{x}_0 - \mathbf{x}^*) \geq \frac{1}{2}\|\hat{\mathbf{x}} - \mathbf{x}_0\|^2$. Therefore

$$\|\hat{\mathbf{x}} - \mathbf{x}^*\|^2 = \|(\mathbf{x}_0 - \mathbf{x}^*) + (\hat{\mathbf{x}} - \mathbf{x}_0)\|^2$$
$$= \|\mathbf{x}_0 - \mathbf{x}^*\|^2 + 2(\mathbf{x}_0 - \mathbf{x}^*)^T(\hat{\mathbf{x}} - \mathbf{x}_0) + \|\hat{\mathbf{x}} - \mathbf{x}_0\|^2$$
$$\leq \|\mathbf{x}_0 - \mathbf{x}^*\|^2.$$

Similarly, $\|\hat{\mathbf{y}} - \mathbf{y}^*\| \leq \|\mathbf{y}_0 - \mathbf{y}^*\|$. Thus

$$\|\hat{\mathbf{z}} - \mathbf{z}^*\|^2 = \|\hat{\mathbf{x}} - \mathbf{x}^*\|^2 + \|\hat{\mathbf{y}} - \mathbf{y}^*\|^2 \leq \|\mathbf{z}_0 - \mathbf{z}^*\|^2 \leq \epsilon^2.$$

It follows that

$$\max_{\mathbf{y} \in \mathcal{Y}} f(\hat{\mathbf{x}}, \mathbf{y}) - \min_{\mathbf{x} \in \mathcal{X}} f(\mathbf{x}, \hat{\mathbf{y}}) \leq 2L \cdot \left( \frac{L_{\mathbf{xy}}^2}{\min\{m_{\mathbf{x}}, m_{\mathbf{y}}\}^2} + 1 \right) \epsilon^2.$$

$\square$

## 6    Implications of Theorem 3

In this section, we discuss how Theorem 3 implies improved bounds for strongly convex-concave problems and convex-concave problems via reductions established in [3].

Let us consider minimax optimization problem $\min_{\mathbf{x} \in \mathcal{X}} \max_{\mathbf{y} \in \mathcal{Y}} f(\mathbf{x}, \mathbf{y})$, where $f(\mathbf{x}, \mathbf{y})$ is $m_{\mathbf{x}}$-strongly convex with respect to $\mathbf{x}$, concave with respect to $\mathbf{y}$, and $(L_{\mathbf{x}}, L_{\mathbf{xy}}, L_{\mathbf{y}})$-smooth. Here, we assume that $\mathcal{X}$ and $\mathcal{Y}$ are bounded sets, with diameters $D_{\mathbf{x}} = \max_{\mathbf{x}, \mathbf{x}' \in \mathcal{X}} \|\mathbf{x} - \mathbf{x}'\|$ and $D_{\mathbf{y}} = \max_{\mathbf{y}, \mathbf{y}' \in \mathcal{Y}} \|\mathbf{y} - \mathbf{y}'\|$.

Following [3], let us consider the function

$$f_{\epsilon, \mathbf{y}}(\mathbf{x}, \mathbf{y}) := f(\mathbf{x}, \mathbf{y}) - \frac{\epsilon\|\mathbf{y} - \mathbf{y}_0\|^2}{2D_{\mathbf{y}}^2}.$$

Recall that $(\hat{\mathbf{x}}, \hat{\mathbf{y}})$ is an $\epsilon$-saddle point of $f$ if $\max_{\mathbf{y} \in \mathcal{Y}} f(\hat{\mathbf{x}}, \mathbf{y}) - \min_{\mathbf{x} \in \mathcal{X}} f(\mathbf{x}, \hat{\mathbf{y}}) \leq \epsilon$. We now show that a $(\epsilon/2)$-saddle point of $f_{\epsilon, \mathbf{y}}$ would be an $\epsilon$-saddle point of $f$. Let $\mathbf{x}^*(\cdot) := \arg\min_{\mathbf{x} \in \mathcal{X}} f(\mathbf{x}, \cdot)$ and $\mathbf{y}^*(\cdot) := \arg\max_{\mathbf{y} \in \mathcal{Y}} f(\cdot, \mathbf{y})$. Obviously, for any $\mathbf{x} \in \mathcal{X}, \mathbf{y} \in \mathcal{Y}$,

$$f(\mathbf{x}, \mathbf{y}) - \frac{\epsilon}{2} \leq f_{\epsilon, \mathbf{y}}(\mathbf{x}, \mathbf{y}) \leq f(\mathbf{x}, \mathbf{y}).$$

Thus, if $(\hat{\mathbf{x}}, \hat{\mathbf{y}})$ is a $(\epsilon/2)$-saddle point of $f_{\epsilon, \mathbf{y}}$, then

$$f(\hat{\mathbf{x}}, \mathbf{y}^*(\hat{\mathbf{x}})) \leq f_{\epsilon, \mathbf{y}}(\hat{\mathbf{x}}, \mathbf{y}^*(\hat{\mathbf{x}})) + \frac{\epsilon}{2} \leq \max_{\mathbf{y} \in \mathcal{Y}} f_{\epsilon, \mathbf{y}}(\hat{\mathbf{x}}, \mathbf{y}) + \frac{\epsilon}{2},$$
$$f(\mathbf{x}^*(\hat{\mathbf{y}}), \hat{\mathbf{y}}) \geq f_{\epsilon, \mathbf{y}}(\mathbf{x}^*(\hat{\mathbf{y}}), \hat{\mathbf{y}}) \geq \min_{\mathbf{x} \in \mathcal{X}} f_{\epsilon, \mathbf{y}}(\mathbf{x}, \hat{\mathbf{y}}).$$

It immediately follows that

$$\max_{\mathbf{y} \in \mathcal{Y}} f(\hat{\mathbf{x}}, \mathbf{y}) - \min_{\mathbf{x} \in \mathcal{X}} f(\mathbf{x}, \hat{\mathbf{y}}) \leq \frac{\epsilon}{2} + \max_{\mathbf{y} \in \mathcal{Y}} f_{\epsilon, \mathbf{y}}(\hat{\mathbf{x}}, \mathbf{y}) - \min_{\mathbf{x} \in \mathcal{X}} f_{\epsilon, \mathbf{y}}(\mathbf{x}, \hat{\mathbf{y}}) \leq \epsilon.$$

Thus, to find an $\epsilon$-saddle point of $f$, we only need to find an $(\epsilon/2)$-saddle point of $f_{\epsilon, \mathbf{y}}$. We can now prove Corollary 1 by reducing to (the constrained version of) Theorem 3.

Observe that $f_{\epsilon, \mathbf{y}}$ belongs to $\mathcal{F}(m_{\mathbf{x}}, \frac{\epsilon}{D_{\mathbf{y}}^2}, L_{\mathbf{x}}, L_{\mathbf{xy}}, L_{\mathbf{y}} + \frac{\epsilon}{D_{\mathbf{y}}^2})$. Thus, by Theorem 3, the gradient complexity of finding a $(\epsilon/2)$-saddle point in $f_{\epsilon, \mathbf{y}}$ is [2]

$$O\left( \sqrt{\frac{L_{\mathbf{x}}}{m_{\mathbf{x}}} + \left( \frac{L \cdot L_{\mathbf{xy}}}{m_{\mathbf{x}}} + L_{\mathbf{y}} \right) \cdot \frac{D_{\mathbf{y}}^2}{\epsilon}} \cdot \ln^4 \left( \frac{(D_{\mathbf{x}} + D_{\mathbf{y}})^2 L^2}{m_{\mathbf{x}}\epsilon} \right) \right) = \tilde{O}\left( \sqrt{\frac{m_{\mathbf{x}} \cdot L_{\mathbf{y}} + L \cdot L_{\mathbf{xy}}}{m_{\mathbf{x}}\epsilon}} \right),$$

which proves Corollary 1.

**Corollary 1.** *If $f(\mathbf{x}, \mathbf{y})$ is $(L_{\mathbf{x}}, L_{\mathbf{xy}}, L_{\mathbf{y}})$-smooth and $m_{\mathbf{x}}$-strongly convex w.r.t. $\mathbf{x}$, via reduction to Theorem 3, the gradient complexity of finding an $\epsilon$-saddle point is $\tilde{O}\left( \sqrt{\frac{m_{\mathbf{x}} \cdot L_{\mathbf{y}} + L \cdot L_{\mathbf{xy}}}{m_{\mathbf{x}}\epsilon}} \right)$.*

In comparison, Lin et al.'s result in this setting is $\tilde{O}\left(\sqrt{\frac{L^2}{m_{\mathbf{x}}\epsilon}}\right)$. Meanwhile a lower bound for this problem has been shown to be $\Omega\left(\sqrt{\frac{L_{\mathbf{xy}}^2}{m_{\mathbf{x}}\epsilon}}\right)$ [8]. It can be seen that when $L_{\mathbf{xy}} \ll L$, our bound is a significant improvement over Lin et al.'s result, as $m_{\mathbf{x}} \cdot L_{\mathbf{y}} + L \cdot L_{\mathbf{xy}} \ll L^2$.

Similarly, if $f : \mathcal{X} \times \mathcal{Y} \to \mathbb{R}$ is convex with respect to $\mathbf{x}$, concave with respect to $\mathbf{y}$ and $(L_{\mathbf{x}}, L_{\mathbf{xy}}, L_{\mathbf{y}})$-smooth, we can consider the function

$$f_\epsilon(\mathbf{x}, \mathbf{y}) := f(\mathbf{x}, \mathbf{y}) + \frac{\epsilon \|\mathbf{x} - \mathbf{x}_0\|^2}{4D_{\mathbf{x}}^2} - \frac{\epsilon \|\mathbf{y} - \mathbf{y}_0\|^2}{4D_{\mathbf{y}}^2}.$$

It can be shown that for any $\hat{\mathbf{x}} \in \mathcal{X}$,

$$\max_{\mathbf{y} \in \mathcal{Y}} \left\{ f(\hat{\mathbf{x}}, \mathbf{y}) + \frac{\epsilon \|\hat{\mathbf{x}} - \mathbf{x}_0\|^2}{4D_{\mathbf{x}}^2} - \frac{\epsilon \|\mathbf{y} - \mathbf{y}_0\|^2}{4D_{\mathbf{y}}^2} \right\} \geq \max_{\mathbf{y} \in \mathcal{Y}} f(\hat{\mathbf{x}}, \mathbf{y}) - \frac{\epsilon}{4}.$$

Similarly, for any $\hat{\mathbf{y}} \in \mathcal{Y}$,

$$\min_{\mathbf{x} \in \mathcal{X}} \left\{ f(\mathbf{x}, \hat{\mathbf{y}}) + \frac{\epsilon \|\mathbf{x} - \mathbf{x}_0\|^2}{4D_{\mathbf{x}}^2} - \frac{\epsilon \|\hat{\mathbf{y}} - \mathbf{y}_0\|^2}{4D_{\mathbf{y}}^2} \right\} \leq \min_{\mathbf{x} \in \mathcal{Y}} f(\mathbf{x}, \hat{\mathbf{y}}) + \frac{\epsilon}{4}.$$

Therefore, if $(\hat{\mathbf{x}}, \hat{\mathbf{y}})$ is an $(\epsilon/2)$-saddle point of $f_\epsilon$, it is an $\epsilon$-saddle point of $f$, as

$$\max_{\mathbf{y} \in \mathcal{Y}} f(\hat{\mathbf{x}}, \mathbf{y}) - \min_{\mathbf{x} \in \mathcal{X}} f(\mathbf{x}, \hat{\mathbf{y}}) \leq \frac{\epsilon}{2} + \max_{\mathbf{y} \in \mathcal{Y}} f_\epsilon(\hat{\mathbf{x}}, \mathbf{y}) - \min_{\mathbf{x} \in \mathcal{X}} f_\epsilon(\mathbf{x}, \hat{\mathbf{y}}) \leq \epsilon.$$

Observe that $f_\epsilon$ belongs to $\mathcal{F}(\frac{\epsilon}{2D_{\mathbf{x}}^2}, \frac{\epsilon}{2D_{\mathbf{y}}^2}, L_{\mathbf{x}} + \frac{\epsilon}{2D_{\mathbf{x}}^2}, L_{\mathbf{xy}}, L_{\mathbf{y}} + \frac{\epsilon}{2D_{\mathbf{y}}^2})$. Thus, by Theorem 3, the gradient complexity of finding an $(\epsilon/2)$-saddle point of $f_\epsilon$ is

$$O\left(\left(\sqrt{\frac{L_{\mathbf{x}}D_{\mathbf{x}}^2 + L_{\mathbf{y}}D_{\mathbf{y}}^2}{\epsilon}} + \frac{D_{\mathbf{x}}D_{\mathbf{y}}\sqrt{L \cdot L_{\mathbf{xy}}}}{\epsilon}\right) \cdot \ln^4\left(\frac{L(D_{\mathbf{x}} + D_{\mathbf{y}})^2}{\epsilon}\right)\right),$$

which proves Corollary 2.

**Corollary 2.** *If $f(\mathbf{x}, \mathbf{y})$ is $(L_{\mathbf{x}}, L_{\mathbf{xy}}, L_{\mathbf{y}})$-smooth and convex-concave, via reduction to Theorem 3, the gradient complexity to produce an $\epsilon$-saddle point is $\tilde{O}\left(\sqrt{\frac{L_{\mathbf{x}} + L_{\mathbf{y}}}{\epsilon}} + \frac{\sqrt{L \cdot L_{\mathbf{xy}}}}{\epsilon}\right)$.*

In comparison, Lin et al.'s result for this setting is $\tilde{O}\left(\frac{L}{\epsilon}\right)$, and the classic result for ExtraGradient is $O\left(\frac{L}{\epsilon}\right)$ [4]. Meanwhile, a lower bound for this setting has shown to be $\Omega\left(\sqrt{\frac{L_{\mathbf{x}}}{\epsilon}} + \frac{L_{\mathbf{xy}}}{\epsilon}\right)$ [7]. Again, our result can be a significant improvement over Lin et al.'s result if $L_{\mathbf{xy}} \ll L$, and is closer to the lower bound.

## 7 Proof of Theorem 4

The details of RHSS($k$) can be found in Algorithm 5. We will start by proving several useful lemmas.

**Lemma 1.** ([2]) *Define $M(\eta) := (\eta\mathbf{P} + \mathbf{S})^{-1}(\eta\mathbf{P} - \mathbf{G})(\eta\mathbf{P} + \mathbf{G})^{-1}(\eta\mathbf{P} - \mathbf{S})$. Then*

$$\rho(\mathbf{M}(\eta)) \leq \|\mathbf{M}(\eta)\|_2 \leq \max_{\lambda_i \in sp(\mathbf{P}^{-1}\mathbf{G})} \left|\frac{\lambda_i - \eta}{\lambda_i + \eta}\right| < 1.$$

*Proof of Lemma 1.* We provide a proof for completeness. First, observe that

$$\mathbf{M}(\eta) = (\eta\mathbf{P} + \mathbf{S})^{-1}(\eta\mathbf{P} - \mathbf{G})(\eta\mathbf{P} + \mathbf{G})^{-1}(\eta\mathbf{P} - \mathbf{S})$$
$$= \mathbf{P}^{-\frac{1}{2}}(\eta\mathbf{I} + \mathbf{P}^{-\frac{1}{2}}\mathbf{S}\mathbf{P}^{-\frac{1}{2}})^{-1}(\eta\mathbf{I} - \mathbf{P}^{-\frac{1}{2}}\mathbf{G}\mathbf{P}^{-\frac{1}{2}})(\eta\mathbf{I} + \mathbf{P}^{-\frac{1}{2}}\mathbf{G}\mathbf{P}^{-\frac{1}{2}})^{-1}(\eta\mathbf{I} - \mathbf{P}^{-\frac{1}{2}}\mathbf{S}\mathbf{P}^{-\frac{1}{2}})\mathbf{P}^{\frac{1}{2}}.$$

Let $\hat{\mathbf{G}} := \mathbf{P}^{-\frac{1}{2}}\mathbf{G}\mathbf{P}^{-\frac{1}{2}}, \hat{\mathbf{S}} := \mathbf{P}^{-\frac{1}{2}}\mathbf{S}\mathbf{P}^{-\frac{1}{2}}$. Then $\mathbf{M}(\eta)$ is similar to

$$(\eta\mathbf{I} + \hat{\mathbf{S}})^{-1}(\eta\mathbf{I} - \hat{\mathbf{G}})(\eta\mathbf{I} + \hat{\mathbf{G}})^{-1}(\eta\mathbf{I} - \hat{\mathbf{S}}),$$

---
**Algorithm 5** RHSS($k$) (Recursive Hermitian-skew-Hermitian Split)
---
**Require:** Initial point $[\mathbf{x}_0; \mathbf{y}_0]$, precision $\epsilon$, parameters $m_\mathbf{x}, m_\mathbf{y}, L_\mathbf{xy}$

$t \leftarrow 0$, $M_1 \leftarrow \frac{192 L^5}{m_\mathbf{x}^2 m_\mathbf{y}^3}$, $M_2 \leftarrow \frac{16 L_\mathbf{xy}}{m_\mathbf{y}}$, $\alpha \leftarrow \frac{m_\mathbf{x}}{m_\mathbf{y}}$, $\beta \leftarrow L_\mathbf{xy}^{-\frac{2}{k}} m_\mathbf{y}^{-\frac{k-2}{k}}$, $\eta \leftarrow L_\mathbf{xy}^{\frac{1}{k}} m_\mathbf{y}^{1-\frac{1}{k}}$, $\tilde{\epsilon} \leftarrow \frac{m_\mathbf{x}\epsilon}{L_\mathbf{xy}+L_\mathbf{x}}$

**repeat**

$$\begin{bmatrix} \mathbf{r}_1 \\ \mathbf{r}_2 \end{bmatrix} \leftarrow \begin{bmatrix} \eta\left(\alpha\mathbf{I} + \beta\mathbf{A}\right) & -\mathbf{B} \\ \mathbf{B}^T & \eta\left(\mathbf{I} + \beta\mathbf{C}\right) \end{bmatrix} \begin{bmatrix} \mathbf{x}_t \\ \mathbf{y}_t \end{bmatrix} + \begin{bmatrix} -\mathbf{u} \\ \mathbf{v} \end{bmatrix}.$$

Call conjugate gradient to compute

$$\begin{bmatrix} \mathbf{x}_{t+1/2} \\ \mathbf{y}_{t+1/2} \end{bmatrix} \leftarrow \text{CG}\left( \begin{bmatrix} \eta\left(\alpha\mathbf{I} + \beta\mathbf{A}\right) + \mathbf{A} & \\ & \eta\left(\mathbf{I} + \beta\mathbf{C}\right) + \mathbf{C} \end{bmatrix}, \begin{bmatrix} \mathbf{r}_1 \\ \mathbf{r}_2 \end{bmatrix}, \begin{bmatrix} \mathbf{x}_t \\ \mathbf{y}_t \end{bmatrix}, \frac{1}{M_1} \right).$$

Compute

$$\begin{bmatrix} \mathbf{w}_1 \\ \mathbf{w}_2 \end{bmatrix} \leftarrow \begin{bmatrix} \eta\alpha\mathbf{I} + \eta\beta\mathbf{A} - \mathbf{A} & 0 \\ 0 & \eta\left(\mathbf{I} + \beta\mathbf{C}\right) - \mathbf{C} \end{bmatrix} \begin{bmatrix} \mathbf{x}_{t+1/2} \\ \mathbf{y}_{t+1/2} \end{bmatrix} + \begin{bmatrix} -\mathbf{u} \\ \mathbf{v} \end{bmatrix}$$

Call RHSS($k-1$) with initial point $[\mathbf{x}_t; \mathbf{y}_t]$ and precision $1/M_2$ to solve

$$\begin{bmatrix} \mathbf{x}_{t+1} \\ \mathbf{y}_{t+1} \end{bmatrix} \leftarrow \begin{bmatrix} \eta\left(\alpha\mathbf{I} + \beta\mathbf{A}\right) & \mathbf{B} \\ -\mathbf{B}^T & \eta\left(\mathbf{I} + \beta\mathbf{C}\right) \end{bmatrix}^{-1} \begin{bmatrix} \mathbf{w}_1 \\ \mathbf{w}_2 \end{bmatrix}.$$

$\quad t \leftarrow t + 1$

**until** $\|\mathbf{J}\mathbf{z}_t - \mathbf{b}\| \leq \tilde{\epsilon}\|\mathbf{J}\mathbf{z}_0 - \mathbf{b}\|$

---

---
**Algorithm 6** The Conjugate Gradient Algorithm: CG($\mathbf{A}$,$\mathbf{b}$,$\mathbf{x}_0$,$\epsilon$) [1]
---
$\quad \mathbf{r}_0 \leftarrow \mathbf{b} - \mathbf{A}\mathbf{x}_0$, $\mathbf{p}_0 \leftarrow \mathbf{r}_0$, $k \leftarrow 0$

**repeat**

$\quad \alpha_k \leftarrow \frac{\mathbf{r}_k^T \mathbf{r}_k}{\mathbf{p}_k^T \mathbf{A}\mathbf{p}_k}$

$\quad \mathbf{x}_{k+1} \leftarrow \mathbf{x}_k + \alpha_k \mathbf{p}_k$

$\quad \mathbf{r}_{k+1} \leftarrow \mathbf{r}_k - \alpha_k \mathbf{A}\mathbf{p}_k$

$\quad \beta_k \leftarrow \frac{\mathbf{r}_{k+1}^T \mathbf{r}_{k+1}}{\mathbf{r}_k^T \mathbf{r}_k}$

$\quad \mathbf{p}_{k+1} \leftarrow \mathbf{r}_{k+1} + \beta_k \mathbf{p}_k$

$\quad k \leftarrow k + 1$

**until** $\|\mathbf{r}_k\| \leq \epsilon\|\mathbf{b} - \mathbf{A}\mathbf{x}_0\|$

Return $\mathbf{x}$

---

which is then similar to

$$(\eta\mathbf{I} - \hat{\mathbf{G}})(\eta\mathbf{I} + \hat{\mathbf{G}})^{-1}(\eta\mathbf{I} - \hat{\mathbf{S}})(\eta\mathbf{I} + \hat{\mathbf{S}})^{-1}.$$

The key observation is that $(\eta\mathbf{I} - \hat{\mathbf{S}})(\eta\mathbf{I} + \hat{\mathbf{S}})^{-1}$ is orthogonal, since

$$\left((\eta\mathbf{I} + \hat{\mathbf{S}})^{-1}\right)^T (\eta\mathbf{I} - \hat{\mathbf{S}})^T(\eta\mathbf{I} - \hat{\mathbf{S}})(\eta\mathbf{I} + \hat{\mathbf{S}})^{-1}$$
$$= (\eta\mathbf{I} - \hat{\mathbf{S}})^{-1}(\eta\mathbf{I} + \hat{\mathbf{S}})(\eta\mathbf{I} - \hat{\mathbf{S}})(\eta\mathbf{I} + \hat{\mathbf{S}})^{-1}$$
$$= (\eta\mathbf{I} - \hat{\mathbf{S}})^{-1}(\eta\mathbf{I} - \hat{\mathbf{S}})(\eta\mathbf{I} + \hat{\mathbf{S}})(\eta\mathbf{I} + \hat{\mathbf{S}})^{-1} = \mathbf{I}.$$

Therefore

$$\rho(\mathbf{M}(\eta)) \leq \|(\eta\mathbf{I} - \hat{\mathbf{G}})(\eta\mathbf{I} + \hat{\mathbf{G}})^{-1}(\eta\mathbf{I} - \hat{\mathbf{S}})(\eta\mathbf{I} + \hat{\mathbf{S}})^{-1}\|_2$$
$$\leq \|(\eta\mathbf{I} - \hat{\mathbf{G}})(\eta\mathbf{I} + \hat{\mathbf{G}})^{-1}\|_2 \cdot \|(\eta\mathbf{I} - \hat{\mathbf{S}})(\eta\mathbf{I} + \hat{\mathbf{S}})^{-1}\|_2$$
$$= \|(\eta\mathbf{I} - \hat{\mathbf{G}})(\eta\mathbf{I} + \hat{\mathbf{G}})^{-1}\|_2$$
$$= \max_{\lambda_i \in sp(\hat{\mathbf{G}})} \left|\frac{\lambda_i - \eta}{\lambda_i + \eta}\right| = \max_{\lambda_i \in sp(\mathbf{P}^{-1}\mathbf{G})} \left|\frac{\lambda_i - \eta}{\lambda_i + \eta}\right|.$$

$\square$

We now proceed to state some useful lemmas for the proof of Theorem 4.

**Lemma 7.** *The following statements about the eigenvalues and singular values of matrices hold:*

1. *The singular values of $\mathbf{J}$ fall in $[m_{\mathbf{x}}, L_{\mathbf{xy}} + L_{\mathbf{x}}]$;*

2. *The condition number of $\eta\mathbf{P} + \mathbf{G}$ is at most $\frac{3L_{\mathbf{x}}}{m_{\mathbf{x}}}\left(\frac{m_{\mathbf{y}}}{L_{\mathbf{xy}}}\right)^{\frac{1}{k}}$;*

3. *The condition number of $\eta\mathbf{P} + \mathbf{G}$ is at most $L_{\mathbf{x}}/m_{\mathbf{x}}$.*

4. *The eigenvalues of $\eta(\alpha\mathbf{I} + \beta\mathbf{A})$ fall in $[\eta\alpha, 2\eta\beta L_{\mathbf{x}}]$. The eigenvalues of $\eta(\mathbf{I} + \beta\mathbf{C})$ fall in $[\eta, 2\eta\beta L_{\mathbf{x}}]$.*

*Proof of Lemma 7.* 1. Consider an arbitrary $\mathbf{x} \in \mathbb{R}^{n+m}$ with $\|\mathbf{x}\|_2 = 1$. Construct a set of orthonormal vectors $\{\mathbf{x}_1, \cdots, \mathbf{x}_{n+m}\}$ with $\mathbf{x}_1 = \mathbf{x}$. Then

$$\mathbf{x}^T\mathbf{J}^T\mathbf{J}\mathbf{x} = \sum_{i=1}^{n+m} \mathbf{x}^T\mathbf{J}^T\mathbf{x}_i\mathbf{x}_i^T\mathbf{J}\mathbf{x} = \sum_{i=1}^{n+m}\left(\mathbf{x}^T\mathbf{J}^T\mathbf{x}_i\right)^2 \geq \left(\mathbf{x}^T\mathbf{J}^T\mathbf{x}\right)^2.$$

Since $\mathbf{J} = \mathbf{G} + \mathbf{S} = \begin{bmatrix} \mathbf{A} & 0 \\ 0 & \mathbf{C} \end{bmatrix} + \begin{bmatrix} 0 & \mathbf{B} \\ -\mathbf{B}^T & 0 \end{bmatrix}$, where $\mathbf{S}$ is skew-symmetric, $\mathbf{x}^T\mathbf{J}^T\mathbf{x} = \mathbf{x}^T\mathbf{G}\mathbf{x} \geq m_{\mathbf{x}}$. Thus

$$\sigma_{min}(\mathbf{J}) = \sqrt{\lambda_{\min}\left(\mathbf{J}^T\mathbf{J}\right)} \geq m_{\mathbf{x}}.$$

Meanwhile,

$$\lambda_{\max}\left(\mathbf{J}^T\mathbf{J}\right) \leq \|\mathbf{G}\|_2 + \|\mathbf{S}\|_2 \leq L_{\mathbf{xy}} + L_{\mathbf{x}}.$$

2. Note that

$$\eta\mathbf{P} + \mathbf{G} = \begin{bmatrix} \eta(\alpha\mathbf{I} + \beta\mathbf{A}) + \mathbf{A} & \\ & \eta(\mathbf{I} + \beta\mathbf{C}) + \mathbf{C} \end{bmatrix}.$$

Thus

$$\|\eta\mathbf{P} + \mathbf{G}\|_2 \leq \max\{\eta\left(\alpha + \beta L_{\mathbf{x}}\right) + L_{\mathbf{x}}, \eta\left(1 + \beta L_{\mathbf{x}}\right) + L_{\mathbf{x}}\} = \eta\left(1 + \beta L_{\mathbf{x}}\right) + L_{\mathbf{x}}.$$

On the other hand

$$\lambda_{\min}\left(\eta\mathbf{P} + \mathbf{G}\right) \geq \min\{\eta\alpha + \eta\beta m_{\mathbf{x}} + m_{\mathbf{x}}, \eta + \eta\beta m_{\mathbf{y}} + m_{\mathbf{y}}\} = \eta\alpha + \eta\beta m_{\mathbf{x}} + m_{\mathbf{x}}.$$

Thus the condition number of $\eta\mathbf{P} + \mathbf{G}$ is at most

$$\frac{\eta\left(1 + \beta L_{\mathbf{x}}\right) + L_{\mathbf{x}}}{\eta\alpha + \eta\beta m_{\mathbf{x}} + m_{\mathbf{x}}} \leq \frac{L_{\mathbf{x}}}{\eta\alpha} + \frac{1 + \beta L_{\mathbf{x}}}{\alpha + \beta m_{\mathbf{x}}} \leq \frac{L_{\mathbf{x}}}{\eta\alpha} + \frac{2\beta L_{\mathbf{x}}}{\alpha} = \frac{L_{\mathbf{x}}}{m_{\mathbf{x}}}\left(\frac{m_{\mathbf{y}}}{L_{\mathbf{xy}}}\right)^{\frac{1}{k}} + \frac{2L_{\mathbf{x}}}{m_{\mathbf{x}}}\left(\frac{m_{\mathbf{y}}}{L_{\mathbf{xy}}}\right)^{\frac{2}{k}} \leq \frac{3L_{\mathbf{x}}}{m_{\mathbf{x}}}\left(\frac{m_{\mathbf{y}}}{L_{\mathbf{xy}}}\right)^{\frac{1}{k}}.$$

3. On the other hand,

$$\frac{\eta\left(1 + \beta L_{\mathbf{x}}\right) + L_{\mathbf{x}}}{\eta\alpha + \eta\beta m_{\mathbf{x}} + m_{\mathbf{x}}} = \frac{\eta + \eta\beta L_{\mathbf{x}} + L_{\mathbf{x}}}{\eta\alpha + \eta\beta m_{\mathbf{x}} + m_{\mathbf{x}}} \leq \max\left\{\frac{1}{\alpha}, \frac{L_{\mathbf{x}}}{m_{\mathbf{x}}}\right\} = \frac{L_{\mathbf{x}}}{m_{\mathbf{x}}}.$$

4. Finally let us consider matrices $\eta(\alpha\mathbf{I} + \beta\mathbf{A})$ and $\eta(\mathbf{I} + \beta\mathbf{C})$. Obviously

$$\eta(\alpha\mathbf{I} + \beta\mathbf{A}) \succcurlyeq \eta\alpha\mathbf{I}, \quad \eta(\mathbf{I} + \beta\mathbf{C}) \succcurlyeq \eta\mathbf{I}.$$

Meanwhile

$$\begin{aligned} \|\eta(\alpha\mathbf{I} + \beta\mathbf{A})\| &\leq \eta \cdot (\alpha + \beta L_{\mathbf{x}}) \\ &\leq \eta\left(1 + \beta L_{\mathbf{x}}\right) && (\alpha < 1) \\ &\leq 2\eta\beta L_{\mathbf{x}}. && (\beta L_{\mathbf{x}} > 1) \end{aligned}$$

Similarly $\|\eta(\mathbf{I} + \beta\mathbf{C})\| \leq 2\eta\beta L_{\mathbf{x}}$. $\quad\square$

**Lemma 8.** *With our choice of $\eta$, $\alpha$ and $\beta$,*

$$\rho(\mathbf{M}(\eta)) \leq \|\mathbf{M}(\eta)\|_2 \leq 1 - \frac{1}{2}\left(\frac{m_{\mathbf{y}}}{L_{\mathbf{xy}}}\right)^{\frac{1}{k}}.$$

*Proof of Lemma 8.* By Lemma 1,

$$\rho(\mathbf{M}(\eta)) \leq \|\mathbf{M}(\eta)\|_2 \leq \max_{\lambda_i \in sp(\mathbf{P}^{-1}\mathbf{G})} \left| \frac{\lambda_i - \eta}{\lambda_i + \eta} \right|.$$

Observe that

$$\mathbf{P}^{-1}\mathbf{G} = \begin{bmatrix} (\alpha\mathbf{I} + \beta\mathbf{A})^{-1}\mathbf{A} & \\ & (\mathbf{I} + \beta\mathbf{C})^{-1}\mathbf{C} \end{bmatrix}.$$

The eigenvalues of $(\alpha\mathbf{I} + \beta\mathbf{A})^{-1}\mathbf{A}$ are contained in

$$\left[ \frac{m_{\mathbf{x}}}{\alpha + \beta m_{\mathbf{x}}}, \frac{L_{\mathbf{x}}}{\alpha + \beta L_{\mathbf{x}}} \right] \subseteq \left[ \frac{m_{\mathbf{y}}}{2}, \frac{1}{\beta} \right]. \qquad (\beta m_{\mathbf{x}} \leq \alpha)$$

Similarly the eigenvalues of $(\mathbf{I} + \beta\mathbf{C})^{-1}\mathbf{C}$ are contained in

$$\left[ \frac{m_{\mathbf{y}}}{1 + \beta m_{\mathbf{y}}}, \frac{L_{\mathbf{x}}}{1 + \beta L_{\mathbf{x}}} \right] \subseteq \left[ \frac{m_{\mathbf{y}}}{2}, \frac{1}{\beta} \right]. \qquad (\beta m_{\mathbf{y}} \leq 1)$$

Recall that $\eta = L_{\mathbf{xy}}^{1/k} m_{\mathbf{y}}^{1-1/k} = \sqrt{m_{\mathbf{y}}/\beta}$. As a result,

$$\max_{\lambda_i \in sp(\mathbf{P}^{-1}\mathbf{G})} \left| \frac{\lambda_i - \eta}{\lambda_i + \eta} \right| \leq \max \left\{ \frac{\frac{1}{\beta} - \sqrt{\frac{m_{\mathbf{y}}}{\beta}}}{\frac{1}{\beta} + \sqrt{\frac{m_{\mathbf{y}}}{\beta}}}, \frac{\sqrt{\frac{m_{\mathbf{y}}}{\beta}} - \frac{m_{\mathbf{y}}}{2}}{\sqrt{\frac{m_{\mathbf{y}}}{\beta}} + \frac{m_{\mathbf{y}}}{2}} \right\} \leq 1 - \frac{\sqrt{\beta m_{\mathbf{y}}}}{2} = 1 - \frac{1}{2}\left( \frac{m_{\mathbf{y}}}{L_{\mathbf{xy}}} \right)^{\frac{1}{k}}.$$

$\square$

**Lemma 9.** *When RHSS(k) terminates* $\|\mathbf{z}_t - \mathbf{z}^*\| \leq \epsilon\|\mathbf{z}_0 - \mathbf{z}^*\|$.

*Proof of Lemma 9.*

$$\sigma_{\min}(\mathbf{J})\|\mathbf{z}_t - \mathbf{z}^*\| \leq \|\mathbf{J}\mathbf{z}_t - \mathbf{b}\| \leq \tilde{\epsilon}\|\mathbf{J}\mathbf{z}_0 - \mathbf{b}\| \leq \sigma_{\max}(\mathbf{J})\|\mathbf{z}_0 - \mathbf{z}^*\|.$$

We know that $\sigma_{\min}(\mathbf{J}) \geq m_{\mathbf{x}}$ and that $\sigma_{\max}(\mathbf{J}) \leq L_{\mathbf{x}} + L_{\mathbf{xy}}$. Thus

$$\|\mathbf{z}_t - \mathbf{z}^*\| \leq \frac{\tilde{\epsilon} \cdot (L_{\mathbf{x}} + L_{\mathbf{xy}})}{m_{\mathbf{x}}}\|\mathbf{z}_0 - \mathbf{z}^*\| = \epsilon\|\mathbf{z}_0 - \mathbf{z}^*\|.$$

$\square$

**Lemma 10** (Proposition 9.5.1, [1]). *CG*$(\mathbf{A}, \mathbf{b}, \mathbf{x}_0, \epsilon)$ *returns (i.e. satisfies* $\|\mathbf{A}\mathbf{x}_T - \mathbf{b}\| \leq \epsilon\|\mathbf{A}\mathbf{x}_0 - \mathbf{b}\|$*) in at most* $\left\lceil \sqrt{\kappa}\ln\left( \frac{2\sqrt{\kappa}}{\epsilon} \right) \right\rceil$ *iterations.*

**Lemma 11.** *In RHSS(k),*

$$\|\mathbf{z}_{t+1} - \mathbf{z}^*\| \leq \left( 1 - \frac{1}{4}\left( \frac{m_{\mathbf{y}}}{L_{\mathbf{xy}}} \right)^{\frac{1}{k}} \right) \|\mathbf{z}_t - \mathbf{z}^*\|. \qquad (16)$$

*Proof of Lemma 11.* Let us define

$$\tilde{\mathbf{z}}_{t+1/2} = \begin{bmatrix} \tilde{\mathbf{x}}_{t+1/2} \\ \tilde{\mathbf{y}}_{t+1/2} \end{bmatrix} = \begin{bmatrix} \eta(\alpha\mathbf{I} + \beta\mathbf{A}) + \mathbf{A} & \\ & \eta(\mathbf{I} + \beta\mathbf{C}) + \mathbf{C} \end{bmatrix}^{-1} \begin{bmatrix} \mathbf{r}_1 \\ \mathbf{r}_2 \end{bmatrix}.$$

Since $\|(\eta\mathbf{P} + \mathbf{G})(\mathbf{z}_{t+1/2} - \tilde{\mathbf{z}}_{t+1/2})\| \leq \frac{1}{M_1}\|(\eta\mathbf{P} + \mathbf{G})(\mathbf{z}_t - \tilde{\mathbf{z}}_{t+1/2})\|$,

$$\begin{aligned}
\|\mathbf{z}_{t+1/2} - \tilde{\mathbf{z}}_{t+1/2}\| &\leq \frac{\|(\eta\mathbf{P} + \mathbf{G})(\mathbf{z}_{t+1/2} - \tilde{\mathbf{z}}_{t+1/2})\|}{\lambda_{\min}(\eta\mathbf{P} + \mathbf{G})} \leq \frac{\|(\eta\mathbf{P} + \mathbf{G})(\mathbf{z}_t - \tilde{\mathbf{z}}_{t+1/2})\|}{M_1\lambda_{\min}(\eta\mathbf{P} + \mathbf{G})} \\
&\leq \frac{\lambda_{\max}(\eta\mathbf{P} + \mathbf{G})}{M_1\lambda_{\min}(\eta\mathbf{P} + \mathbf{G})}\|\mathbf{z}_t - \tilde{\mathbf{z}}_{t+1/2}\| \\
&\leq \frac{L_{\mathbf{x}}}{M_1 m_{\mathbf{x}}}\|\mathbf{z}_t - \tilde{\mathbf{z}}_{t+1/2}\| = \frac{m_{\mathbf{x}} m_{\mathbf{y}}^3}{192 L^4}\|\mathbf{z}_t - \tilde{\mathbf{z}}_{t+1/2}\|.
\end{aligned} \qquad (17)$$

Because $\tilde{\mathbf{z}}_{t+1/2} - \mathbf{z}^* = (\eta\mathbf{P} + \mathbf{G})^{-1}(\eta\mathbf{P} - \mathbf{S})(\mathbf{z}_t - \mathbf{z}^*)$,

$$
\begin{aligned}
\|\tilde{\mathbf{z}}_{t+1/2} - \mathbf{z}^*\| &\leq \|(\eta\mathbf{P}+\mathbf{G})^{-1}\|_2 \|\eta\mathbf{P} - \mathbf{S}\|_2 \|\mathbf{z}_t - \mathbf{z}^*\| \\
&\leq \frac{1}{\eta\alpha} \cdot (L_{\mathbf{xy}} + \eta\alpha + \eta\beta L_{\mathbf{x}}) \cdot \|\mathbf{z}_t - \mathbf{z}^*\| \\
&\leq \left(1 + \frac{2L}{m_{\mathbf{x}}}\right)\|\mathbf{z}_t - \mathbf{z}^*\|.
\end{aligned}
$$

It follows that

$$
\|\mathbf{z}_t - \tilde{\mathbf{z}}_{t+1/2}\| \leq \|\mathbf{z}_t - \mathbf{z}^*\| + \|\tilde{\mathbf{z}}_{t+1/2} - \mathbf{z}^*\| \leq \left(2 + \frac{2L}{m_{\mathbf{x}}}\right)\|\mathbf{z}_t - \mathbf{z}^*\|.
$$

By plugging this into (17), one gets

$$
\|\mathbf{z}_{t+1/2} - \tilde{\mathbf{z}}_{t+1/2}\| \leq \frac{m_{\mathbf{x}} m_{\mathbf{y}}^3}{192 L^4} \cdot \left(2 + \frac{2L}{m_{\mathbf{x}}}\right)\|\mathbf{z}_t - \mathbf{z}^*\| \leq \frac{m_{\mathbf{y}}^3}{48 L^3}\|\mathbf{z}_t - \mathbf{z}^*\|. \tag{18}
$$

Now, let us define

$$
\begin{aligned}
\tilde{\mathbf{z}}_{t+1} &:= (\eta\mathbf{P} + \mathbf{S})^{-1}\left[(\eta\mathbf{P} - \mathbf{G})\tilde{\mathbf{z}}_{t+1/2} + \mathbf{b}\right], \\
\hat{\mathbf{z}}_{t+1} &:= (\eta\mathbf{P} + \mathbf{S})^{-1}\left[(\eta\mathbf{P} - \mathbf{G})\mathbf{z}_{t+1/2} + \mathbf{b}\right].
\end{aligned}
$$

First let us try to bound $\|\tilde{\mathbf{z}}_{t+1} - \hat{\mathbf{z}}_{t+1}\|$. Observe that $\hat{\mathbf{z}}_{t+1} - \mathbf{z}^* = (\eta\mathbf{P} + \mathbf{S})^{-1}(\eta\mathbf{P} - \mathbf{G})(\mathbf{z}_{t+1/2} - \mathbf{z}^*)$, so

$$
\begin{aligned}
\|\tilde{\mathbf{z}}_{t+1} - \hat{\mathbf{z}}_{t+1}\| &= \|(\tilde{\mathbf{z}}_{t+1} - \mathbf{z}^*) - (\hat{\mathbf{z}}_{t+1} - \mathbf{z}^*)\| \\
&= \|(\eta\mathbf{P} + \mathbf{G})^{-1}(\eta\mathbf{P} - \mathbf{S})(\tilde{\mathbf{z}}_{t+1/2} - \mathbf{z}_{t+1/2})\| \\
&\leq \|(\eta\mathbf{P} + \mathbf{G})^{-1}(\eta\mathbf{P} - \mathbf{S})\|_2 \cdot \|\tilde{\mathbf{z}}_{t+1/2} - \mathbf{z}_{t+1/2}\| \\
&\leq \frac{3L^2}{m_{\mathbf{y}}^2} \cdot \frac{m_{\mathbf{y}}^3}{48 L^3}\|\mathbf{z}_t - \mathbf{z}^*\| = \frac{m_{\mathbf{y}}}{16L}\|\mathbf{z}_t - \mathbf{z}^*\|. \tag{19}
\end{aligned}
$$

Next, by Lemma 9 on RHSS$(k-1)$,

$$
\|\mathbf{z}_{t+1} - \hat{\mathbf{z}}_{t+1}\| \leq \frac{1}{M_2}\|\mathbf{z}_t - \hat{\mathbf{z}}_{t+1}\| \leq \frac{1}{M_2}\left(\|\mathbf{z}_t - \mathbf{z}^*\| + \|\hat{\mathbf{z}}_{t+1} - \mathbf{z}^*\|\right).
$$

By Lemma 8,

$$
\|\tilde{\mathbf{z}}_{t+1} - \mathbf{z}^*\| = \|\mathbf{M}(\eta)(\mathbf{z}_t - \mathbf{z}^*)\|_2 \leq \left(1 - \frac{1}{2}\left(\frac{m_{\mathbf{y}}}{L_{\mathbf{xy}}}\right)^{\frac{1}{k}}\right)\|\mathbf{z}_t - \mathbf{z}^*\|. \tag{20}
$$

Thus

$$
\|\mathbf{z}_{t+1} - \hat{\mathbf{z}}_{t+1}\| \leq \frac{1}{M_2}\left(2\|\mathbf{z}_t - \mathbf{z}^*\| + \|\tilde{\mathbf{z}}_{t+1} - \hat{\mathbf{z}}_{t+1}\|\right). \tag{21}
$$

Combining (19) and (21), one gets

$$
\begin{aligned}
\|\mathbf{z}_{t+1} - \tilde{\mathbf{z}}_{t+1}\| &\leq \|\mathbf{z}_{t+1} - \hat{\mathbf{z}}_{t+1}\| + \|\hat{\mathbf{z}}_{t+1} - \tilde{\mathbf{z}}_{t+1}\| \\
&\leq \frac{2}{M_2}\|\mathbf{z}_t - \mathbf{z}^*\| + \left(1 + \frac{2}{M_2}\right)\|\hat{\mathbf{z}}_{t+1} - \tilde{\mathbf{z}}_{t+1}\| \\
&\leq \frac{m_{\mathbf{y}}}{8 L_{\mathbf{xy}}}\|\mathbf{z}_t - \mathbf{z}^*\| + \frac{m_{\mathbf{y}}}{8L}\|\mathbf{z}_t - \mathbf{z}^*\| \leq \frac{m_{\mathbf{y}}}{4 L_{\mathbf{xy}}}\|\mathbf{z}_t - \mathbf{z}^*\|.
\end{aligned}
$$

Combining this with (20), one gets

$$
\begin{aligned}
\|\mathbf{z}_{t+1} - \mathbf{z}^*\| &\leq \|\tilde{\mathbf{z}}_{t+1} - \mathbf{z}^*\| + \|\tilde{\mathbf{z}}_{t+1} - \mathbf{z}_{t+1}\| \\
&\leq \left(1 - \frac{1}{2}\left(\frac{m_{\mathbf{y}}}{L_{\mathbf{xy}}}\right)^{\frac{1}{k}}\right)\|\mathbf{z}_t - \mathbf{z}^*\| + \frac{m_{\mathbf{y}}}{4 L_{\mathbf{xy}}}\|\mathbf{z}_t - \mathbf{z}^*\| \\
&\leq \left(1 - \frac{1}{4}\left(\frac{m_{\mathbf{y}}}{L_{\mathbf{xy}}}\right)^{\frac{1}{k}}\right)\|\mathbf{z}_t - \mathbf{z}^*\|.
\end{aligned}
$$

$\square$

Finally, we are ready to prove Theorem 4.

**Theorem 4.** *There exists constants $C_1$, $C_2$, such that the number of matrix-vector products needed to find $(\mathbf{x}_T, \mathbf{y}_T)$ such that $\|\mathbf{z}_T - \mathbf{z}^*\| \leq \epsilon$ is at most*

$$\sqrt{\frac{L_{\mathbf{xy}}^2}{m_{\mathbf{x}} m_{\mathbf{y}}} + \left(\frac{L_{\mathbf{x}}}{m_{\mathbf{x}}} + \frac{L_{\mathbf{y}}}{m_{\mathbf{y}}}\right)\left(1 + \left(\frac{L_{\mathbf{xy}}}{\max\{m_{\mathbf{x}}, m_{\mathbf{y}}\}}\right)^{\frac{1}{k}}\right)} \cdot \left(C_1 \ln\left(\frac{C_2 L^2}{m_{\mathbf{x}} m_{\mathbf{y}}}\right)\right)^{k+3} \ln\left(\frac{\|\mathbf{z}_0 - \mathbf{z}^*\|}{\epsilon}\right).$$

(22)

*Proof of Theorem 4.* By Lemma 11, when running RHSS($k$),

$$\|\mathbf{z}_T - \mathbf{z}^*\| \leq \left(1 - \frac{1}{4}\left(\frac{m_{\mathbf{y}}}{L_{\mathbf{xy}}}\right)^{\frac{1}{k}}\right)^T \|\mathbf{z}_0 - \mathbf{z}^*\|.$$

Thus, when $T > 4\left(\frac{L_{\mathbf{xy}}}{m_{\mathbf{y}}}\right)^{1/k} \cdot \ln\left(\frac{\|\mathbf{z}_0 - \mathbf{z}^*\|}{\epsilon}\right)$, one can ensure that $\|\mathbf{z}_T - \mathbf{z}^*\| \leq \epsilon$. Now we can focus on the number of matrix-vector products needed per iteration, which comes in two parts: the cost of calling conjugate gradient and the cost of calling RHSS($k - 1$).

**Conjugate Gradient cost**  The matrix to be solved via conjugate gradient is $\eta\mathbf{P} + \mathbf{G}$. By Lemma 7, its condition number is upper bounded by $\frac{3L_{\mathbf{x}}}{m_{\mathbf{x}}}\left(\frac{m_{\mathbf{y}}}{L_{\mathbf{xy}}}\right)^{1/k}$. By Lemma 10, the number of matrix-vector products needed for calling CG is

$$\left\lceil \sqrt{\frac{3L_{\mathbf{x}}}{m_{\mathbf{x}}}\left(\frac{m_{\mathbf{y}}}{L_{\mathbf{xy}}}\right)^{1/k}} \ln\left(2\sqrt{\frac{3L_{\mathbf{x}}}{m_{\mathbf{x}}}\left(\frac{m_{\mathbf{y}}}{L_{\mathbf{xy}}}\right)^{1/k}} M_1\right)\right\rceil \leq c_1 \sqrt{\frac{L_{\mathbf{x}}}{m_{\mathbf{x}}}\left(\frac{m_{\mathbf{y}}}{L_{\mathbf{xy}}}\right)^{\frac{1}{k}}} \cdot \ln\left(\frac{c_2 L^2}{m_{\mathbf{x}} m_{\mathbf{y}}}\right), \quad (23)$$

for some constants $c_1, c_2 > 0$.

**RHSS($k - 1$) cost**  By Lemma 7, the new saddle point problem involving $\eta\mathbf{P} + \mathbf{S}$ has parameters $m_{\mathbf{x}}' = \eta\alpha$, $m_{\mathbf{y}}' = \eta$, $L_{\mathbf{x}}' = L_{\mathbf{y}}' = 2\eta\beta L_{\mathbf{x}}$, $L_{\mathbf{xy}}' = L_{\mathbf{xy}}$. It is easy to see that $m_{\mathbf{y}}' = \eta \geq m_{\mathbf{y}}$, $m_{\mathbf{x}}' = (m_{\mathbf{x}}/m_{\mathbf{y}})m_{\mathbf{y}}' \geq m_{\mathbf{x}}$, and that $L_{\mathbf{x}}' = L_{\mathbf{y}}' \leq 2L_{\mathbf{x}}$. Thus $L' = \max\{L_{\mathbf{x}}', L_{\mathbf{y}}', L_{\mathbf{xy}}'\} \leq 2L$. Assuming that Theorem 4 holds for RHSS($k - 1$), then the number of matrix-vector products needed for the new saddle point problem can be bounded by

$$\underbrace{\sqrt{\frac{L_{\mathbf{xy}}^2}{m_{\mathbf{x}}' m_{\mathbf{y}}'} + \left(\frac{L_{\mathbf{x}}'}{m_{\mathbf{x}}'} + \frac{L_{\mathbf{y}}'}{m_{\mathbf{y}}'}\right)\left(1 + \left(\frac{L_{\mathbf{xy}}}{\max\{m_{\mathbf{x}}', m_{\mathbf{y}}'\}}\right)^{1/(k-1)}\right)}}_{(a)} \cdot \left(C_1 \ln\left(\frac{C_2 L'^2}{m_{\mathbf{y}}' m_{\mathbf{x}}'}\right)\right)^{k+2} \cdot \underbrace{\ln\left(\frac{4L^2 M_2}{m_{\mathbf{x}}^2}\right)}_{(b)}.$$

Here we used Lemma 9, that when $\|\mathbf{z}_t - \mathbf{z}^*\| \leq \left(\frac{m_{\mathbf{x}}}{L_{\mathbf{x}} + L_{\mathbf{xy}}}\right)^2 \|\mathbf{z}_0 - \mathbf{z}^*\|$, RHSS($k - 1$) returns. Assume that $C_1 > 8$. Note that

$$\frac{L_{\mathbf{xy}}^2}{m_{\mathbf{x}}' m_{\mathbf{y}}'} = \frac{L_{\mathbf{xy}}^2}{\alpha\eta^2} = \frac{L_{\mathbf{xy}}^2}{\frac{m_{\mathbf{x}}}{m_{\mathbf{y}}} \cdot L_{\mathbf{xy}}^{2/k} m_{\mathbf{y}}^{2-2/k}} = \frac{L_{\mathbf{xy}}^2}{m_{\mathbf{x}} m_{\mathbf{y}}}\left(\frac{m_{\mathbf{y}}}{L_{\mathbf{xy}}}\right)^{\frac{2}{k}},$$

$$\frac{L_{\mathbf{xy}}}{m_{\mathbf{y}}'} = \frac{L_{\mathbf{xy}}}{\eta} = \frac{L_{\mathbf{xy}}}{L_{\mathbf{xy}}^{1/k} m_{\mathbf{y}}^{1-1/k}} = \left(\frac{L_{\mathbf{xy}}}{m_{\mathbf{y}}}\right)^{\frac{k-1}{k}}.$$

Therefore

$$(a) \leq \sqrt{\frac{L_{\mathbf{xy}}^2}{m_{\mathbf{x}}m_{\mathbf{y}}}\left(\frac{m_{\mathbf{y}}}{L_{\mathbf{xy}}}\right)^{\frac{2}{k}} + \frac{2L_{\mathbf{x}}}{m_{\mathbf{x}}}\left(\frac{m_{\mathbf{y}}}{L_{\mathbf{xy}}}\right)^{\frac{2}{k}}\cdot\left(1 + \left(\frac{L_{\mathbf{xy}}}{m_{\mathbf{y}}}\right)^{\frac{1}{k}}\right)}$$

$$\leq 2\sqrt{\frac{L_{\mathbf{xy}}^2}{m_{\mathbf{x}}m_{\mathbf{y}}}\left(\frac{m_{\mathbf{y}}}{L_{\mathbf{xy}}}\right)^{\frac{2}{k}} + \frac{L_{\mathbf{x}}}{m_{\mathbf{x}}}\left(\frac{m_{\mathbf{y}}}{L_{\mathbf{xy}}}\right)^{\frac{1}{k}}},$$

$$\ln\left(\frac{C_2 L'^2}{m'_{\mathbf{y}}m'_{\mathbf{x}}}\right) \leq \ln\left(\frac{4C_2 L^2}{m_{\mathbf{x}}m_{\mathbf{y}}}\right) \leq 2\ln\left(\frac{C_2 L^2}{m_{\mathbf{x}}m_{\mathbf{y}}}\right),$$

$$(b) \leq \ln\left(\frac{64 L^4}{m_{\mathbf{x}}^2 m_{\mathbf{y}}^2}\right) \leq 2\ln\left(\frac{8L^2}{m_{\mathbf{x}}m_{\mathbf{y}}}\right).$$

Thus the cost of calling RHSS$(k-1)$ is at most

$$4C_1^{k+2}\ln^{k+3}\left(\frac{C_2 L^2}{m_{\mathbf{x}}m_{\mathbf{y}}}\right)\sqrt{\frac{L_{\mathbf{xy}}^2}{m_{\mathbf{x}}m_{\mathbf{y}}}\left(\frac{m_{\mathbf{y}}}{L_{\mathbf{xy}}}\right)^{\frac{2}{k}} + \frac{L_{\mathbf{x}}}{m_{\mathbf{x}}}\left(\frac{m_{\mathbf{y}}}{L_{\mathbf{xy}}}\right)^{\frac{1}{k}}}. \tag{24}$$

In the case where $k = 2$, RHSS$(k-1)$ is exactly Proximal Best Response (Algorithm 4). Hence, by Theorem 3, the number of matrix-vector products needed is at most

$$O\left(\sqrt{\frac{L_{\mathbf{xy}}\cdot\max\{L_{\mathbf{xy}}, L'\}}{m'_{\mathbf{x}}m'_{\mathbf{y}}} + \frac{L'_{\mathbf{x}}}{m'_{\mathbf{x}}} + \frac{L'_{\mathbf{y}}}{m'_{\mathbf{y}}}}\cdot\ln^4\left(\frac{L'^2}{m'_{\mathbf{x}}m'_{\mathbf{y}}}\right)\ln\left(\frac{L'M_2}{m'_{\mathbf{x}}m'_{\mathbf{y}}}\right)\right)$$

$$=O\left(\sqrt{\frac{L_{\mathbf{xy}}}{m_{\mathbf{x}}} + \frac{L_{\mathbf{x}}\sqrt{m_{\mathbf{y}}}}{m_{\mathbf{x}}\sqrt{L_{\mathbf{xy}}}}}\ln^5\left(\frac{L^2}{m_{\mathbf{x}}m_{\mathbf{y}}}\right)\right).$$

By this, we mean there exists constants $c_3, c_4 > 0$ such that the number of matrix-vector products needed is

$$c_3\sqrt{\frac{L_{\mathbf{xy}}}{m_{\mathbf{x}}} + \frac{L_{\mathbf{x}}\sqrt{m_{\mathbf{y}}}}{m_{\mathbf{x}}\sqrt{L_{\mathbf{xy}}}}}\ln^5\left(\frac{c_4 L^2}{m_{\mathbf{x}}m_{\mathbf{y}}}\right).$$

Thus, (24) also holds for $k = 2$, provided that $C_2 \geq c_4$ and $C_1 \geq c_3$.

**Total cost.** By combining (23) and (24), we can see that the cost (i.e. number of matrix-vector products) of RHSS$(k)$ per iteration is

$$\left(4C_1^{k+2} + c_1\right)\ln^{k+3}\left(\frac{\max\{c_2, C_2\}L^2}{m_{\mathbf{x}}m_{\mathbf{y}}}\right)\sqrt{\frac{L_{\mathbf{xy}}^2}{m_{\mathbf{x}}m_{\mathbf{y}}}\left(\frac{m_{\mathbf{y}}}{L_{\mathbf{xy}}}\right)^{\frac{2}{k}} + \frac{L_{\mathbf{x}}}{m_{\mathbf{x}}}\left(\frac{m_{\mathbf{y}}}{L_{\mathbf{xy}}}\right)^{\frac{1}{k}}}.$$

Let us choose $C_2 > \max\{c_2, 8\}$ and $C_1 > \max\{c_1, 20\}$. Then, in order to ensure that $\|\mathbf{z}_T - \mathbf{z}^*\| \leq \epsilon$, the number of matrix-vector products that RHSS$(k)$ needs is

$$4\left(\frac{L_{\mathbf{xy}}}{m_{\mathbf{y}}}\right)^{1/k}\ln\left(\frac{\|\mathbf{z}_0 - \mathbf{z}^*\|}{\epsilon}\right)\cdot\left(4C_1^{k+2} + c_1\right)\ln^{k+3}\left(\frac{C_2 L^2}{m_{\mathbf{x}}m_{\mathbf{y}}}\right)\sqrt{\frac{L_{\mathbf{xy}}^2}{m_{\mathbf{x}}m_{\mathbf{y}}}\left(\frac{m_{\mathbf{y}}}{L_{\mathbf{xy}}}\right)^{\frac{2}{k}} + \frac{L_{\mathbf{x}}}{m_{\mathbf{x}}}\left(\frac{m_{\mathbf{y}}}{L_{\mathbf{xy}}}\right)^{\frac{1}{k}}}$$

$$\leq 20C_1^{k+2}\ln\left(\frac{\|\mathbf{z}_0 - \mathbf{z}^*\|}{\epsilon}\right)\ln^{k+3}\left(\frac{C_2 L^2}{m_{\mathbf{x}}m_{\mathbf{y}}}\right)\sqrt{\frac{L_{\mathbf{xy}}^2}{m_{\mathbf{x}}m_{\mathbf{y}}}\left(\frac{m_{\mathbf{y}}}{L_{\mathbf{xy}}}\right) + \frac{L_{\mathbf{x}}}{m_{\mathbf{x}}}\left(\frac{L_{\mathbf{xy}}}{m_{\mathbf{y}}}\right)^{\frac{1}{k}}}$$

$$\leq \sqrt{\frac{L_{\mathbf{xy}}^2}{m_{\mathbf{x}}m_{\mathbf{y}}} + \left(\frac{L_{\mathbf{x}}}{m_{\mathbf{x}}} + \frac{L_{\mathbf{y}}}{m_{\mathbf{y}}}\right)\left(1 + \left(\frac{L_{\mathbf{xy}}}{m_{\mathbf{y}}}\right)^{1/k}\right)}\cdot\left(C_1\ln\left(\frac{C_2 L^2}{m_{\mathbf{x}}m_{\mathbf{y}}}\right)\right)^{k+3}\ln\left(\frac{\|\mathbf{z}_0 - \mathbf{z}^*\|}{\epsilon}\right).$$

$$\square$$

We now discuss how to choose the optimal $k$. Observe that

$$(22) \leq \sqrt{\frac{L_{\mathbf{xy}}^2}{m_{\mathbf{x}}m_{\mathbf{y}}} + \frac{L_{\mathbf{x}}}{m_{\mathbf{x}}} + \frac{L_{\mathbf{y}}}{m_{\mathbf{y}}}}\ln\left(\frac{\|\mathbf{z}_0 - \mathbf{z}^*\|}{\epsilon}\right)\cdot\underbrace{\left(\frac{L^2}{m_{\mathbf{x}}m_{\mathbf{y}}}\right)^{\frac{1}{2k}}\left(C_1\ln\left(\frac{C_2 L^2}{m_{\mathbf{x}}m_{\mathbf{y}}}\right)\right)^{k+3}}_{(a)}.$$

Compared to the lower bound, there is only one additional factor $(a)$, whose logarithm is

$$\ln\left(\left(\frac{L^2}{m_{\mathbf{x}}m_{\mathbf{y}}}\right)^{\frac{1}{2k}}C_1^{k+3}\ln^{k+3}\left(\frac{C_2 L^2}{m_{\mathbf{x}}m_{\mathbf{y}}}\right)\right) = \frac{1}{2k}\ln\left(\frac{L^2}{m_{\mathbf{x}}m_{\mathbf{y}}}\right) + (k+3)\ln\left(C_1\ln\left(\frac{C_2 L^2}{m_{\mathbf{x}}m_{\mathbf{y}}}\right)\right),$$

which is minimized when $k = \sqrt{\frac{\ln\left(\frac{L^2}{m_{\mathbf{x}}m_{\mathbf{y}}}\right)}{2\ln\left(C_1\ln\left(\frac{L^2}{m_{\mathbf{x}}m_{\mathbf{y}}}\right)\right)}}$, and the minimum value is

$$3\ln\left(C_1\ln\left(\frac{C_2 L^2}{m_{\mathbf{x}}m_{\mathbf{y}}}\right)\right) + \sqrt{\frac{1}{2}\ln\left(\frac{L^2}{m_{\mathbf{x}}m_{\mathbf{y}}}\right)\ln\left(C_1\ln\left(\frac{C_2 L^2}{m_{\mathbf{x}}m_{\mathbf{y}}}\right)\right)} = o\left(\ln\left(\frac{L^2}{m_{\mathbf{x}}m_{\mathbf{y}}}\right)\right).$$

I.e. $(a)$ is sub-polynomial in $\frac{L^2}{m_{\mathbf{x}}m_{\mathbf{y}}}$. This proves our corollary that, when $k = \Theta\left(\sqrt{\ln\left(\frac{L^2}{m_{\mathbf{x}}m_{\mathbf{y}}}\right)/\ln\ln\left(\frac{L^2}{m_{\mathbf{x}}m_{\mathbf{y}}}\right)}\right)$, the number of matrix vector products that RHSS$(k)$ needs to find $\mathbf{z}_T$ such that $\|\mathbf{z}_T - \mathbf{z}^*\| \le \epsilon$ is

$$\sqrt{\frac{L_{\mathbf{xy}}^2}{m_{\mathbf{x}}m_{\mathbf{y}}} + \frac{L_{\mathbf{x}}}{m_{\mathbf{x}}} + \frac{L_{\mathbf{y}}}{m_{\mathbf{y}}}}\ln\left(\frac{\|\mathbf{z}_0 - \mathbf{z}^*\|}{\epsilon}\right) \cdot \left(\frac{L^2}{m_{\mathbf{x}}m_{\mathbf{y}}}\right)^{o(1)}.$$

## Footnotes

[1] Here $g(\mathbf{x}, \mathbf{y})$ refers to the argument passed to Algorithm 3, which in our case has the form $f(\mathbf{x}, \mathbf{y}) + \beta \|\mathbf{x} - \hat{\mathbf{x}}_{t'-1}\|^2$.

[2]Here it is assumed that $\epsilon$ is sufficiently small, i.e. $\epsilon \leq \max\{L_{\mathbf{xy}}, m_{\mathbf{x}}\}D_{\mathbf{y}}^2$.