[Reviews · NeurIPS 2020]

Review 1

Summary and Contributions: This paper studies the gradient complexity of strongly convex-strongly concave minimax optimization. The algorithms introduced in this paper improve on existing bounds for minimax optimization of f(x,y) when the interaction term between x and y is relatively small (in particular, when it is less than the smoothness term for each of x and y individually). The paper additionally obtains a nearly tight (up to sub-polynomial factors) bound when f is quadratic. Post-rebuttal: Thanks for your response.

Strengths: The paper introduces some neat ideas which improve upon existing work for the fundamental problem of strongly convex-strongly concave minimax optimization: - It introduces the idea of using (an approximate version of) alternating best response as the inner loop of a (triple-loop) algorithm. Prior work (e.g., Lin et al., [19]) had some form of accelerated proximal point or accelerated gradient method for all 3 loops. - The obtained bounds obtain the tight log(1/eps) dependence on eps, in contrast to prior work where it was log^3(1/eps). - The paper's result for the quadratic case shows that proving stronger lower bounds than what are currently known will require using non-quadratic functions, which seems to require new techniques.

Weaknesses: - The upper bounds established in the paper do not quite match the best-known lower bounds (for the general strongly convex-strongly concave case). - The problem studied by this paper is somewhat esoteric; often in minimax optimization one phrases bounds solely in terms of the maximum L of the Lipschitz constants L_x, L_y, L_{xy}, in which case there is no difference between this paper's bounds and prior work. [post-rebuttal: apart from the improvement in log(1/eps) factors]

Correctness: I did not find any issues with correctness.

Clarity: The paper is very well-written, and does a good job of explaining the idea behind the algorithms and the analysis in the main body. Minor comment: line 165: extra "for" here

Relation to Prior Work: This is adequately addressed.

Reproducibility: Yes

Additional Feedback:


Review 2

Summary and Contributions: The paper studies proposes a first-order algorithm for solving convex-concave min-max problems with near optimal iteration complexity.

Strengths: The analysis and algorithm in the quadratic case seems novel and interesting. It would have been a better fit if the relation to machine learning NeurIPS community had discussed better in details.

Weaknesses: The reviewer has a few questions/concerns regarding the submitted manuscript: - The paper seems to be more complex than [19]. It contains at least 4 loops. This fact should be discussed in the paper. Also since the logarithmic factors are important in comparing with [19], the authors should avoid using \tilde{O} notation. In particular, in the abstract, they should reveal the complete iteration complexity of the proposed method and compare it with [19] in O(.) notation and not \tilde{O} notation. - The lower bound developed in [37] is for gradient oracle algorithms. It is not about matrix-vector product. However, section 5 claims that it achieves this lower-bound by counting the matrix-vector product operations. Please explain why these two are equivalent in your problem. - Can you highlight the differences between the lower bound proposed in [28] and [37]? Are they the same? One of them is briefly mentioned in the paper but not used while the other one is used in the plot. - Motivation behind the class of studied problem should be discussed. When does this problem appear in practice? - Although the problem setting is different, the algorithm seems very similar to the algorithm proposed in Ostrovskii, D. M., Lowy, A., & Razaviyayn, M. (2020). Efficient search of first-order nash equilibria in nonconvex-concave smooth min-max problems. arXiv preprint arXiv:2002.07919. Can you highlight the differences? - As a minor concern, it would help the reader if the paper can provide the constants throughout the paper. For example, what is o(1) in coro 3? What are the constants in Theorem 4? ====== Edit after the rebuttal: I read the authors rebuttal and I think this paper is above the acceptance threshold and hence I am keeping my score.

Correctness: Please see the previous comment.

Clarity: The presentation can improve significantly. For example, AGD algorithm is not described (which version of accelerated gradient descent does the paper refer to?). What is the function g(.) in algorithm 1? If the authors assume Lx = Ly, then why defining it separately and make the notations complex?

Relation to Prior Work: In addition to the papers mentioned above, the relation to the literature of monotone VI should be discussed in details. The convex-concave min-max falls into the category of monotone VIs and there are many works in the literature addressing that. For example, the following papers should be discussed: Ronald E Bruck Jr. On the weak convergence of an ergodic iteration for the solution of variational inequalities for monotone operators in hilbert space. Journal of Mathematical Analysis and Applica- tions, 61(1):159–164, 1977. Andrzej Ruszczynski. A partial regularization method for saddle point seeking. 1994. Arkadi Nemirovski. Efficient methods for solving variational inequalities. 17:344–359, 1981. Yurii Nesterov. Dual extrapolation and its applications to solving variational inequalities and related problems. Mathematical Programming, 109(2-3):319–344, 2007. Renato DC Monteiro and Benar Fux Svaiter. On the complexity of the hybrid proximal extragradient method for the iterates and the ergodic mean. SIAM Journal on Optimization, 20(6):2755–2787, 2010. Anatoli Juditsky and Arkadi Nemirovski. Solving variational inequalities with monotone operators on domains given by linear minimization oracles. Mathematical Programming, 156(1-2):221–256, 2016. Gauthier Gidel, Tony Jebara, and Simon Lacoste-Julien. Frank-wolfe algorithms for saddle point problems. In Artificial Intelligence and Statistics, pages 362–371, 2017. E Yazdandoost Hamedani, A Jalilzadeh, NS Aybat, and UV Shanbhag. Iteration complexity of randomized primal-dual methods for convex-concave saddle point problems. arXiv preprint arXiv:1806.04118, 2018. Niao He, Anatoli Juditsky, and Arkadi Nemirovski. Mirror prox algorithm for multi-term com- posite minimization and semi-separable problems. Computational Optimization and Applications, 61(2):275–319, 2015. Paul Tseng. On accelerated proximal gradient methods for convex-concave optimization. submitted to SIAM Journal on Optimization, 2:3, 2008. Le Thi Khanh Hien, Renbo Zhao, and William B Haskell. An inexact primal-dual smoothing frame- work for large-scale non-bilinear saddle point problems. arXiv preprint arXiv:1711.03669, 2017.

Reproducibility: Yes

Additional Feedback: Section 5 and the relation to lower bound requires further discussion as explained in the previous comments.


Review 3

Summary and Contributions: This paper is an improvement of Lin et al. [19] that aims to close the gap towards the lower bound in strongly-convex-strongly-concave minimax optimization, proposed by Zhang et al. [37]. --after rebuttal-- I have carefully read the author response and my questions are well addressed. Overall the proofs could have been better written and the related work should have been better addressed, but after the rebuttal I believe the proofs should be correct. However, this theoretical work is still somehow incremental for me as the improvements over the condition number and log(1/epsilon) (I agree with R3 that tilde{O} notation should be modified) are small. I will change the score to 6 due to the value of this paper on improving the best upper bound.

Strengths: This paper improves previous results by Lin et al. [19] by having linear convergence and a tighter condition number bound. It combines Nesterov's momentum (in the weakly coupled case) with Accelerated Proximal Point Algorithm (APPA) in Lin et al. (strongly coupled case) to produce a new algorithm (APPA-ABR) that has a better upper bound when the two sets of variables are weakly coupled. The lower bound by Zhang et al. [37] is achieved for quadratic cases up to sub-polynomial factors using the Hermitian-Skew-Hermitian-Split algorithm in [5]. It should be relevant to the NeurIPS community as an interesting attempt towards closing the gap in strongly-convex-strongly-concave minimax optimization.

Weaknesses: I feel the improvement is not quite strong as there are many other gradient algorithms for fast linear convergence in the same settings and the authors should have compared more carefully (see the relation part). The difference of the condition numbers compared to Lin et al. is not large. Moreover, this problem in fact has limited practicability since most applications in machine learning are non-convex. Finally, there are some parts of the proofs that seem hard to understand (see below).

Correctness: Proof: The outline of the proof in the main paper seems plausible for me. However, some parts of the proofs in the appendices are difficult to follow. At least the authors should include them somewhere in the appendices. For example: 1) Line 133: why can we change the order? Is it from Sion's theorem? But this theorem only holds if the domains are compact. From eq. (1) the authors are dealing with unconstrained cases. Maybe strong-convexity could alleviate this assumption? At least the authors should clarify. 2) Supplementary Line 53: How do we derive the theorem for this inequality? I could only get something like T = log(\epsilon/4 * \sqrt{L_x/min{m_x, m_y}}). 3) Supplementary Line 38 and 39: \tilde{x}_{t+1} is defined twice; are they the same? 4) Supplementary Lemma 3: Why does this lemma follow directly from [3, Thm 4.1]? I checked their proof but they did not seem to give such a sequence in Appendix B.2. More details should be written. 5) Supp. Line 76: you said both hold for t = 0, but is x^*_0 defined anywhere?

Clarity: The paper is mostly well written except that there are a few typos: 1) Line 46, "Hermition-..." 2) Line 79, "\nabla f(x^*, y^*) = 0", 0 should boldface 3) Line 80, "an close" 4) Theorem 1: L_{xy} < 1/2 \sqrt{m_x m_y}. Should be less or equal to? Because in Line 138 you said L_xy \leq 1/2 \sqrt{2\beta 2\beta} and we could use ABR. 5) Line 140: is the third factor \tilde{O}(\sqrt{L/L_{xy}} a typo? Because before you mentioned \tilde{O}(\sqrt{L_x/L_{xy}}. 6) Line 199: "need to" -> "needs to", and what does "||z_T - z^*||\leq \epsilon-saddle point" mean? 7) Line 165: "for the reduction is for ..." 8) Capitalization in the reference, such as "Wasserstein gan", "skew-hermitian" in [2], [5]. 9) Supplementary Line 25: L_{xy^2}.

Relation to Prior Work: This paper discussed the difference from Lin et al., but it seems that a more thorough literature review could have been done: 1) (Strongly) convex-concave minimax can also formulated as (strongly) monotone variational inequality (VI). Related work should be cited for algorithms that solve VI. See e.g. https://arxiv.org/pdf/1908.08465.pdf and the references therein. 2) There is another paper https://arxiv.org/pdf/1906.07300.pdf that also addresses the lower bound for minimax games and especially quadratic games. How is your work related to it? 3) Line 31: "their depedence is far from optimal". There are many recent algorithms that has linear convergence for strongly-convex-strongly-concave (SCSC) minimax problems, such as the ones mentioned in Sec. 3. How close are they w.r.t. the lower bound? 4) This paper seems to provide an algorithm that matches the lower bound for strongly monotone VIs: https://arxiv.org/pdf/2001.00602.pdf. How is your work related? [1]: Linear Lower Bounds and Conditioning of Differentiable Games, Ibrahim et al., https://arxiv.org/pdf/1906.07300.pdf, ICML 2020. [2]: Accelerating Smooth Games by Manipulating Spectral Shapes, Azizian et al., https://arxiv.org/pdf/2001.00602.pdf.

Reproducibility: No

Additional Feedback: I have another question for the authors: 1) Line 187: I know the goal is to prove a matching upper bound for quadratic games, but why don't we just solve by Conjugate Gradient (we could symmetrize it by squaring) in line 174, since you are already using CG in your proof? I don't see a strong motivation of this algorithm from a practical point of view.

[Author Response · NeurIPS 2020]

**Reviewer 1:** Thanks for your encouraging comments. We would like to note that even when $L_{\mathbf{x}} = L_{\mathbf{xy}} = L_{\mathbf{y}}$, our
work still improves the result of Lin et al.'s by reducing the dependence on $\ln(1/\epsilon)$.

**Reviewer 3:** Thank you for the detailed comments and the pointers to several related papers.

1. For a quadratic function $\frac{1}{2}\mathbf{x}^T \mathbf{A}\mathbf{x} - \mathbf{b}^T\mathbf{x}$, querying the gradient oracle returns $\mathbf{A}\mathbf{x} - \mathbf{b}$, i.e. the matrix-vector product
(plus a constant). In other words, one can implement a matrix-vector product oracle via the gradient oracle, and vice
versa. Thus counting the number of matrix-vector products is indeed equivalent to counting the gradient complexity.
2. The lower bound in [28] is for the convex-concave setting, while the lower bound in [37] is for the strongly
convex-strongly concave setting.
3. Motivation behind the class of studied problems: the class of strongly convex-strongly concave minimax problems is
a fundamental class of minimax problems and has been studied extensively in the literature. Moreover, efficient
algorithms for this class can be translated to efficient algorithms for more general strongly convex-concave and
convex concave problems (via the reduction developed in [19]).
4. Thank you for bringing our attention to the paper by Ostrovskii et al., whose algorithm also relies on solving proximal
problems in a multi-loop manner. However, there are important differences between our algorithm and theirs. The
outer loop of their algorithm updates both $\mathbf{x}$ and $\mathbf{y}$ by solving a proximal point problem (without momentum), while
the two inner loops use accelerated gradient method to solve the regularized problem.In comparison, The two outer
loops of our algorithm use accelerated proximal method (with momentum) on $\mathbf{x}$ and $\mathbf{y}$ respectively to reduce the
problem to solving a regularized problem, and the innermost loop solves this using the alternating best response
scheme. The analysis and the results of both papers are also quite different.
5. The explicit form of the subpolynomial factor in Corollary 3 can be found on line 309 of the SM.
6. The full description of AGD is deferred to page 3 of the SM due to lack of space. Our final bounds hold for any $L_{\mathbf{x}}$
and $L_{\mathbf{y}}$. We only assume $L_{\mathbf{x}} = L_{\mathbf{y}}$ for ease of presentation. It is without loss of generality as the more general case
$L_{\mathbf{x}} \neq L_{\mathbf{y}}$ can be handled by scaling (see Line 70-73).
7. **Relation to Prior Work:** Thank you for bringing our attention to the literature on variational inequalities. We will
cite and discuss them accordingly in our revision.

**Reviewer 4:** Thank you for your detailed review comments.

**About correctness of the proof:**

1) Yes, the minimax theorem holds for non-compact sets in the strongly convex-strongly concave case; see e.g. Hartung,
An extension of Sion's minimax theorem with an application to a method for constrained games.
2) We can prove Theorem 1 from line 53 by multiplying the upper bound of $T$ with the number of gradient evaluations
per $t$, which is given in the algorithm $(2\sqrt{\kappa_{\mathbf{x}}}\ln(24\kappa_{\mathbf{x}}) + 2\sqrt{\kappa_{\mathbf{y}}}\ln(24\kappa_{\mathbf{y}}))$.
3) $\tilde{\mathbf{x}}_t$ is not used in the proof. We will remove both definitions in our revision as they are redundant.
4) We meant to cite an earlier version of Lin et al.'s paper (2002.02417v1), where the sequence $\{\Lambda_t\}$ appears in the
proof of Lemma B.1 (page 24) as $\{\Lambda_t(\mathbf{x}^*)\}$.
5) This is a typo; it should be $t = 1$ instead of $t = 0$.
We will clarify the above points and fix the typos in the next version.

**About comparison to related work:** Thank you for the pointers and we will cite them accordingly in our revision.

1) We agree that previous work on monotone variational inequalities are very relevant.
2) Thanks for pointing it out. Indeed, this paper provides the same lower bound as in [37], although it only stated it for
a weaker class of algorithms. However, we have verified that this lower bound in fact holds for a larger class of
algorithms, and is directly comparable to our upper bounds. We will mentioned this explicitly in the next version.
3) These algorithms have suboptimal dependency on the condition number: The bound for GDA and Hamiltonian Gra-
dient Descent is $O((\frac{L}{m_{\mathbf{x}}} + \frac{L}{m_{\mathbf{y}}})^2 \ln(\frac{1}{\epsilon}))$, while the bound for ExtraGradient and OGDA is $O\left(\left(\frac{L}{m_{\mathbf{x}}} + \frac{L}{m_{\mathbf{y}}}\right)\ln(\frac{1}{\epsilon})\right)$.
Suppose that $m_{\mathbf{x}} = 1$, $m_{\mathbf{y}} = A^2$, $L = A^4$ ($A \gg 1$), then the two upper bounds become $O(A^8 \ln(1/\epsilon))$ and
$O(A^4 \ln(1/\epsilon))$ respectively, while the lower bound is $\Omega(A^3 \ln(1/\epsilon))$.
4) First, as we understand it, Azizian et al. only provide local convergence guarantees, while we can show global
convergence. Second, the spectral shape framework does not fully capture the properties of a minimax optimization
problem. E.g., the spectral shape does not reflect both $m_{\mathbf{x}}$ and $m_{\mathbf{y}}$, only $\min\{m_{\mathbf{x}}, m_{\mathbf{y}}\}$. Consequently, a minimax
optimization algorithm that is optimal for its spectral shape could still be suboptimal if $m_{\mathbf{x}}$ and $m_{\mathbf{y}}$ are very different.
**About using CG to solve the quadratic problem:** One can indeed solve $\mathbf{J}\mathbf{z} = \mathbf{b}$ by solving the squared equation
$\mathbf{J}^T\mathbf{J}\mathbf{z} = \mathbf{J}^T\mathbf{b}$. However the condition number of $\mathbf{J}^T\mathbf{J}$ can be very large, so the resulting complexity bound, which is
$\tilde{O}\left(\frac{L}{\min\{m_{\mathbf{x}}, m_{\mathbf{y}}\}}\right)$, has suboptimal dependence on condition numbers and is much worse than our result.

[Meta-Review · NeurIPS 2020]

The paper studies minimax optimization for strongly convex / strong concave objectives. Authors propose a novel algorithm that achieves linear convergence rate and improved dependence on the condition numbers, compared to prior work. Overall, an interesting paper. Accept!